# Super-fast rates of convergence for Neural Networks Classifiers under the Hard Margin Condition

**Nathanael Tepakbong**                                                *ntepakbo-c@my.cityu.edu.hk*
*Department of Data Science*
*City University of Hong Kong*

**Xiang Zhou**                                                *xiang.zhou@cityu.edu.hk*
*Department of Mathematics*
*City University of Hong Kong*

**Ding-Xuan Zhou**                                                *dingxuan.zhou@sydney.edu.au*
*School of Mathematics and Statistics*
*The University of Sydney*

**Reviewed on OpenReview:** *https://openreview.net/forum?id=HXun3l0Feu*

## Abstract

We study the classical binary classification problem for hypothesis spaces of Deep Neural Networks (DNNs) under Tsybakov's low-noise condition with exponent $q > 0$, as well as its limit case $q = \infty$, which we refer to as the *hard margin condition*. We demonstrate that, for a wide range of commonly used activation functions (including but not limited to ReLU, LeakyReLU, ELU, CELU, SELU, Softplus, GELU, SiLU, Swish, Mish, and Softmax), DNN solutions to the empirical risk minimization (ERM) problem with square loss surrogate and $\ell_p$ penalty on the weights ($0 < p < \infty$) can achieve excess risk bounds of order $\mathcal{O}\left(n^{-\alpha}\right)$ for $\alpha$ close to 1 under the low-noise condition, and for arbitrarily large $\alpha > 1$ under the hard-margin condition, provided that the Bayes regression function $\eta$ satisfies a *distribution-adapted smoothness* condition relative to the marginal data distribution $\rho_X$. Furthermore, when the activation function is chosen as tanh or sigmoid, we show that the same rates follow from the standard assumption that $\eta \in \mathcal{C}^s$. Finally, we establish minimax lower bounds, showing that these rates cannot be improved upon whenever $q \geq 2$. Our proof relies on a novel decomposition of the excess risk for general ERM-based classifiers which might be of independent interest.

## 1 Introduction

In this article, we study the problem of classifying high-dimensional data points with binary labels. It is well-known that, in the absence of structural assumptions about the data or the underlying model, convergence rates for classification tasks typically decay as $\mathcal{O}\left(n^{-c/d}\right)$ for some constant $c > 0$, which becomes arbitrarily slow as the dimensionality $d$ increases. This phenomenon is often referred to as the curse of dimensionality (CoD). However, it has been observed that many models used in practice — particularly Deep Neural Networks in recent years — are capable of efficiently solving extremely high-dimensional classification tasks, achieving convergence rates that appear to defy the CoD (Goodfellow et al., 2016; Krizhevsky et al., 2012).

This gap between theoretical results and practical observations can often be bridged by introducing suitable regularity assumptions on the problem. In the context of supervised binary classification, such assumptions frequently take the form of *margin conditions*. First introduced in the seminal work of (Mammen and Tsybakov, 1999), margin conditions characterize the behavior of the data distribution near the decision boundary — the region where classification is most challenging. Over the years, these conditions have enabled the derivation of CoD-free rates of convergence for classifiers based on various hypothesis spaces

(Tsybakov, 2004; Audibert and Tsybakov, 2007). Remarkably, margin conditions can not only eliminate the curse of dimensionality, but can also lead to "fast" rates of convergence — faster than the standard $\mathcal{O}\left(n^{-1/2}\right)$ — and, under their strongest form, even "super-fast" rates, exceeding $\mathcal{O}\left(n^{-1}\right)$.

Notable examples of hypothesis spaces for which these super-fast (sometimes even exponential) rates of convergence have been observed include local polynomial estimators (Audibert and Tsybakov, 2007), support vector machines (Steinwart and Scovel, 2005; Steinwart and Christmann, 2008; Cabannnes and Vigogna, 2023) or Reproducing Kernel Hilbert Spaces (RKHS) (Koltchinskii and Beznosova, 2005; Smale and Zhou, 2007; Vigogna et al., 2022). More recently, it has even been shown that for data coming from an infinite-dimensional Hilbert space, the Delaigle-Hall condition (Delaigle and Hall, 2012), which can be thought of as an infinite-dimensional analogue of the classical margin conditions, can lead to super-fast rates of convergence for RKHS classifiers (Wakayama and Imaizumi, 2024).

Perhaps surprisingly however, for Deep Neural Networks (DNNs), no such "super-fast" rates of convergence have been shown to hold, even under the strongest margin and regularity assumptions. This fact seemingly contradicts the observation that DNNs outperform all other traditional methods by far when it comes to high-dimensional classification. This naturally raises the question: are Neural Networks truly inferior to traditional classification methods in the hard-margin regime? In this work, we answer negatively to this question by showing "super-fast" rates of convergence for DNN classifiers under the hard-margin condition. Before presenting our setup and results in greater detail, we briefly review related literature in the following section.

## 1.1 Related works

When considering a binary classification problem on $[0,1]^d$ with labels $\{1,-1\}$, there are different possible objects which can be used to characterize the regularity of the problem:

- the Bayes regression function $\eta : x \in [0,1]^d \mapsto \mathbb{E}[Y \mid X = x]$ which, up to an affine transformation, represents the conditional probability of $\{Y = 1\}$ given $\{X = x\}$,

- the Bayes classifier $c$ induced by the Bayes regression function: $c : x \mapsto \text{sign}(\eta(x))$. It is the optimal classifier in the sense that it minimizes the expected 0-1 loss over all admissible classifiers, and is therefore what we are implicitly trying to learn.

- the decision region $\Omega := c^{-1}(\{1\})$ and the induced decision boundary $\partial\Omega$.

The margin condition which we refer to in this work, and originally introduced in (Mammen and Tsybakov, 1999), assumes that for all $t > 0$, $\mathbb{P}(|\eta(X)| \leq t) \lesssim t^q$, where $q > 0$ is a constant called the *margin exponent* (note that depending on the source, this is also referred to as a *low-noise* condition). In (Kim et al., 2021), it has been shown that such a margin condition coupled with additional assumptions on respectively the regression function $\eta$, the decision boundary $\partial\Omega$, or the probability for data points to be near the decision boundary $\partial\Omega$, leads to minimax optimal fast rates of convergence for sparse DNN classifiers obtained by hinge-loss empirical risk minimization. For instance, if $\eta$ is assumed to be Hölder continuous, they prove the excess risk bound:
$$\mathcal{E}(\hat{f}_{DNN}) \lesssim \left(\frac{\log^3 n}{n}\right)^{\frac{\beta(q+1)}{\beta(q+2)+d}},$$
where $\beta$ is the Hölder exponent of $\eta$. As we can see, when the margin exponent $q \to \infty$, their result leads to the "fast rate" of $\mathcal{O}\left(n^{-1}\right)$.

In a similar vein, by assuming different kinds of regularity on these objects, and leveraging recent advances on the approximation rates and complexity measures of DNNs hypothesis spaces, various minimax optimal rates of convergence of this kind have been obtained for DNNs under different settings. A non-exhaustive list of such works includes (Feng et al., 2021; Meyer, 2023; Petersen and Voigtlaender, 2021; Bos and Schmidt-Hieber, 2022; Hu et al., 2025; Ko et al., 2023). As it has been mentioned earlier, while these results for DNNs clearly highlight their ability to generalize with CoD-free rates of convergence, none of them obtain a

rate faster than $\mathcal{O}\left(n^{-1}\right)$, even under the most idealized regularity assumptions, unlike the more traditional methods.

To the best of the authors' knowledge, it has only been shown in (Hu et al., 2021) that the *hard-margin condition* (which can informally be seen as the limit $q = \infty$ of Tsybakov's low-noise condition), can lead to exponential rates of convergence for the excess risk for Neural Networks: they prove the result for shallow networks in the Neural Tangent Kernel (NTK) regime (Jacot et al., 2018), which are trained to minimize the Empirical Risk with square loss surrogate. (Nitanda and Suzuki, 2021) similarly show how, in the NTK regime, the hard-margin condition leads to an exponential convergence of the averaged stochastic gradient descent (SGD) algorithm with respect to the number of epochs. However, these results are not fully satisfactory, as it is known that the NTK regime does not accurately represent the expressive power of deeper Networks (Bietti and Bach, 2021). This work is thus, to the best of our knowledge, the first to prove super-fast rates of convergence for conventional DNNs hypothesis spaces under the hard-margin condition.

## 1.2 Our Contributions

We study the binary classification problem over a hypothesis space of parametric functions. The classifiers are learned in a standard supervised learning fashion by minimizing an empirical risk with the square loss as a surrogate and an $\ell_p$ penalty on the model's weights, where $0 < p < \infty$. For a real-valued, measurable function $f$, denote the *excess risk* $\mathcal{E}(f)$ of the classifier induced by $f$ as

$$\mathcal{E}(f) := \mathbb{P}_{(X,Y)\sim\rho}\left(\text{sign } f(X) \neq Y\right) - \mathbb{P}_{(X,Y)\sim\rho}\left(c^*(X) \neq Y\right),$$

where $c^*$ is the Bayes classifier, induced by the Bayes regression function $\eta$. Our main contributions can be stated as follows:

- In Theorem 2, we provide a novel error decomposition for the excess risk of classifiers induced by general classes of parametric functions under both "weak" ($q > 0$) and "hard" ($q = \infty$) margin conditions. The proof is elementary and relies on an inequality we learned from (Vigogna et al., 2022).

- As a direct application of Theorem 2, we show in Theorem 4 that when the hypothesis space consists of DNNs with ReLU activation, and the regression function $\eta$ satisfies a *distribution-adapted smoothness* condition relative to the marginal data distribution $\rho_X$, the excess risk $\mathcal{E}(\hat{f}_{NN})$ converges at a rate $\mathcal{O}\left(n^{-\alpha}\right)$. Specifically, $\alpha \to 1/r$ as $s \to \infty$ under the weak-margin condition, and $\alpha \to \infty$ as $s \to \infty$ under the hard-margin assumption. Here, $r > 1$ roughly quantifies the flatness of the loss landscape near minmizers, and $s > 0$ quantifies how efficiently $\eta$ can be approximated by the DNN space and can be informally interpreted as a smoothness parameter. Minimax lower bounds are provided in Theorem 6, showing that our results are close to optimal.

- Leveraging a result of (Zhang et al., 2024a), we extend Theorem 4 to DNNs with activation function in a very broad class, including most commonly used activation functions, such as LeakyReLU, ELU, CELU, GELU, Softplus, Swish or Mish. This result is stated as Corollary 1. We furthermore show in Theorem 8 that if the activation function is chosen as tanh or sigmoid, the same rates hold under the standard assumption that $\eta \in \mathcal{C}^s$, instead of the distribution-adapted smoothness assumption.

- Lastly, we apply Theorem 2 again to a simplified version of the teacher-student setting, which we recast as a binary classification problem in which the training labels are given by fuzzy predictions of a "teacher" neural network: we show in Theorem 10 that if the teacher network is realizable by the student network, then the excess risk $\mathcal{E}(\hat{f}_{DNN})$ converges at a rate $\mathcal{O}\left(e^{-\beta n}\right)$ for some constant $\beta > 0$.

In all of our results, the excess risk bounds hold in the almost-sure sense and are non-asymptotic: they hold for any integer $n \geq n_0$ for some constant $n_0$ whose expression we give explicitly in terms of the problem's parameters.

### 1.3 Notations

**Function Spaces:**

Let $d \geq 1$ be an integer. For a closed subset $\mathcal{X} \subseteq \mathbb{R}^d$, a Borel measurable $\mathcal{Y} \subseteq \mathbb{R}$, and an integer $k \geq 0$ we will denote by

- $\mathcal{M}(\mathcal{X}, \mathcal{Y})$ the space of Borel measurable functions from $\mathcal{X}$ to $\mathcal{Y}$,

- $\mathcal{C}^k(\mathcal{X}, \mathcal{Y})$ the space of $\mathcal{Y}$-valued, $k$ times continuously differentiable functions on $\mathcal{X}$,

- $L^p(\mathcal{X}, \mathcal{Y}, \mu)$ the space of Borel measurable $\mathcal{Y}$-valued functions on $\mathcal{X}$ whose absolute $p$-th power is $\mu$-integrable, where $\mu$ is a measure on $\mathcal{X}$ and $p \in [1, \infty]$. Whenever $\mu$ is the Lebesgue measure, we will omit it from notation and simply write $L^p(\mathcal{X}, \mathcal{Y})$.

For any of these function spaces, we might drop the domain $\mathcal{X}$ and/or the co-domain $\mathcal{Y}$ from notation if context already makes it clear.

**Norms:**

For any positive integers $d, u, v$, real $0 < p < \infty$, $x = (x_1, \ldots, x_d)^T \in \mathbb{R}^d$, $A = (a_{i,j}) \in \mathbb{R}^{u \times v}$ and $f \in \mathcal{M}(\mathcal{X}, \mathbb{R})$, we will denote by

- respectively $|x|_p := (|x_1|^p + \ldots + |x_d|^p)^{1/p}$, $|x|_0 := |x_1|^0 + \ldots |x_d|^0$ (with the convention $0^0 := 0$) and $|x|_\infty := \max_{1 \leq i \leq d} |x_i|$, the $\ell_p$, $\ell_0$ and $\ell_\infty$ (quasi-)norm of $x$.

- $|A|_p := \left( \sum_{i,j} |a_{i,j}|^p \right)^{1/p}$ the $\ell_{p,p}$ norm of $A$,

- respectively $\|f\|_{\mathcal{C}^k(\mathcal{X})}$ and $\|f\|_{L^p(\mu)}$ the $\mathcal{C}^k(\mathcal{X}, \mathbb{R})$ norm and $L^p(\mathcal{X}, \mathbb{R}, \mu)$ norm of $f$, which are defined in the standard way.

**Other Symbols:**

We will also denote by

- $\mathbb{N} := \{1, 2, \ldots\}$ the set of all natural numbers, and $\mathbb{N}_0 := \{0\} \cup \mathbb{N}$,

- $\mathbb{1}_A$ the indicator function of a set $A$, which equals 1 on $A$ and 0 everywhere else,

- $\text{sign}(x) := \mathbb{1}_{(0,\infty)}(x) - \mathbb{1}_{(-\infty,0)}(x)$ the sign of a real number $x$. We will also denote by $\text{sign } f := \text{sign} \circ f$ the composition of a real-valued function $f$ with sign,

- $\mathbb{E}[Z]$ the expectation of a random variable $Z$. If $Z = f(X, Y)$, we may write $\mathbb{E}_X[Z]$ or $\mathbb{E}_Y[Z]$ to indicate with respect to which variables the expectation is taken, or equivalently $\mathbb{E}_\mu[Z]$ to indicate with respect to which distribution the expectation is taken,

- For two sequences of real numbers $(A_n)_{n \geq 1}$ and $(B_n)_{n \geq 1}$, we will write $A_n \lesssim B_n$ if $A_n \leq C B_n$ for some absolute constant $C > 0$, and $A_n = \mathcal{O}(B_n)$ if there exists $n_0 \in \mathbb{N}$ such that $|A_n| \lesssim |B_n|$ for all $n \geq n_0$.

## 2 Problem setting

Let $d \geq 2$ be an integer. We are given a sample of $n$ observations $(x_i, y_i) \in \mathcal{X} \times \mathcal{Y}$ where $\mathcal{X} \equiv [0,1]^d$ is the $d$-dimensional unit cube and $\mathcal{Y} \equiv \{-1, 1\}$ is the set of possible labels. Each sample is assumed to be i.i.d.

data points generated from a distribution $\rho$ on the probability space $(\Omega, \mathfrak{A}, \mathbb{P})$. We will call any measurable map $c : \mathcal{X} \to \mathcal{Y}$ a *classifier*, and for any such function $c$ we define its *misclassification risk* by

$$\mathcal{R}(c) := \mathbb{P}_{(X,Y) \sim \rho}(c(X) \neq Y) \tag{1}$$

For any function $f : \mathcal{X} \to \mathbb{R}$, we thus see that $\operatorname{sign} f$ is always a classifier, and we will call $\operatorname{sign} f$ the classifier *induced* by $f$. It is well known that the misclassification risk is minimized by the Bayes classifier $c^* := \operatorname{sign} \eta$ (Devroye et al., 2013), where

$$\eta(x) := \mathbb{E}_{(X,Y) \sim \rho}[Y \mid X = x]$$

is the so-called Bayes regression function.

We will denote by $\mathcal{R}^* := \mathcal{R}(c^*)$ the optimal risk. As $c^*$ depends on the unknown distribution $\rho$, it is a priori not possible to achieve the optimal risk $\mathcal{R}^*$, hence we instead aim to learn a classifier $\widehat{c}_n$ from the observations $(x_1, y_1), \dots, (x_n, y_n)$, such that the *excess risk* $\mathcal{R}(\widehat{c}_n) - \mathcal{R}^*$ converges to zero as fast as possible when $n$ goes to infinity.

## 2.1 Empirical Risk Minimization

The misclassification risk (1) being a function of $\rho$, it can't be explicitly computed and hence minimized. We instead minimize the following *Empirical Risk* with square surrogate loss:

$$\widehat{\mathcal{R}}_\ell(f) := \frac{1}{n} \sum_{i=1}^n \left(f(x_i) - y_i\right)^2 \tag{2}$$

Our choice of the square loss $\ell(f(x), y) := (f(x) - y)^2$ as a surrogate is motivated by at least three reasons :

- Empirical evidence suggests that square loss may perform just as well if not better than cross-entropy for classification tasks (Hui and Belkin, 2021). Our result thus provides some theoretical backing for this observation.

- (Hu et al., 2021) prove rates of convergence under the hard-margin condition for Neural Networks classifiers in the NTK regime learned with square loss. Our work shows that their results extend outside of the NTK regime, as they correctly conjectured.

- Most convergence rate results for kernel-based classifiers under margin conditions also consider the square loss as a surrogate (Steinwart and Scovel, 2005; Steinwart and Christmann, 2008). We thus have an analogous setting for DNNs and can meaningfully compare the two approaches.

To match what is often done in practice, we also introduce a penalty function $\mathcal{P} : \mathcal{H} \to \mathbb{R}_{\geq 0}$ and a regularization parameter $\lambda \geq 0$. This leads to the following $\lambda$-*Regularized Empirical Risk Minimization* ($\lambda$-ERM) problem :

$$\widehat{f}_\lambda := \underset{f \in \mathcal{H}}{\operatorname{argmin}} \left\{ \widehat{\mathcal{R}}_\ell(f) + \lambda \mathcal{P}(f) \right\}. \tag{3}$$

As stated earlier, we will set the hypothesis space $\mathcal{H}$ as a parametric family of functions, and the penalty $\mathcal{P}$ as the $\ell_p$ norm. We aim to give fast rates of convergence for the excess risk of $\operatorname{sign} \widehat{f}_\lambda$, the classifier induced by $\widehat{f}_\lambda$.

## 2.2 Hypothesis Spaces of Parametric Functions

### 2.2.1 Parametric Function Families

We start by defining the parametric families of functions we will be considering in this paper, and the associated terminology. Given integers $L, a_0, a_1, \dots, a_L \in \mathbb{N}$, we call *parameter vector* and denote by

$$\boldsymbol{\theta} := ((W_1, B_1), \dots, (W_L, B_L))$$

a tuple of matrix-vector pairs, where $W_l \in \mathbb{R}^{a_l \times a_{l-1}}$ and $B_l \in \mathbb{R}^{a_l}$ are respectively referred to as *weight matrices* and *bias vectors*.

We call $\mathbf{a} = (a_0, a_1, \ldots, a_L) \in \mathbb{N}^{L+1}$ an *architecture vector*, and given any such $\mathbf{a}$, we define the sets of all respectively bounded and unbounded *parametrizations* as:

$$\mathcal{P}_{\mathbf{a},R} := \bigtimes_{l=1}^{L} \left( [-R, R]^{a_l \times a_{l-1}} \times [-R, R]^{a_l} \right), \quad \mathcal{P}_{\mathbf{a},\infty} := \bigtimes_{l=1}^{L} \left( \mathbb{R}^{a_l \times a_{l-1}} \times \mathbb{R}^{a_l} \right) \tag{4}$$

where $R > 0$ is a fixed *parameter bound*. We then call a *realization mapping* any fixed map

$$\mathcal{F} : \mathcal{P}_{\mathbf{a},\infty} \to \mathcal{C}(\mathcal{X}, \mathbb{R}) \tag{5}$$

and from then we define

$$\mathcal{H}_{\mathcal{F},\mathbf{a},R} := \{ \mathcal{F}(\boldsymbol{\theta}) \mid \boldsymbol{\theta} \in \mathcal{P}_{\mathbf{a},R} \}$$

as the hypothesis space of functions *induced by $\mathcal{F}$*, parametrized by $\mathbf{a}$ and with parameters bounded by $R$.

A quantity of interest, after having defined $\mathcal{H}_{\mathcal{F},\mathbf{a},R}$ as we did, is the *sparsity* of $\mathcal{H}_{\mathcal{F},\mathbf{a},R}$. That is, the number of non-zero parameters needed to describe an arbitrary element of $\mathcal{H}_{\mathcal{F},\mathbf{a},R}$. We will denote that quantity — which implicitly depends on $\mathcal{F}$ — $P(\mathbf{a})$, and remark that we always have $P(\mathbf{a}) \leq \sum_{l=1}^{L} a_l(a_{l-1}+1)$. We also remark that any tuple $\boldsymbol{\theta} \in \mathcal{P}_{\mathbf{a},\infty}$ can naturally be identified, up to permutation, with a vector $\widetilde{\boldsymbol{\theta}} \in \mathbb{R}^{P(\mathbf{a})}$, hence the name *parameter vector*.

This rather general and seemingly arbitrary representation for parametric families of function is motivated by hypothesis spaces of Deep Neural Networks, to which it is particularly adapted. However, this representation can be used to represent essentially any kind of parametric family of real-valued functions one might use for binary classification in practice, such as:

- Linear classifiers, induced by maps of the form $x \mapsto W^T x + B$. In this case, the architecture vector is simply given by $\mathbf{a} = (d, 1)$ with corresponding parametrization $\mathcal{P}_{\mathbf{a},\infty} = \mathbb{R}^d \times \mathbb{R}$. The associated realization mapping is defined for all $\boldsymbol{\theta} \equiv (W, B) \in \mathcal{P}_{\mathbf{a},\infty}$ by

$$\mathcal{F}(\boldsymbol{\theta}) = \left( f : x \mapsto W^T x + B \right).$$

- Logistic regression, induced by maps of the form $x \mapsto 2(1 + \exp(W^T x + B))^{-1} - 1$. For this, the architecture vector and parametrization are again given by $\mathbf{a} = (d, 1)$ and $\mathcal{P}_{\mathbf{a},\infty} = \mathbb{R}^d \times \mathbb{R}$. The associated realization mapping is then defined for all $\boldsymbol{\theta} \equiv (W, B) \in \mathcal{P}_{\mathbf{a},\infty}$ by

$$\mathcal{F}(\boldsymbol{\theta}) = \left( f : x \mapsto 2(1 + \exp(W^T x + B))^{-1} - 1 \right).$$

- Kernel classifiers, induced by maps of the form $x \mapsto \sum_{i=1}^{n} \alpha_i K(x_i, x)$ where $K$ is a Mercer kernel defined on $\mathcal{X} \times \mathcal{X}$ and $x_1, \ldots, x_n \in \mathcal{X}$ are the training data points. The architecture and associated parametrization in this case are respectively $\mathbf{a} = (n, 1)$ and $\mathcal{P}_{\mathbf{a},\infty} = \mathbb{R}^n \times \mathbb{R}$. The associated realization mapping is then defined for all $\boldsymbol{\theta} \equiv (\alpha, B) \in \mathcal{P}_{\mathbf{a},\infty}$ by

$$\mathcal{F}(\boldsymbol{\theta}) = \left( f : x \mapsto \sum_{i=1}^{n} \alpha_i K(x_i, x) \right).$$

  Note that for this hypothesis space, the bias parameter is not used. Hence we have an effective number $P(\mathbf{a}) = n$ of parameters.

In all of the remaining text, we will slightly abuse notation and identify any $f \in \mathcal{H}_{\mathcal{F},\mathbf{a},R}$, which is defined on all of $\mathbb{R}^d$, with its restriction to the unit cube $\mathcal{X}$.

### 2.2.2 Clipping the function outputs

To study the generalization error of our hypothesis space $\mathcal{H}_{\mathcal{F},\mathbf{a},R}$, it is necessary to ensure that the functions within have uniformly bounded supremum norm, as the complexity may grow unboundedly otherwise. A simple way to guarantee this is the following: given a *clipping constant* $D > 0$, we compose all the functions in $\mathcal{H}_{\mathcal{F},\mathbf{a},R}$ with $\mathrm{clip}_D : \mathbb{R} \to \mathbb{R}$ defined by

$$\mathrm{clip}_D(x) = \begin{cases} D & \text{if } x \geq D \\ x & \text{if } -D \leq x \leq D \\ -D & \text{if } x \leq -D \end{cases} \tag{6}$$

Although the clipping operator ensures boundedness of outputs, one may worry about it negatively affecting the approximation power of the hypothesis space. The following lemma guarantees that as long as the clipping constant $D$ is chosen larger than $\|\eta\|_{L^\infty}$, the approximation error does not increase.

**Lemma 1.** *Let $f^* \in L^\infty(\mathcal{X}, \mathbb{R})$ and $D \geq \|f^*\|_{L^\infty(\mathcal{X}, \mathbb{R})}$. For any $f \in L^\infty(\mathcal{X}, \mathbb{R})$, we have*

$$\| \mathrm{clip}_D \circ f - f^* \|_{L^\infty(\mathcal{X},\mathbb{R})} \leq \|f - f^*\|_{L^\infty(\mathcal{X},\mathbb{R})}$$

*where $\mathrm{clip}_D$ is as defined in* (6).

*Proof.* By the assumption on $D$, we have $f^*(x) = \mathrm{clip}_D \circ f^*(x)$ for almost all $x \in \mathcal{X}$. Hence, by 1-Lipschitz continuity of $\mathrm{clip}_D$,

$$| \mathrm{clip}_D \circ f(x) - f^*(x)| = | \mathrm{clip}_D \circ f(x) - \mathrm{clip}_D \circ f^*(x)| \leq |f(x) - f^*(x)|$$

holds for almost all $x$, and the conclusion follows by definition of the essential supremum. $\square$

Likewise, it is immediate to see that for any $D > 0$, $f$ and $\mathrm{clip}_D \circ f$ induce the same classifier. Since the composition with $\mathrm{clip}_D$ does not affect the number of free parameters, and $\|\eta\|_{L^\infty(\mathcal{X})} \leq 1$, we will fix $D = 1$ and assume in the following that all functions we consider have been composed with $\mathrm{clip}_D$, without making it explicit in the notation, which is justified thanks to Lemma 1 above.

### 2.2.3 $\ell_p$ Regularization

Lastly, we fix $0 < p < \infty$ and regularize the objective (2) with an $\ell_p$ penalty term.

We thus define the regularized empirical risk as

$$\widehat{\mathcal{R}}_{\ell,\lambda}(\boldsymbol{\theta}) := \frac{1}{n} \sum_{i=1}^n (f(x_i; \boldsymbol{\theta}) - y_i)^2 + \lambda |\boldsymbol{\theta}|_p^p \tag{7}$$

where, for a parameter vector $\boldsymbol{\theta} = ((W_l, B_l))_{l=1}^L \in \mathcal{P}_{\mathbf{a},\infty}$,

$$|\boldsymbol{\theta}|_p^p := \sum_{l=1}^L |W_l|_p^p + |B_l|_p^p.$$

This penalty is very popular in practical applications. For $p = 2$, in which case it is often referred to as weight decay, it is known to help training and improve generalization (Krogh and Hertz, 1991). Similarly, $p = 1$ is a popular choice as it tends to promote sparse solutions, which are less expensive to store and more efficient to compute with (Candes et al., 2008). Although not as common, taking $0 < p < 1$ also has its merits, as it can be used as a differentiable approximation of the $\ell_0$ penalty, which induces very sparse models but is not compatible with standard gradient-based optimization algorithms (Louizos et al., 2018).

For fixed $R > 0$ and $\lambda \geq 0$, the $\lambda$-ERM problem (3) thus consists in finding $\widehat{\boldsymbol{\theta}}_\lambda$ satisfying

$$\widehat{\boldsymbol{\theta}}_\lambda \in \operatorname*{argmin}_{\boldsymbol{\theta} \in \mathcal{P}_{\mathbf{a},R}} \widehat{\mathcal{R}}_{\ell,\lambda}(\boldsymbol{\theta}) \tag{8}$$

Note that the objective (8) is, for most hypothesis spaces, highly non-convex. This implies that the set of minimizers is generally not reduced to a singleton. Therefore, we will only consider the minimum norm solutions throughout this paper, i.e. we only consider

$$\widehat{\boldsymbol{\theta}}_\lambda \in \operatorname{argmin} \left\{ |\boldsymbol{\theta}|_\infty, \text{ for } \boldsymbol{\theta} \in \operatorname*{argmin}_{\boldsymbol{\theta} \in \mathcal{P}_{\mathbf{a},R}} \widehat{\mathcal{R}}_{\ell,\lambda}(\boldsymbol{\theta}) \right\}. \tag{9}$$

### 2.3 Technical Assumptions

In this section, we present the technical assumptions under which we establish our main results.

**(A1)** The Bayes regression function $\eta : x \mapsto \mathbb{E}[Y \mid X = x]$ satisfies Tsybakov's *low-noise condition*: there exists a *noise exponent* $q > 0$ and a positive constant $C > 0$ such that

$$\mathbb{P}\left(|\eta(X)| \leq \delta\right) \leq C\delta^q \quad \text{for all } \delta > 0.$$

At times, we will also refer to Assumption **(A1)** as the "weak-margin' condition, using the two terms interchangeably and without particular preference. The so-called *hard-margin condition*, can be thought of as a "limit" of the low-noise condition when $q = \infty$:

**(A2)** The Bayes regression function $\eta : x \mapsto \mathbb{E}[Y \mid X = x]$ satisfies the *hard-margin condition*: there exists $\delta > 0$ such that

$$\mathbb{P}\left(|\eta(X)| > \delta\right) = 1.$$

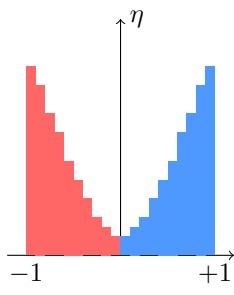

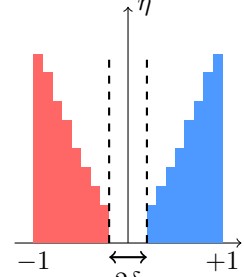

(a) Weak Margin Condition             (b) Hard Margin Condition

Figure 1: Visualization of margin conditions **(A1)** and **(A2)** through histograms of values taken by $\eta$.

Assumption **(A2)** was originally introduced in (Mammen and Tsybakov, 1999) as a characterization of classification problems for which the two classes are in some sense "separable", and has been repeatedly shown in the literature to lead to faster rates of convergence for various hypothesis classes.

Consider the regularized population risk $\mathcal{R}_{\ell,\lambda}$, which is given for all $\lambda \geq 0$ and $\boldsymbol{\theta} \in \mathcal{P}_{\mathbf{a},R}$ by

$$\mathcal{R}_{\ell,\lambda}(\boldsymbol{\theta}) := \mathbb{E}_{(x,y)\sim\rho}\left[(f(x;\boldsymbol{\theta}) - y)^2\right] + \lambda|\boldsymbol{\theta}|_p^p, \tag{10}$$

which for convenience, we also denote $\mathcal{R}_\ell$ whenever $\lambda = 0$. We likewise denote by $\widehat{\mathcal{R}}_{\ell,n}$ the unregularized version of the empirical risk $\widehat{\mathcal{R}}_{\ell,\lambda}$ defined in (7), where the dependence on $n$ is made explicit in the notation. Lastly, denote by

$$\operatorname{argmin}^* \widehat{\mathcal{R}}_{\ell,n} := \operatorname{argmin} \left\{ |\boldsymbol{\theta}|_\infty : \boldsymbol{\theta} \in \operatorname*{argmin}_{\boldsymbol{\theta} \in \mathcal{P}_{\mathbf{a},\infty}} \widehat{\mathcal{R}}_{\ell,n} \right\} \tag{11}$$

the set of minimum-norm minimizers of $\widehat{\mathcal{R}}_{\ell,n}$, taken as a function defined on the unrestricted parameter space $\mathcal{P}_{\mathbf{a},\infty}$ (4). We will assume the following:

**(A3)** The sets $\operatorname{argmin}_{\boldsymbol{\theta}\in\mathcal{P}_{\mathbf{a},\infty}} \mathcal{R}_\ell$ and $\operatorname{argmin}_{\boldsymbol{\theta}\in\mathcal{P}_{\mathbf{a},\infty}} \widehat{\mathcal{R}}_{\ell,n}$ are (almost surely) not empty, and there exists a constant $R_0 > 0$ such that almost surely over all possible i.i.d. draws $(x_i, y_i)_{i\geq 1}$ with distribution $\rho$, we have

$$\sup_{n\geq 1} \left\{ |\boldsymbol{\theta}|_\infty : \boldsymbol{\theta} \in \operatorname{argmin*} \widehat{\mathcal{R}}_{\ell,n} \right\} \leq R_0. \tag{12}$$

Besides the requirement that minimizers exist, which is standard and often implicitly assumed when studying empirical risk minimization, Assumption **(A3)** states that the minimum-norm solutions of the unregularized (i.e. $\lambda = 0$) ERM problem (9) almost surely do not run off to infinity as the sample size $n$ increases. Although it is intuitively expected that minimum-norm solutions do not diverge, in practice such an event could have a small, positive probability. Assumption **(A3)** thus requires $\rho$ to give zero measure to "pathological" datasets where such a thing happens. As the range of validity of this assumption is not immediately clear, we provide in Remark 1 below simple examples of settings under which it holds.

**Remark 1** (Examples of sufficient conditions for Assumption **(A3)**). *Assumption **(A3)** holds, for example, in each of the following settings.*

(a) ***Kernel classifier model with well-conditioned Gram matrix.*** *Assume $f(x;\theta) = \langle \theta, \tilde{\phi}(x) \rangle$ for some feature map $\tilde{\phi} : \mathcal{X} \to \mathbb{R}^m$, and that $\|\tilde{\phi}(X)\|_2 \leq B$ almost surely. Let $\tilde{\Phi}_n \in \mathbb{R}^{n\times m}$ be the design matrix with rows $\tilde{\phi}(x_i)^\top$, and assume that for some $c > 0$,*

$$\lambda_{\min}\left(\frac{1}{n}\tilde{\Phi}_n^\top \tilde{\Phi}_n\right) \geq c \quad \text{almost surely for all } n \geq 1.$$

*Then any minimum-$\ell_\infty$ minimizer $\hat{\theta}_n \in \operatorname{argmin*} \widehat{\mathcal{R}}_{\ell,n}$ satisfies*

$$|\hat{\theta}_n|_\infty \leq |\hat{\theta}_n|_2 \leq \left| \left(\frac{1}{n}\tilde{\Phi}_n^\top \tilde{\Phi}_n\right)^{-1} (\tilde{\Phi}_n^T y) \right|_2 \leq \frac{B}{c},$$

*so (12) holds with $R_0 = B/c$.*

(b) ***Geometric margin condition (linear case).*** *Consider $f(x;(w,b)) = \langle w, x \rangle + b$ and the augmented variables $\bar{x} := (x,1)$, $\bar{w} := (w,b)$. For a sample $(x_i, y_i)_{i=1}^n$, define its geometric margin as*

$$\gamma_n := \max_{\|\bar{w}\|_2 \leq 1} \min_{1\leq i\leq n} y_i \langle \bar{w}, \bar{x}_i \rangle.$$

*Assume $\inf_{n\geq 1} \gamma_n \geq \gamma > 0$ almost surely. Then for each $n$ we can pick $\bar{w}_n^\star$ attaining $\gamma_n$ and set $\theta_n^\dagger := \bar{w}_n^\star/\gamma_n$, so that $y_i f(x_i; \theta_n^\dagger) \geq 1$ for all $i \leq n$. With clipping at $D = 1$ this yields $\widehat{\mathcal{R}}_{\ell,n}(\theta_n^\dagger) = 0$, hence by minimality, $\sup\{|\theta|_\infty : \theta \in \operatorname{argmin*} \widehat{\mathcal{R}}_{\ell,n}\} \leq |\theta_n^\dagger|_\infty \leq |\theta_n^\dagger|_2 \leq 1/\gamma$, and (12) holds with $R_0 = 1/\gamma$.*

(c) ***Geometric margin condition (feature map case).*** *Consider $f(x;(\theta,b)) = \langle \theta, \phi(x) \rangle + b$ for some feature map $\phi : \mathcal{X} \to \mathbb{R}^m$, and define $\bar{\phi}(x) := (\phi(x), 1)$ and $\bar{\theta} := (\theta, b)$. For a sample $(x_i, y_i)_{i=1}^n$, define*

$$\gamma_n^{(\phi)} := \max_{\|\bar{\theta}\|_2 \leq 1} \min_{1\leq i\leq n} y_i \langle \bar{\theta}, \bar{\phi}(x_i) \rangle.$$

*If $\inf_{n\geq 1} \gamma_n^{(\phi)} \geq \gamma > 0$ almost surely, then scaling an optimizer as above yields an interpolating parameter vector with $|\bar{\theta}|_2 \leq 1/\gamma$. Hence we once again get $|\bar{\theta}|_\infty \leq 1/\gamma$, and (12) holds with $R_0 = 1/\gamma$.*

Under Assumption **(A3)** we can uniformly bound the norms of $\lambda$-ERM solutions, as well as that of approximation error minimizers:

**Lemma 2.** *Assume that **(A3)** holds with upper bound $R_0 > 0$. The following is true:*

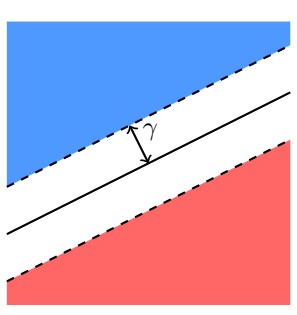
(a) Linear case in input space.

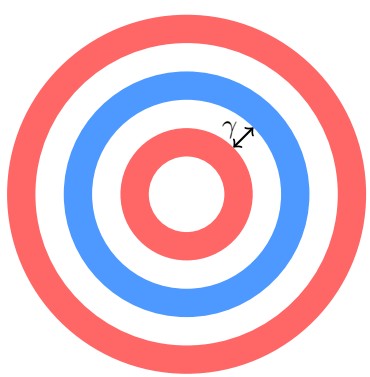
(b) Nonlinear case in feature space.

Figure 2: Illustration of geometric margin conditions corresponding to Remarks (b) and (c).

i. *Almost surely for all $n \geq 1, \lambda > 0$ the sets $\operatorname{argmin}_{\boldsymbol{\theta} \in \mathcal{P}_{\mathbf{a},\infty}} \widehat{\mathcal{R}}_{\ell,\lambda}$ are not empty, and we have the inequality*

$$\sup_{\lambda \geq 0, n \geq 1} \left\{ |\boldsymbol{\theta}|_\infty : \boldsymbol{\theta} \in \operatorname{argmin}^* \widehat{\mathcal{R}}_{\ell,\lambda} \right\} \leq R_0 P(\mathbf{a})^{1/p},$$

*where $P(\mathbf{a})$ denotes the number of parameters in the architecture $\mathcal{P}_{\mathbf{a},\infty}$ and $\operatorname{argmin}^* \widehat{\mathcal{R}}_{\ell,\lambda}$ is defined as in (11).*

ii. *There exists $\boldsymbol{\theta}^* \in \mathcal{P}_{\mathbf{a},\infty}$ such that*

$$|\boldsymbol{\theta}^*|_\infty \leq R_0 \ \text{ and } \ \boldsymbol{\theta}^* \in \operatorname*{argmin}_{\boldsymbol{\theta} \in \mathcal{P}_{\mathbf{a},\infty}} \|\mathcal{F}(\boldsymbol{\theta}) - \eta\|_{L^2(\rho_X)}.$$

Lemma 2, whose proof we defer to the last section, guarantees that if the parameter bound $R$ is chosen equal to $R_0 P(\mathbf{a})^{1/p}$ or greater, the hypothesis space $\mathcal{H}_{\mathcal{F},\mathbf{a},R}$ will both contain global solutions for the $\lambda$-ERM problem (9), and global minimizers of the $L^2(\rho_X)$ approximation error.

We now briefly introduce some necessary terminology.

**Definition 1** (Analytic functions). *A real-valued function $f$ defined on an open set $U \subseteq \mathbb{R}^k$ is said to be real analytic on $U$ if for all $u \in U$, there exists an open ball $B$ containing $u$ and real coefficients $(c_\alpha)_{\alpha \in \mathbb{N}_0^k}$ such that $f(x) = \sum_{\alpha \in \mathbb{N}_0^k} c_\alpha (x-u)^\alpha$ for all $x \in B$. An $\mathbb{R}^k$-valued function $f = (f_1, \ldots, f_k)^T$ is said to be analytic if each of its components is analytic.*

**Definition 2** (Semialgebraic sets and functions). *A set $S \subseteq \mathbb{R}^k$ is said to be semialgebraic (Bochnak et al., 1998) if it can be written in the form*

$$S = \bigcup_{i=1}^s \bigcap_{j=1}^t \left\{ x \in \mathbb{R}^k : p_{ij}(x) = 0, q_{ij}(x) > 0 \right\},$$

*where $s, t \in \mathbb{N}$ and $p_{ij}, q_{ij}$ are polynomial functions for all $1 \leq i \leq s, 1 \leq j \leq t$.*
*We call a function $f : \mathbb{R}^k \to \mathbb{R}$ semialgebraic if its graph $\{(x, f(x)) : x \in \mathbb{R}^k\}$ is semialgebraic.*

**Definition 3** (Subanalytic sets and functions). *A set $S \subseteq \mathbb{R}^k$ is said to be subanalytic (Shiota, 1997) if each point of $\mathbb{R}^k$ has a neighborhood $U$ such that $S \cap U$ is a finite union of sets of the form $\operatorname{Im} f_1 \setminus \operatorname{Im} f_2$, where $f_1$ and $f_2$ are analytic $\mathbb{R}^k$ valued proper maps defined on analytic manifolds.*
*We call a function $f : \mathbb{R}^k \to \mathbb{R}$ subanalytic if its graph $\{(x, f(x)) : x \in \mathbb{R}^k\}$ is subanalytic.*

The concept of semialgebraic function is relatively straightforward: it refers to functions whose graph can be described using finite unions and intersections of polynomial equalities and inequalities. Subanalyticity

extends this concept in some sense by allowing not just polynomial, but analytic equalities and inequalities, allowing for a much richer range of function graphs (Shiota, 1997; Bochnak et al., 1998).

As it turns out, most choices of loss functions, hypothesis spaces, and regularizations will lead to subanalytic, if not semialgebraic, population and empirical risk. For DNN hypothesis spaces, (Zeng et al., 2019) give sufficient conditions on the choice of loss, activation function and regularization to ensure subanalyticity of the resulting risk. Those conditions are verified for most practically used models, including, e.g., ReLU, sigmoid, GELU networks with $\ell_p$ regularization and square, hinge or cross-entropy loss (Zeng et al., 2019, Appendix B).

The last notion we introduce is that of a *limiting subdifferential*:

**Definition 4** (limiting subdifferential). *Given a function $f : \mathbb{R}^k \to \mathbb{R}$ and $x \in \mathbb{R}^k$, we call* Fréchet *subdifferential of $f$ at $x$ and denote $\hat{\partial} f(x)$ the set of all vectors $v \in \mathbb{R}^k$ which satisfy the following:*

$$\liminf_{\substack{y \neq x \\ y \to x}} \frac{f(y) - f(x) - v^T(y - x)}{|y - x|_2} \geq 0.$$

*The limiting subdifferential of $f$ at $x \in \mathbb{R}^k$, which we denote $\partial f(x)$ is then defined as (Rockafellar and Wets, 1998):*

$$\partial f(x) := \left\{ v \in \mathbb{R}^k : \exists (x_i)_{i \in \mathbb{N}} \to x, f(x_i) \to f(x), v_i \in \hat{\partial} f(x_i) \to v \right\}.$$

The limiting subdifferential $\partial f$ generalizes the concept of subdifferential to functions with even less regularity. Just like the former, it necessarily contains zero at any extremum of $f$, and coincides with its gradient $\nabla f$ whenever $f$ is differentiable (Rockafellar and Wets, 1998).

We can now introduce the following assumption:

**(A4)** For any $\lambda \geq 0$, the regularized population risk $\mathcal{R}_{\ell,\lambda} : \mathbb{R}^{P(\mathbf{a})} \to \mathbb{R}$ is subanalytic, and for any $\boldsymbol{\theta} \in \mathbb{R}^{P(\mathbf{a})}$, $\partial \mathcal{R}_{\ell,\lambda}(\boldsymbol{\theta})$ is not empty.
Furthermore, for any $R > 0$, the restriction $\mathcal{R}_{\ell,\lambda} : [-R, R]^{P(\mathbf{a})} \to \mathbb{R}$ is Lipschitz continuous, and we will denote by $\mathrm{Lip}_R(\mathcal{R}_{\ell,\lambda}) \equiv \mathrm{Lip}(\mathcal{R}_{\ell,\lambda})$ its Lipschitz constant.

In light of the preceding discussion, we observe that Assumption **(A4)** is very mild. Both subanalyticity and (limiting) subdifferentiability are satisfied in most practical setups, and local Lipschitz-ness is also guaranteed as long as, e.g., the parametric hypothesis space consists of compositions of locally Lipschitz mappings, which is often the case as well.

The need for Assumption **(A4)** is motivated by the following theorem, due to (Bolte et al., 2007):

**Theorem 1** (Adapted from Corollary 16 in (Bolte et al., 2007)). *If $f : \mathbb{R}^k \to \mathbb{R}$ is a lower semi-continuous, globally subanalytic function, and $f(x_0) = 0$, then there exist $c, \rho > 0$ and $0 < \kappa < 1$ such that for all $x \in \{x \in \mathbb{R}^k : 0 < |f(x)| < \rho\}$, we have:*

$$c|f(x)|^{\kappa} \leq |x^*|_2, \quad \text{for all } x^* \in \partial f(x) \tag{13}$$

Functions satisfying the conclusion of Theorem 1 are said to satisfy the Kurdyka-Lojasiewicz (KL) property, named after pioneering works of Lojasiewicz (Lojasiewicz, 1963) and Kurdyka (Kurdyka, 1998). This property is fundamental in non-convex optimization, as it is a key ingredient when proving convergence guarantees for first-order methods in general settings (Li and Pong, 2018). This property has also been used in recent works to obtain convergence rates for first-order optimization in Deep Learning (Xu et al., 2024; Zeng et al., 2019).

As a consequence of Assumption **(A4)**, we have the following Proposition, whose proof we provide in the last section:

**Proposition 1.** *Assume that Assumption (A4) holds. Fix $R := R_0 P(\mathbf{a})^{1/p} > 0$, where $R_0$ satisfies (12), and let $\Lambda > 0$ be arbitrary. There exist constants $K, \rho > 0$, and $r > 1$, such that for all $0 \leq \lambda \leq \Lambda$,*

$0 < t < \rho/(2\,\mathrm{Lip}_R(\mathcal{R}_{\ell,\lambda}))$, and $\boldsymbol{\theta}_\lambda \in \mathrm{argmin}_{\boldsymbol{\theta} \in \mathcal{P}_{\mathbf{a},R}}\, \mathcal{R}_{\ell,\lambda}$

$$\inf_{\boldsymbol{\theta} \in \mathcal{P}_{\mathbf{a},R}:\mathrm{dist}(\boldsymbol{\theta},\mathrm{argmin}\,\mathcal{R}_{\ell,\lambda}) \geq t} \mathcal{R}_{\ell,\lambda}(\boldsymbol{\theta}) - \mathcal{R}_{\ell,\lambda}(\boldsymbol{\theta}_\lambda) \geq Kt^r, \tag{14}$$

where $\mathrm{dist}(a, A)$ denotes the $\ell_\infty$ distance between a vector $a \in \mathbb{R}^k$ and a set $A \subseteq \mathbb{R}^k$.

We call the number $r > 1$ in (14) the *separation exponent*. It quantifies how sharply the population objective $\mathcal{R}_{\ell,\lambda}$ grows as we move away from the set of minimizers: for small $t$, a larger $r$ corresponds to a flatter landscape near $\mathrm{argmin}\,\mathcal{R}_{\ell,\lambda}$, so parameters at a small positive distance from the minimizers can still have nearly optimal risk, while a smaller $r$ forces the objective to increase more quickly away from the minimizer set. Conditions similar to (14) are often assumed in empirical process theory under the name *well-separation* and are used to prove consistency of $M$-estimators (Van der Vaart, 2000; Sen, 2018). In that context, the polynomial lower bound $Kt^r$ is typically replaced by an unspecified function $\psi(t)$ that is only assumed to be positive for small $t > 0$. Proposition 1 shows that, under Assumption **(A4)**, such a well-separation condition holds with a lower bound in closed-form, which will be a key input in the error analysis that follows.

## 3 Main results

### 3.1 A general upper bound for the excess risk of parametric classifiers

Before stating our main results, we introduce a few useful definitions. The first being that of an $\varepsilon$-*covering* :

**Definition 5** ($\varepsilon$-cover)**.** *Let $\varepsilon > 0$ and $\mathcal{G} \subseteq L^\infty(\mathcal{X}, \mathbb{R})$ be a family of functions. Any finite collection of functions $g_1, \ldots, g_N \in L^\infty(\mathcal{X}, \mathbb{R})$ with the property that for any $g$ in $\mathcal{G}$ there is an index $j \equiv j(g)$ such that*

$$\|g - g_j\|_{L^\infty} \leq \varepsilon$$

*is called an $\varepsilon$-covering (or cover) of $\mathcal{G}$ with respect to $\|\cdot\|_{L^\infty}$.*

For a given $\varepsilon$, we can think of the cardinality of an $\varepsilon$-cover as a measure of complexity for the family $\mathcal{G}$. This motivates the definition of a *covering number* :

**Definition 6** ($\varepsilon$-covering number)**.** *Let $\varepsilon > 0$ and $\mathcal{G} \subseteq L^\infty(\mathcal{X}, \mathbb{R})$. We denote by $\mathbf{Cov}(\mathcal{G}, \|\cdot\|_{L^\infty}, \varepsilon)$ the size of the smallest $\varepsilon$-cover of $\mathcal{G}$ with respect to $\|\cdot\|_{L^\infty}$, with the convention $\mathbf{Cov}(\mathcal{G}, \|\cdot\|_{L^\infty}, \varepsilon) := \infty$ when no finite cover exists. $\mathbf{Cov}(\mathcal{G}, \|\cdot\|_{L^\infty}, \varepsilon)$ will be called an $\varepsilon$-covering number of $\mathcal{G}$ with respect to $\|\cdot\|_{L^\infty}$.*

*For $\mathcal{G} = \mathcal{H}_{\mathcal{F},\mathbf{a},R}$, we will abbreviate and denote*

$$\mathbf{Cov}(\mathcal{H}_{\mathcal{F},\mathbf{a},R}, \|\cdot\|_{L^\infty}, \varepsilon) =: \mathbf{Cov}_\infty(\mathcal{H}, \varepsilon).$$

The above two quantities, whose definitions are adapted from (Györfi et al., 2002), are ubiquitous in the learning theory literature, as they give a lot of information on the statistical properties of our estimators.

Another related measure of complexity for parametric hypothesis spaces is the Lipschitz constant of the realization map (5):

**Definition 7.** *Given a parametrization $\mathcal{P}_{\mathbf{a},R}$, recall the definition of the realization mapping $\mathcal{F} : \mathcal{P}_{\mathbf{a},R} \to \mathcal{C}(\mathcal{X}, \mathbb{R})$ (5). We will denote by*

$$\mathrm{Lip}_R(\mathcal{F}) := \sup_{\substack{\boldsymbol{\theta},\boldsymbol{\theta}' \in \mathcal{P}_{\mathbf{a},R} \\ \boldsymbol{\theta} \neq \boldsymbol{\theta}'}} \frac{\|\mathcal{F}(\boldsymbol{\theta}) - \mathcal{F}(\boldsymbol{\theta}')\|_{L^\infty(\mathcal{X})}}{|\boldsymbol{\theta} - \boldsymbol{\theta}'|_\infty}$$

*its Lipschitz constant.*

Intuitively, the Lipschitz constant of the realization map estimates the complexity of the induced hypothesis space in the sense that it controls how different two realizations can be given that their parametrizations are close. For this reason, the problem of estimating a Neural Network's Lipschitz constant has garnered a lot of interest over recent years (Fazlyab et al., 2019; Virmaux and Scaman, 2018).

We are now ready to state our main result, which gives an upper bound on the excess risk of parametric classifiers under our setting.

**Theorem 2.** *Assume that Assumptions (A3) and (A4) hold. Fix an architecture* **a** *with parameter bound* $R \equiv R_0 P(\mathbf{a})^{1/p}$, *where* $R_0 > 0$ *satisfies* (12), *and denote*

$$\varepsilon_{approx} := \inf_{f \in \mathcal{H}_{\mathcal{F},\mathbf{a},R}} \|f - \eta\|_{L^2(\rho_X)}$$

*the approximation error of* $\mathcal{H}_{\mathcal{F},\mathbf{a},R}$. *We have the following excess risk bounds:*

- *If the low-noise condition (A1) holds, then for all* $\delta > \varepsilon_{approx}$ *and* $0 < \nu < \delta$, *any minimum-norm solution* $\widehat{\boldsymbol{\theta}}_\lambda$ *of the* $\lambda$-*ERM problem* (9) *with* $0 \le \lambda < (\delta - \varepsilon_{approx})^2 (P(\mathbf{a})2^p R^p)^{-1}$ *satisfies for all* $n \ge 1$:

$$
\begin{aligned}
\mathcal{R}(\operatorname{sign} f(\cdot;\widehat{\boldsymbol{\theta}}_\lambda)) - \mathcal{R}^* \le{}& \varepsilon_{approx} + \sqrt{\lambda P(\mathbf{a})2^p R^p} + C\delta^q \\
&+ (\delta - \nu)^{-2} \left( \varepsilon_{approx} + \sqrt{\lambda P(\mathbf{a})2^p R^p} \right)^2 \\
&+ 4\, \mathbf{Cov}_\infty \left( \mathcal{H}, \frac{K\nu^r}{24\operatorname{Lip}_{2R}(\mathcal{F})^{1+r}} \right) \exp \left( \frac{-nK^2\nu^{2r}}{288\operatorname{Lip}_{2R}(\mathcal{F})^{2r}} \right)
\end{aligned}
\tag{15}
$$

- *If the hard-margin condition (A2) holds with margin* $\delta > 0$, *and* $\varepsilon_{approx} < \delta$, *then for all* $0 < \nu < \delta$, *any minimum-norm solution* $\widehat{\boldsymbol{\theta}}_\lambda$ *of the* $\lambda$-*ERM problem* (9) *with* $0 \le \lambda < (\delta - \varepsilon_{approx})^2 (P(\mathbf{a})2^p R^p)^{-1}$ *satisfies for all* $n \ge 1$:

$$
\begin{aligned}
\mathcal{R}(\operatorname{sign} f(\cdot;\widehat{\boldsymbol{\theta}}_\lambda)) - \mathcal{R}^* \le{}& \varepsilon_{approx} + \sqrt{\lambda P(\mathbf{a})2^p R^p} \\
&+ (\delta - \nu)^{-2} \left( \varepsilon_{approx} + \sqrt{\lambda P(\mathbf{a})2^p R^p} \right)^2 \\
&+ 4\, \mathbf{Cov}_\infty \left( \mathcal{H}, \frac{K\nu^r}{24\operatorname{Lip}_{2R}(\mathcal{F})^{1+r}} \right) \exp \left( \frac{-nK^2\nu^{2r}}{288\operatorname{Lip}_{2R}(\mathcal{F})^{2r}} \right)
\end{aligned}
\tag{16}
$$

The bounds in Theorem 2 may appear quite different from classical oracle inequalities encountered in the literature (Steinwart and Christmann, 2008; Koltchinskii, 2011; Schmidt-Hieber, 2020). Indeed, those bounds are typically obtained by bounding the error by the sum of an approximation error term and the supremum of some empirical process, which can then further be controlled using tools from empirical process theory and complexity bounds. We take a different path: instead of bounding the second summand by the supremum of an empirical process, we use an inequality (Lemma 6) which, combined with other results, allows us to control this summand by the tail probability of a certain random variable.

At first glance, it is not immediately evident that Theorem 2 can enable improved rates of convergence for ERM-based binary classification. Indeed, although the non-exponential summands in (15) and (16) can often be effectively controlled when the regression function $\eta$ belongs to an appropriate function space, the exponential term presents a challenge. In some cases, this term may fail to vanish as the architecture size **a** grows with the sample size $n$. In the next sections, however, we demonstrate that for DNN hypothesis spaces, Theorem 2 can indeed yield fast rates of convergence.

## 3.2 Complexity measure bounds for neural networks hypothesis spaces

We now focus our attention on the case where the hypothesis space $\mathcal{H}_{\mathcal{F},\mathbf{a},R}$ consists of Fully Connected Neural Networks (FCNNs). That is, we let $\sigma : \mathbb{R} \to \mathbb{R}$ be any activation function, and for a given architecture $\mathbf{a} = (a_0, \ldots, a_L) \in \mathbb{N}^{L+1}$, and parameter vector $\boldsymbol{\theta} = ((W_1, B_1), \ldots, (W_L, B_L)) \in \mathcal{P}_{\mathbf{a},\infty}$, we define the affine maps

$$T_\ell : \mathbb{R}^{a_{\ell-1}} \to \mathbb{R}^{a_\ell}, \quad x \mapsto W_\ell x + B_\ell, \quad (1 \le \ell \le L).$$

We then define for all $\boldsymbol{\theta} \in \mathcal{P}_{\mathbf{a},\infty}$ the realization mapping as

$$\mathcal{F}_{NN}(\boldsymbol{\theta}) := \left[ x \mapsto T_L \circ \sigma \circ T_{L-1} \circ \cdots \circ \sigma \circ T_1 \right], \tag{17}$$

where $\sigma$ is understood component-wise when applied to vectors.

As is common, we respectively refer to $|\mathbf{a}|_\infty \equiv W$ and $L$ as the *width* and *depth* of the neural network. For notational convenience, we will denote by $\mathcal{NN}(\mathbf{a}, W, L, R)$ the hypothesis space $\mathcal{H}_{\mathcal{F}_{NN}, \mathbf{a}, R}$ induced by $\mathcal{F}_{NN}$. As the width and depth of a Neural Network architecture play a crucial role in bounding its complexity, it is advisable to make them clearly visible in the notation.

In order to apply the estimate given by Theorem 2, we need a clear control of the complexity measures $\mathrm{Lip}_R(\mathcal{F}_{NN})$ and $\mathbf{Cov}_\infty(\mathcal{F}_{NN}, \varepsilon)$ in terms of the network's parameters. To this end, we first introduce the following bound on $\mathrm{Lip}_R(\mathcal{F}_{NN})$, which is a generalization of a similar results for ReLU networks which was shown in (Berner et al., 2020):

**Lemma 3.** *Assume that $\sigma : \mathbb{R} \to \mathbb{R}$ is globally Lipschitz with constant*

$$L_\sigma := \sup_{x \neq y} \frac{|\sigma(x) - \sigma(y)|}{|x - y|} < \infty,$$

*and set $C_\sigma := \max\{1, L_\sigma + |\sigma(0)|\}$. Then, for any architecture vector $\mathbf{a}$ with depth $L \in \mathbb{N}$ and width $W \in \mathbb{N}$, and any parameter bound $R \geq 1$, we have the upper bound on $\mathrm{Lip}_R(\mathcal{F}_{NN})$*

$$\mathrm{Lip}_R(\mathcal{F}_{NN}) \leq 2 L^2 (C_\sigma R)^{L-1} W^L.$$

Lemma 3, whose proof we give in the last section, shows that the Lipschitz constant of $\mathcal{F}_{NN}$ grows *exponentially* with depth. As one could expect, the covering number behaves similarly:

**Lemma 4.** *Let $\varepsilon > 0$. For any DNN architecture $\mathbf{a}$ with depth $L \in \mathbb{N}$ and width $W \in \mathbb{N}$, and any parameter bound $R > 0$, we have the upper bound*

$$\mathbf{Cov}_\infty(\mathcal{NN}, \varepsilon) \leq \left(1 + \frac{2R\,\mathrm{Lip}_R(\mathcal{F}_{NN})}{\varepsilon}\right)^{P(\mathbf{a})}$$

*Proof of Lemma 4.* Let $\boldsymbol{\theta}, \boldsymbol{\theta}' \in \mathcal{P}_{\mathbf{a}, R}$. Because of the inequality

$$\|f(\cdot; \boldsymbol{\theta}) - f(\cdot; \boldsymbol{\theta}')\|_{L^\infty(\mathcal{X})} \leq \mathrm{Lip}_R(\mathcal{F}_{NN})|\boldsymbol{\theta} - \boldsymbol{\theta}'|_\infty,$$

we get that $\mathbf{Cov}_\infty(\mathcal{NN}, \varepsilon)$ is bounded by the number of $\ell_\infty$ balls of radius $\varepsilon/\mathrm{Lip}(\mathcal{F}_{NN})$ needed to cover the hypercube $[-R, R]^{P(\mathbf{a})}$. It is straightforward to check that the collection of such balls centered at the points

$$-R\vec{\mathbf{1}} + \varepsilon\vec{\mathbf{k}}, \quad \text{where } \vec{\mathbf{k}} = [k_1, k_2, \ldots k_{P(\mathbf{a})}]^T, \text{ and } k_i \in \left\{0, 1, \ldots, \left\lceil \frac{2R\,\mathrm{Lip}_R(\mathcal{F}_{NN})}{\varepsilon} \right\rceil\right\},$$

where $\vec{\mathbf{1}}$ is the vector whose entries are all ones, covers $[-R, R]^{P(\mathbf{a})}$ and has $\lceil 2R\,\mathrm{Lip}_R(\mathcal{F}_{NN})/\varepsilon \rceil^{P(\mathbf{a})}$ elements. The proof is thus complete. $\square$

### 3.3 Super fast rates with distribution-adapted smoothness: ReLU activation and beyond

Combining our generalization error bound, with the complexity measure bounds from the last section, we will first derive convergence rates for ReLU neural networks. We will then leverage a result of (Zhang et al., 2024a) to extend these rates to a broad class of activation functions, which we will detail below.

#### 3.3.1 Upper bounds for ReLU networks

In light of the estimates from the previous section, we see that the only way for Theorem 2 to yield any useful results is for the approximation error to decay faster than complexity increases with respect to the network's size. A rich literature on DNN approximation theory has shown that for target functions in suitable smoothness spaces, such as Sobolev, Hölder, Besov or Korobov spaces, fast rates of approximation were possible for neural networks with ReLU activation (Yarotsky, 2017; Suzuki, 2019; Petersen and Voigtlaender, 2018; Mao and Zhou, 2022). In particular, we have the following result due to (Lu et al., 2021), which provides exact approximation bounds of $s$ times continuously differentiable functions by Deep ReLU FCNNs:

**Theorem 3** (Theorem 1.1 from (Lu et al., 2021)). *Let $s \in \mathbb{N}$ and $h \in \mathcal{C}^s(\mathcal{X})$. For any $W_0, L_0 \in \mathbb{N}$ there exists a neural network $f(\cdot; \boldsymbol{\theta}) \in \mathcal{NN}(\mathbf{a}, W, L, R)$ with width $W(\mathbf{a}) = C_1(W_0 + 2) \log_2(8W_0)$ and depth $L(\mathbf{a}) = C_2(L_0 + 2) \log_2(4L_0) + 2d$ such that*

$$\|f(\cdot; \boldsymbol{\theta}) - h\|_{L^\infty(\mathcal{X})} \le C_3 \|h\|_{\mathcal{C}^s(\mathcal{X})} W_0^{-2s/d} L_0^{-2s/d},$$

*where $C_1 = 17s^{d+1}3^d d$, $C_2 = 18s^2$ and $C_3 = 85(s+1)^d 8^s$.*

We now introduce the following Assumption **(A5)**, which specializes Theorem 3 to the $L^2(\rho_X)$ norm, and incorporates the interaction between the regression function $\eta$ and the marginal data distribution:

**(A5)** Let $\eta$ be the Bayes regression function. There exist a natural number $L_0 \in \mathbb{N}$, and a smoothness parameter $s > 0$, such that for any $W_0 \in \mathbb{N}_{\ge 2}$, one can find a neural network $f(\cdot; \boldsymbol{\theta}) \in \mathcal{NN}(\mathbf{a}, W, L, \infty)$ with width $W(\mathbf{a}) = C_1 W_0 \log_2(W_0)$ and depth $L(\mathbf{a}) = C_2 L_0 \log_2(L_0) + 2d$ such that

$$\|f(\cdot; \boldsymbol{\theta}) - \eta\|_{L^2(\rho_X)} \le C_3 W_0^{-2s/d},$$

where $C_1 = ds^d$, $C_3 = C_\eta \cdot d(3s)^d 8^s$ for some $C_\eta > 0$ depending on $\eta$ only, and the positive quantity $C_2 \equiv C_2(s)$ is such that $C_2(s)/s \to 0$ as $s \to \infty$.

Assumption **(A5)**, which we refer to as *distribution-adapted smoothness*, reframes Theorem 3 by shifting the metric to the $L^2(\rho_X)$ norm. While it shares some similarities with Theorem 3, the assumption places additional constraints on the depth parameter $L$, requiring it to grow at a slightly slower rate with respect to the smoothness parameter $s$, specifically at $o(s)$. This slower growth requirement addresses limitations identified in prior works showing that, under a uniform measure, achieving $L^2$ approximation error rates of $(WL)^{-2s/d}$ for non-linear $C^2$ functions requires a depth proportional to $s/d$ (Safran and Shamir, 2017; Voigtlaender and Petersen, 2019). Assumption **(A5)** accounts for this by considering not only the regularity of $\eta$, but also the interaction between $\eta$ and the marginal data distribution $\rho_X$. This interaction allows for slightly faster approximation rates when $\rho_X$ is "adapted" to $\eta$. As this assumption is highly non-standard, we provide in Section 6.5 four non-trivial examples of conditions on $(\rho_X, \eta)$ which can make **(A5)** hold. A full characterization of the necessary and sufficient conditions seems challenging, and we leave it as an open problem for future research.

This approximation error bound directly quantifies the width and depth required to achieve a given error level, leading to the following excess risk bounds for deep FCNN classifiers:

**Theorem 4.** *Let $\sigma$ be the ReLU activation. Assume that Assumptions **(A3)**, **(A4)**, and **(A5)** hold, and let $\alpha > 0$ be a desired rate of convergence. For any $n \in \mathbb{N}$, there exists a FCNN architecture $\mathbf{a}_n$ with width $W_n$, depth $L_n$ and parameter bound $R_n$ satisfying*

$$\begin{cases} W_n &= \tilde{\mathcal{O}}(n^{\alpha d/2rs}) \\ L_n &= C_2 L_0 \log_2 L_0 + 2d \\ R_n &= R_0 \left[ L_n(W_n^2 + W_n) \right]^{1/p} = \tilde{\mathcal{O}}(n^{\alpha d/ps}), \end{cases}$$

*where $C_2$ and $L_0$ are given in **(A5)**, $R_0$ is given in **(A3)**, and $\tilde{\mathcal{O}}$ hides polylogarithmic factors, such that the following holds:*

- *If the low-noise condition **(A1)** holds, and $\alpha < \left(1 + \frac{C_2 B_1 + B_2}{s}\right)^{-1}$, then any minimum-norm solution $\widehat{\boldsymbol{\theta}}_\lambda$ of the $\lambda$-ERM problem (9) with $\lambda = \tilde{\mathcal{O}}\left(n^{-\frac{2\alpha(s+d)}{rs}}\right)$, satisfies the excess risk bound:*

$$\mathcal{R}\left(\text{sign } f(\cdot; \widehat{\boldsymbol{\theta}}_\lambda)\right) - \mathcal{R}^* \lesssim \max\left\{ n^{-\frac{\alpha}{r}}, n^{-\frac{\alpha q}{2r}} \right\}, \tag{18}$$

  *where $q > 0$ is the noise exponent in Assumption **(A1)**, $r > 1$ is the "separation exponent" in (14), $B_1 = rd L_0 \log_2(L_0)(2 + 2p)/p$, and $B_2 = 2rd \left[ d(2 + 2p) - 1 \right]/p$.*

- *If the hard-margin condition* **(A2)** *holds with margin* $\delta > 0$, $\alpha < \frac{s}{C_2 B_1 + B_2}$, *and* $n > (\delta/2)^{-\alpha/r}$, *then any minimum-norm solution* $\widehat{\boldsymbol{\theta}}_\lambda$ *of the* $\lambda$-*ERM problem* (9) *with* $\lambda = \tilde{\mathcal{O}}\left(n^{-\frac{2\alpha(s+d)}{rs}}\right)$ *satisfies the excess risk bound:*

$$\mathcal{R}\left(\operatorname{sign} f(\cdot; \widehat{\boldsymbol{\theta}}_\lambda)\right) - \mathcal{R}^* \lesssim n^{-\frac{\alpha}{r}}, \tag{19}$$

  *where* $r > 1$ *is the "separation exponent" in* (14), $B_1 = rdL_0 \log_2(L_0)(2+2p)/p$, *and* $B_2 = 2rd\big[d(2+2p)-1\big]/p$.

As the smoothness parameter $s$ increases to infinity, we get, in the limit $\alpha \to \left(1 + \frac{C_2 B_1 + B_2}{s}\right)^{-1}$, a convergence rate of $\mathcal{O}\left(n^{-\frac{\min\{1, q/2\}}{r}}\right)$ under the low-noise condition **(A1)**. Since $r > 1$, the bound we get is thus slightly worse than the minimax optimal "fast-rate" of $\mathcal{O}\left(n^{-1}\right)$ which typically holds under Assumption **(A1)** when $q \to \infty$ (Kim et al., 2021; Audibert and Tsybakov, 2007).

On the other hand, under the hard-margin Assumption **(A2)**, we find that the exponent $\alpha/r$ grows unboundedly as the approximation speed $s$ goes to $\infty$. Theorem 4 thus shows how deep FCNNs can leverage the hard-margin condition **(A2)** together with higher regularity to achieve potentially arbitrarily fast rates of convergence for the excess risk. A result which, to the best of our knowledge, is the first of its kind for this hypothesis space.

### 3.3.2   Extension to diverse activation functions

We will now show how to transfer Theorem 4 from ReLU to other activation functions. To that end, we define the class $\mathscr{A}$ of admissible activations, as originally introduced in (Zhang et al., 2024a).

**Definition 8** (Activation class $\mathscr{A}$)**.** *Recall that* $\sigma : \mathbb{R} \to \mathbb{R}$ *denotes an activation function defined on the real line. We define* $\mathscr{A} := \mathscr{A}_1 \cup \mathscr{A}_2 \cup \mathscr{A}_3$, *where:*

- *We say* $\sigma \in \mathscr{A}_1$ *if there exist an open interval* $I = (a_0, b_0)$ *and a point* $x_0 \in I$ *such that, for some integer* $k \geq 0$, $\sigma \in C^k(I)$, $\sigma \notin C^{k+1}(I)$, *and the* $k$-*th derivative has a jump at* $x_0$ *in the sense that* $\sigma^{(k)}(x_0^-)$ *and* $\sigma^{(k)}(x_0^+)$ *exist and* $\sigma^{(k)}(x_0^-) \neq \sigma^{(k)}(x_0^+)$. *Examples include ReLU and LeakyReLU (with* $k = 0$*), and* $(\operatorname{ReLU})^m$ *(with* $k = m - 1$*).*

- *We say* $\sigma \in \mathscr{A}_2$ *if there exist* $b_0, b_1 \in \mathbb{R}$ *and a bounded function* $h : \mathbb{R} \to \mathbb{R}$ *with finite limits* $\lim_{x \to -\infty} h(x)$ *and* $\lim_{x \to \infty} h(x)$ *that are not equal, such that* $\sigma(x) = (x + b_0)h(x) + b_1$ *for all* $x \in \mathbb{R}$. *Examples include Softplus, ELU/CELU/SELU, GELU, SiLU/Swish, Mish.*

- *We say* $\sigma \in \mathscr{A}_3$ *if* $\sigma$ *is bounded, twice differentiable, has finite unequal limits at* $\pm\infty$, *and satisfies* $\sigma''(x_0) \neq 0$ *for some* $x_0 \in \mathbb{R}$. *Examples include sigmoid, tanh, arctan, and softsign.*

The class $\mathscr{A}$ contains essentially all standard activations used in practice, including their variants and combinations according to the properties discussed in (Zhang et al., 2024a). We can now state the result we will need to extend Theorem 4 to activations in $\mathscr{A}$:

**Theorem 5** (Adapted from (Zhang et al., 2024a), Theorem 1)**.** *Let* $\sigma \in \mathscr{A}$, *and respectively denote by* $\mathcal{F}_{NN}^\sigma$ *and* $\mathcal{F}_{NN}^{\operatorname{ReLU}}$ *the realization maps in* (17) *with activation* $\sigma$ *and ReLU. Fix* $\mathbf{a} = (a_0, \ldots, a_L) \in \mathbb{N}^{L+1}$ *and* $\boldsymbol{\theta} \in \mathcal{P}_{\mathbf{a}, \infty}$, *and set* $f = \mathcal{F}_{NN}^{\operatorname{ReLU}}(\boldsymbol{\theta})$. *Then for every* $\varepsilon > 0$ *there exist* $\mathbf{b} = (b_0, \ldots, b_{2L}) \in \mathbb{N}^{2L+1}$ *with* $|\mathbf{b}|_\infty \leq 3|\mathbf{a}|_\infty$ *and* $\boldsymbol{\vartheta} \in \mathcal{P}_{\mathbf{b}, \infty}$ *such that, with* $g = \mathcal{F}_{NN}^\sigma(\boldsymbol{\vartheta})$,

$$\|f - g\|_{L^\infty(\mathcal{X})} < \varepsilon.$$

Theorem 5 provides an activation-transfer principle: ReLU neural networks can be uniformly approximated on $\mathcal{X}$ by any network with activation $\sigma \in \mathscr{A}$, with only constant-factor changes in network size. Since this approximation result does not bound the weights of the approximating network, those weights may depend on $\varepsilon$. Under Assumption **(A3)** however, this dependence becomes irrelevant. This yields the following corollary, extending Theorem 4 to all activations in $\mathscr{A}$ which are globally Lipschitz:

**Corollary 1** (Extension of Theorem 4 to activations in $\mathscr{A}$). *Let $\sigma \in \mathscr{A}$ be globally Lipschitz. Assume that Assumptions (A3), (A4), and (A5) hold. Then Theorem 4 holds with the ReLU activation replaced by $\sigma$. In particular, the excess-risk bounds* (18)–(19) *hold with the same powers of $n$, up to changes in the implicit constants.*

### 3.3.3 Lower bounds

A natural question which arises from Theorem 4, is whether the obtained rates of convergence can be improved further, in the minimax sense. Although we've seen that the rates obtained under the low-noise condition **(A1)** are slightly suboptimal, one may still wonder whether those obtained under Assumption **(A2)** can be improved further. The following Theorem gives a negative answer to this question.

**Theorem 6.** *Let $s > 0$ be fixed, and $C_2 \equiv C_2(s), B_1, B_2 > 0$ and $r > 1$ all be as in Theorem 4. For $q, \delta > 0$, denote respectively by $\mathcal{P}_{s,q}^{(1)}$ and $\mathcal{P}_{s,\delta}^{(2)}$ the sets of probability distributions on $\mathcal{X} \times \{-1, 1\}$ such that both Assumption (A5) and Assumption (A1) with exponent $q$ (respectively Assumption (A2) with margin $\delta$) hold. We have the following lower bounds:*

- *There exists a constant $\kappa_1 > 0$ such that for any $n \in \mathbb{N}$, and any classifier $\hat{c}_n : (\mathcal{X} \times \{-1, 1\})^n \to \mathcal{M}(\mathcal{X}, \{-1, 1\})$, we have*

$$\sup_{\mathbb{P} \in \mathcal{P}_{s,q}^{(1)}} \left\{ \mathbb{E}_{\mathbb{P}^{\otimes n}} \left[ \mathcal{R}_{\mathbb{P}}(\hat{c}_n) - \mathcal{R}^* \right] \right\} \geq \begin{cases} \kappa_1 n^{-\frac{2q+2}{r(d+q-2)}} & \text{if } 0 < q \leq 1, \\ \kappa_1 n^{-\frac{s}{r(s+C_2 B_1 + B_2)}} & \text{if } q > 1. \end{cases} \tag{20}$$

- *There exists a constant $\kappa_2 > 0$ such that for any $n \in \mathbb{N}$, and any classifier $\hat{c}_n : (\mathcal{X} \times \{-1, 1\})^n \to \mathcal{M}(\mathcal{X}, \{-1, 1\})$, we have*

$$\sup_{\mathbb{P} \in \mathcal{P}_{s,\delta}^{(2)}} \left\{ \mathbb{E}_{\mathbb{P}^{\otimes n}} \left[ \mathcal{R}_{\mathbb{P}}(\hat{c}_n) - \mathcal{R}^* \right] \right\} \geq \kappa_2 n^{-\frac{s}{r(C_2 B_1 + B_2)}}. \tag{21}$$

We make some remarks regarding Theorem 6. First, we can see that for Assumption **(A1)** in the regime $0 < q \leq 1$, the excess risk bound of $\mathcal{O}\left(n^{-\frac{qs}{2r(s+C_2 B_1 + B_2)}}\right)$ provided by Theorem 4 is quite loose, as $C_2 B_1 + B_2 \geq 4d^2$. Likewise, in the regime $1 < q < 2$, we fall short of the minimax optimal lower bound by a factor of $q/2$. However, for both the $q \geq 2$ regime and the case where Assumption **(A2)** holds, we recover exactly the same rates as in Theorem 4. We still need to highlight however that, formally, the best rates in Theorem 4 can not be exactly attained, but can be approached arbitrarily close, hence the minimum norm DNN classifiers can not be said to be minimax optimal in a strict sense, but we can call them *essentially* minimax optimal, in the sense that they can achieve rates arbitrarily close to the optimal one.

### 3.4 Super fast rates under standard smoothness: tanh and sigmoid activations

#### 3.4.1 Upper bounds

A limitation of Theorem 4 is the reliance on Assumption **(A5)** whose complete characterization is challenging. In this section, we show that Assumption **(A5)** can in fact be replaced by standard $C^s(\mathcal{X})$ smoothness and lead to the exact same results in the case where the activation function $\sigma$ is chosen as tanh or sigmoid. To establish this, we will need the following theorem due to (De Ryck et al., 2021):

**Theorem 7** (Adapted from (De Ryck et al., 2021), Theorem 5.1). *Let $s \in \mathbb{N}$ and $h \in \mathcal{C}^s(\mathcal{X})$. For any $W_0 \in \mathbb{N}$ such that $W_0 \geq (3d/2)^{1/d}$, there exists a neural network $f(\cdot; \boldsymbol{\theta}) \in \mathcal{NN}(\mathbf{a}, W, L, R)$ with activation $\sigma \equiv \tanh$, width $W(\mathbf{a}) = C_1 W_0$, and depth $L(\mathbf{a}) = 2$ such that*

$$\|f(\cdot; \boldsymbol{\theta}) - h\|_{L^\infty(\mathcal{X})} \leq C_3 \|h\|_{\mathcal{C}^s(\mathcal{X})} W_0^{-2s/d},$$

*where $C_1 = 3d5^{d+1}$ and $C_3 = 3(3d/2)^s$.*

The remarkable improvement of Theorem 7 over the equivalent Theorem 3 for ReLU networks is that the same rates of approximation can be achieved with *constant depth*. Note also that, since the logistic sigmoid $\sigma : x \mapsto (e^x - e^{-x})/(e^x + e^{-x})$ is equal to the hyperbolic tangent up to an affine transformation, Theorem 7 applies to it in the exact same manner. We therefore deduce the following theorem, which extends Theorem 4 to the case where the regression function $\eta$ is assumed to be $\mathcal{C}^s$-smooth:

**Theorem 8.** *Let $\sigma$ be either the tanh or logistic sigmoid activation. Assume that Assumptions (A3) and (A4) hold, and assume that the regression function $\eta$ is in $\mathcal{C}^s(\mathcal{X})$. Let $\alpha > 0$ be a desired rate of convergence. For any $n \in \mathbb{N}$, there exists a FCNN architecture $\mathbf{a}_n$ with width $W_n$, depth $L_n$ and parameter bound $R_n$ satisfying*

$$\begin{cases} W_n & = \tilde{\mathcal{O}}(n^{\alpha d/2rs}) \\ L_n & = 2 \\ R_n & = R_0 \left[ L_n(W_n^2 + W_n) \right]^{1/p} = \tilde{\mathcal{O}}(n^{\alpha d/ps}), \end{cases}$$

*where $R_0$ is given in (A3), and $\tilde{\mathcal{O}}$ hides polylogarithmic factors, such that the following holds:*

- *If the low-noise condition (A1) holds, and $\alpha < \left(1 + \frac{B_2}{s}\right)^{-1}$, then any minimum-norm solution $\widehat{\boldsymbol{\theta}}_\lambda$ of the $\lambda$-ERM problem (9) with $\lambda = \tilde{\mathcal{O}}\left(n^{-\frac{2\alpha(s+d)}{rs}}\right)$, satisfies the excess risk bound:*

$$\mathcal{R}\left( \operatorname{sign} f(\cdot; \widehat{\boldsymbol{\theta}}_\lambda) \right) - \mathcal{R}^* \lesssim \max\left\{ n^{-\frac{\alpha}{r}}, n^{-\frac{\alpha q}{2r}} \right\}, \tag{22}$$

  *where $q > 0$ is the noise exponent in Assumption (A1), $r > 1$ is the "separation exponent" in (14), and $B_2 = 2rd\big[d(2 + 2p) - 1\big]/p$.*

- *If the hard-margin condition (A2) holds with margin $\delta > 0$, $\alpha < s/B_2$, and $n > (\delta/2)^{-\alpha/r}$, then any minimum-norm solution $\widehat{\boldsymbol{\theta}}_\lambda$ of the $\lambda$-ERM problem (9) with $\lambda = \tilde{\mathcal{O}}\left(n^{-\frac{2\alpha(s+d)}{rs}}\right)$ satisfies the excess risk bound:*

$$\mathcal{R}\left( \operatorname{sign} f(\cdot; \widehat{\boldsymbol{\theta}}_\lambda) \right) - \mathcal{R}^* \lesssim n^{-\frac{\alpha}{r}}, \tag{23}$$

  *where $r > 1$ is the "separation exponent" in (14), and $B_2 = 2rd\big[d(2 + 2p) - 1\big]/p$.*

### 3.4.2 Lower bounds

We now briefly discuss the optimality of the convergence rates given by Theorem 8 in the minimax sense. To that end, we recall the relevant minimax lower bound for binary classifiers under weak margin condition (A1), due to (Audibert and Tsybakov, 2007):

**Theorem 9** (Adapted from (Audibert and Tsybakov, 2007), Theorem 4.1)**.** *Let $s \in \mathbb{N}$ be fixed. Denote by $\mathcal{P}_{s,q}$ the set of probability distributions on $\mathcal{X} \times \{-1, 1\}$ such that $\eta \in \mathcal{C}^s(\mathcal{X})$ and the weak margin Assumption (A1) holds with exponent $q \in [0, \infty]$. There exists a constant $\kappa > 0$ such that for any $n \in \mathbb{N}$, and any classifier $\hat{c}_n : (\mathcal{X} \times \{-1, 1\})^n \to \mathcal{M}(\mathcal{X}, \{-1, 1\})$, we have*

$$\sup_{\mathbb{P} \in \mathcal{P}_{s,q}} \left\{ \mathbb{E}_{\mathbb{P}^{\otimes n}} \left[ \mathcal{R}_{\mathbb{P}}(\hat{c}_n) - \mathcal{R}^* \right] \right\} \geq \kappa n^{-\frac{s(q+1)}{s(q+2)+d}}. \tag{24}$$

Comparing the rate (22) we've established with the lower bound (24), we see that our bound, although comparable, is looser than the minimax optimal one, for all values of $q$. For the hard-margin case (A2) however, the situation is much more dramatic. Indeed, various authors, including (Koltchinskii and Beznosova, 2005; Audibert and Tsybakov, 2007; Smale and Zhou, 2007) have shown that *exponential* convergence rates of the form $\mathcal{O}(e^{-\alpha n})$ were possible when combining $\mathcal{C}^s$-smoothness with (A2). Our polynomial rates (23) in this same setting are thus far weaker, and it is an interesting question to check whether such exponential rates can genuinely be attained for fully connected neural networks under the hard-margin condition.

### 3.5 A case of exponential convergence rate: well-specified teacher-student learning

In light of the preceding discussion, we are compelled to investigate the existence of settings in which FCNNs classifiers can attain exponential rates. The takeaway message from Theorem 4 is that whenever the regression function $\eta$ lies in a suitable space, such that it can be approximated by FCNNs whose size grows slowly, the margin conditions **(A1)** and **(A2)** can lead to fast rates for the excess risk. Taking this idea a step further, we look in this subsection at what happens when the regression function $\eta$ is *exactly representable* by our hypothesis space of FCNNs. Our starting point is the following Lemma:

**Lemma 5.** *Let $R^* > 0, L^* \in \mathbb{N}$ be fixed, and $\mathbf{a}^* \in \mathbb{N}^{L^*+1}$ be an arbitrary FCNN architecture. For any parametrization $\boldsymbol{\theta}^* \in \mathcal{P}_{\mathbf{a}^*, R^*}$, there exists a distribution $\rho_{\boldsymbol{\theta}^*}$ on $\mathcal{X} \times \mathcal{Y}$ such that*

$$\mathbb{E}_{(X,Y)\sim\rho_{\boldsymbol{\theta}^*}}[Y \mid X = x] = f(x; \boldsymbol{\theta}^*), \quad \text{for } \rho_X\text{-a.e. } x \in \mathcal{X},$$

*where $f(\cdot; \boldsymbol{\theta}^*) : \mathcal{X} \to [-1, 1]$ is the function realized by $\boldsymbol{\theta}^*$.*

*Proof.* Let $X \sim \rho_X$ and $U \sim \text{Uniform}([-1,1])$ be two independent random variables on the same probability space, and define:

$$Y := \mathbb{1}[U \leq f(X; \boldsymbol{\theta}^*)] - \mathbb{1}[U > f(X; \boldsymbol{\theta}^*)] = \begin{cases} 1, & \text{if } U \leq f(X; \boldsymbol{\theta}^*), \\ -1, & \text{if } U > f(X; \boldsymbol{\theta}^*). \end{cases}$$

Now let $\rho_{\boldsymbol{\theta}^*}$ be the joint distribution of $(X, Y)$: we then have that for $\rho_X$-almost every $x \in \mathcal{X}$,

$$\begin{aligned} \mathbb{E}[Y \mid X = x] &= \mathbb{E}[\mathbb{1}[U \leq f(x; \boldsymbol{\theta}^*)] - \mathbb{1}[U > f(x; \boldsymbol{\theta}^*)]] \\ &= \mathbb{P}[U \leq f(x; \boldsymbol{\theta}^*)] - \mathbb{P}[U > f(x; \boldsymbol{\theta}^*)] \\ &= \frac{1}{2}(1 + f(x; \boldsymbol{\theta}^*)) - \frac{1}{2}(1 - f(x; \boldsymbol{\theta}^*)) \\ &= f(x; \boldsymbol{\theta}^*). \end{aligned}$$

$\square$

From Lemma 5, it follows that any function realized by a DNN can serve as the Bayes regression function $\eta(x) = \mathbb{E}[Y \mid X = x]$ of a carefully constructed classification problem. Specifically, given a target DNN $f(\cdot; \boldsymbol{\theta}^*)$ with fixed architecture and parameters, we can construct a data distribution $\rho_{\boldsymbol{\theta}^*}$ on $\mathcal{X} \times \mathcal{Y}$ such that the associated Bayes regression function $\eta$ is exactly $f(\cdot; \boldsymbol{\theta}^*)$. This observation can be thought of as a formalization of the *knowledge distillation* framework, which consists in training Neural Networks of small size to solve problems at which bigger Neural Networks are very successful with comparable performance. This approach, also known as the *teacher-student setting*, is typically implemented by training a smaller ("student") network to predict the outputs of a larger ("teacher") network, and has shown to be very successful in practice (Hinton et al., 2015; Xu et al., 2023).

Previous work has characterized the expressivity of deep ReLU fully-connected neural networks (FCNNs). Given an architecture $\mathbf{a}$ with input dimension $d$, width $W$, and depth $L$, the number of linear regions it can induce ranges from $\mathcal{O}(1)$ to $\mathcal{O}(W^{dL})$ (Montufar et al., 2014; Serra et al., 2018). This exponential gap suggests that a large network with width $W$ and depth $L$ can, in principle, be represented by a much smaller network with width $W' \ll W$ and depth $L' \ll L$. Such observations help explain the practical success of knowledge distillation and lend support to the *lottery ticket hypothesis*, which posits that large networks contain small subnetworks capable of comparable generalization performance (Frankle and Carbin, 2019).

Building on this understanding, we now examine the learning rates achievable in our idealized "teacher-student" setting, where the "teacher" network is realizable by the "student" network. In this ideal scenario, we can show that the hard-margin condition **(A2)** does lead to an exponential convergence rate of the excess risk, as formalized in the following theorem:

**Theorem 10.** *Let $f(\cdot; \boldsymbol{\theta}^*) \in \mathcal{NN}(\mathbf{a}^*, W^*, L^*, R^*)$ be a "teacher" neural network, and let $\rho_{\boldsymbol{\theta}^*}$ be a distribution on $\mathcal{X} \times \mathcal{Y}$ such that the conclusion of Lemma 5 holds. Suppose that the following conditions are satisfied for $\rho_{\boldsymbol{\theta}^*}$:*

1. Assumptions **(A3)** and **(A4)** hold.

2. For some $W_0 < W^*$, $L_0 < L^*$, $\mathbf{a}_0 \in \mathbb{N}^{L_0+1}$, and $R_0 > 0$, we have $f(\cdot; \boldsymbol{\theta}^*) \in \mathcal{NN}(\mathbf{a}_0, W_0, L_0, R_0)$.

3. The hard-margin condition **(A2)** holds with margin $\delta > 0$.

Then, for any minimum-norm solution $\widehat{\boldsymbol{\theta}}_\lambda$ of the $\lambda$-regularized ERM problem (9) over the hypothesis space $\mathcal{NN}(\mathbf{a}_0, W_0, L_0, R_0)$ with i.i.d. data $\big((x_i, y_i)\big)_{1 \le i \le n}$ sampled from $\rho_{\boldsymbol{\theta}^*}$, and with $0 \le \lambda \le \frac{\exp(-2n\beta_1)}{P(\mathbf{a}_0) R_0^P}$, the excess risk satisfies for all $n \ge 1$:

$$\mathcal{R}\left(\operatorname{sign} f(\cdot; \widehat{\boldsymbol{\theta}}_\lambda)\right) - \mathcal{R}^* \le \beta_2 \exp\left(-n\beta_1\right) + \frac{4}{\delta^2} \exp\left(-2n\beta_1\right),$$

where

$$\beta_1 = \frac{K^2 (2^{-L_0}\delta)^{2r}}{288 \operatorname{Lip}_{2R_0}(\mathcal{F}_{NN})^{2r}}, \quad \beta_2 = 1 + 4 \mathbf{Cov}_\infty\left(\mathcal{NN}, \frac{K(2^{-L_0}\delta)^r}{24 \operatorname{Lip}_{2R_0}(\mathcal{F}_{NN})^{1+r}}\right)$$

are constants which do not depend on $n$.

*Proof.* We have that the Bayes regression function $\eta$ is given by $f(\cdot; \boldsymbol{\theta}^*) \in \mathcal{NN}(\mathbf{a}_0, W_0, L_0, R_0)$. This means that when applying Theorem 2 for the hypothesis space $\mathcal{NN}(\mathbf{a}_0, W_0, L_0, R_0)$, we get $\varepsilon_{\text{approx}} = 0$, while $R_0, L_0, P(\mathbf{a}_0), \operatorname{Lip}_{2R_0}(\mathcal{F}_{NN})$ are all independent of $n$. Applying Theorem 2 with $\nu \equiv \delta/2$ thus immediately yields the desired result. $\qquad\square$

**Remark 2.** *Interestingly, in this well-specified setting, directly applying Theorem 2 under* **(A2)** *does not yield improved results compared to those in the previous sections. That being said, since* **(A2)** *implies* **(A1)** *for arbitrarily large $q > 0$, and since the realizability assumption 2 implies* **(A5)** *with arbitrarily large $s > 0$, we can apply Theorem 4 to obtain a rate of $\mathcal{O}(n^{(1-\epsilon)/r})$ for arbitrarily small $\epsilon > 0$ in this setting.*

## 4 Numerical experiments

In this section, we illustrate Theorem 4 through simple numerical experiments, considering both the weak margin and strong margin cases. To achieve this, we generate data points from two toy distributions and empirically verify that the margin assumptions **(A1)** and **(A2)** are satisfied. This verification is performed by plotting the histogram of the outputs of a DNN trained on each distribution using the square loss.

For each case, we train a DNN with the square loss for 5000 epochs, using an increasing number of training samples $n$, and plot the evolution of the test error as $n$ grows. Consistent with Theorem 4, we observe a convergence rate slower than $O(n^{-1})$ under the weak margin condition **(A1)**, and faster than $O(n^{-1})$ under the hard margin condition **(A2)**.

In all the numerical experiments presented below, the DNN under consideration is a fully connected ReLU network with architecture $\mathbf{a} = (2, 64, 32, 1)$.

## 4.1 Convergence rate under weak margin condition

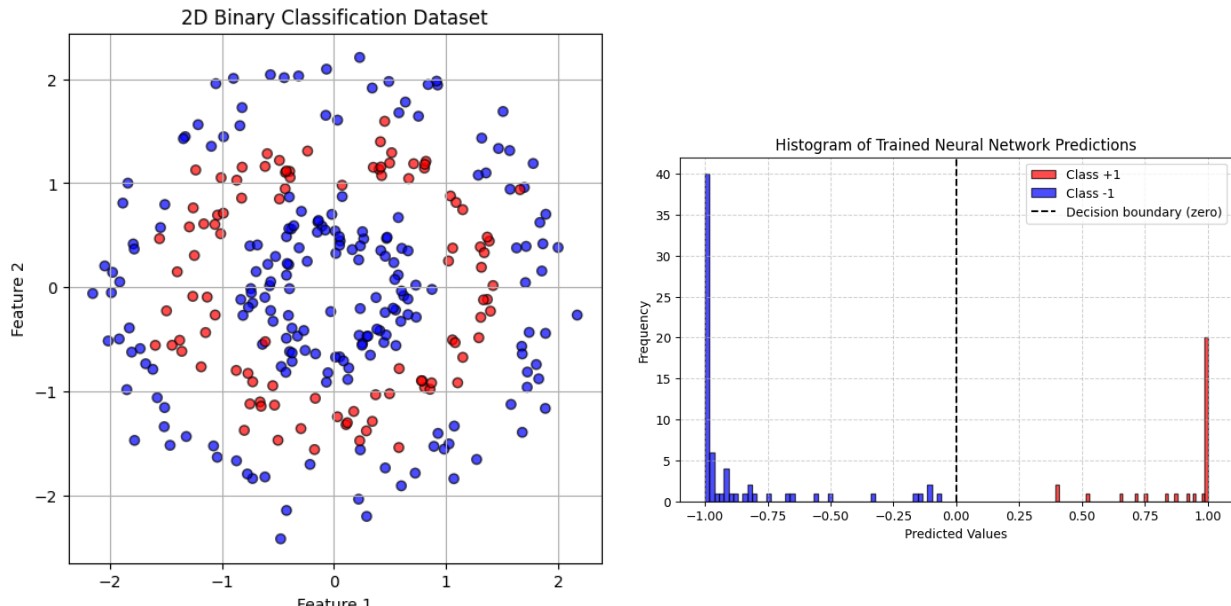

(a) A toy dataset where the weak margin condition holds. (b) Histogram of predicted labels for a DNN trained on this dataset.

Figure 3: A toy dataset which satisfies the weak margin condition. By plotting the histogram of a DNN trained to predict each class label, we confirm empirically that the condition is satisfied.

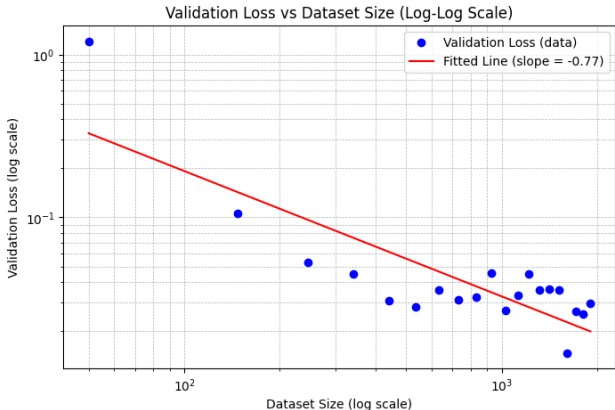

Figure 4: Evolution of the testing error for a DNN trained on $n$ datapoints. The linear fit suggest an error decay rate of $O(n^{-0.77})$.

## 4.2 Convergence rate under strong margin condition

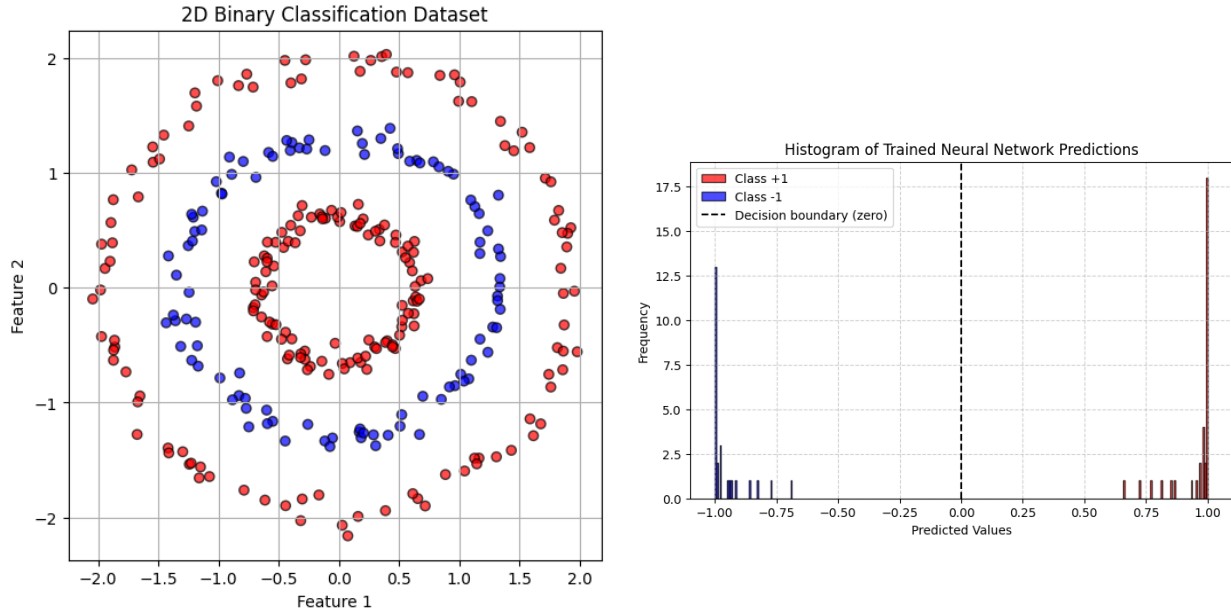

(a) A toy dataset where the hard margin condition holds. (b) Histogram of predicted labels for a DNN trained on this dataset.

Figure 5: A toy dataset which satisfies the hard margin condition. By plotting the histogram of a DNN trained to predict each class label, we confirm empirically that the condition is satisfied.

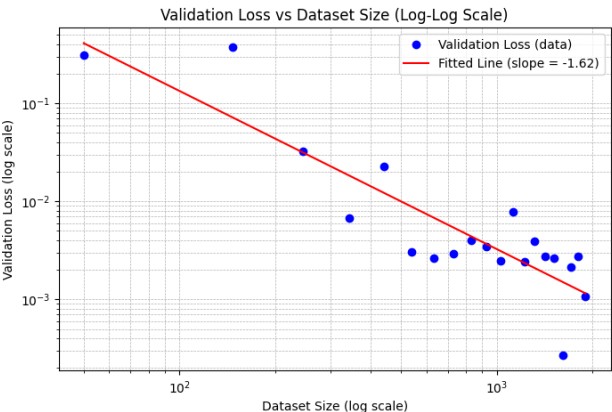

Figure 6: Evolution of the testing error for a DNN trained on $n$ datapoints. The linear fit suggest an error decay rate of $O(n^{-1.62})$.

## 5   Conclusion and discussion

We have established in this work a general upper bound on the excess risk of square-loss ERM classifiers induced by parametric function classes under Mammen-Tsybakov margin conditions. As a consequence, we have demonstrated for the first time "super-fast" rates of convergence for suitably overparametrized DNN classifiers with diverse activation functions under suitable regularity conditions on the regression function $\eta$ and the marginal data distribution $\rho_X$. Unlike the related results of (Kim et al., 2021), we do not require

optimizing over a space of networks with pre-specified sparsity and uniformly bounded output functions, making our result more relevant to practical applications. Together, these findings show that DNNs can leverage margin conditions at least as effectively as standard methods for supervised classification. Although our analysis is exclusively limited to the binary case, we believe that by defining appropriate notions of Bayes regression function and margin conditions tailored to the multiclass setting, as done in (Vigogna et al., 2022), our argument should extend to that case too.

We now discuss in this section some interesting open questions which arise from this work that we believe would be worth investigating further:

- **Full characterization of (A5).** Our analysis crucially relies on Assumption (A5) to carry through. Although we have provided some sufficient conditions for this Assumption to hold in Section 6.5, a full characterization of the pairs $(\eta, \rho_X)$ under which this Assumption is satisfied would greatly improve our understanding of this "super-fast rates" phenomenon for Deep Neural Networks.

- **More efficient approximation.** It would be interesting to investigate in more detail the existence of architectures and activation functions whose approximation power and complexity scale such that usual smoothness assumptions on $\eta$ are enough to derive these super-fast rates, akin to our Theorem 8 for tanh and sigmoid. We believe that potential such candidates could either be shallow neural networks over appropriate smoothness spaces (Yang and Zhou, 2024), DNNs with so-called "super-expressive" activation functions (Wang et al., 2025), or flexible and parameter-efficient architectures such as Convolutional Neural Networks (Zhou, 2020) or Kolmogorov-Arnold Networks (Liu et al., 2025).

- **Other loss functions.** Our theoretical analysis crucially relies on a property that is specific to the square loss: namely that its population minimizer is the regression function $\eta$ and that the excess surrogate risk can be identified with $\|f - \eta\|_{L^2(\rho_X)}^2$. For other commonly used surrogates such as hinge loss or cross-entropy, the population minimizer typically differs from $\eta$, and relating surrogate excess risk to misclassification excess risk requires additional tools such as classification-calibration bounds, which are particularly well understood for convex surrogates (Bartlett et al., 2006). In this direction, fast rates for DNN classification with logistic loss have recently been established (Zhang et al., 2024b). Understanding how to extend these results to other surrogates and to the super-fast regime is an interesting direction to explore in future work.

# 6 Proofs

## 6.1 Proof of Lemma 2

*Proof of Lemma 2.* Proof of (i.): Denote by $\mathbf{0} \in \mathcal{P}_{\mathbf{a},\infty}$ the parametrization whose entries are all zeros, and $b := \widehat{\mathcal{R}}_{\ell,\lambda}(\mathbf{0})$. Clearly, $\widehat{\mathcal{R}}_{\ell,\lambda}(f(\cdot;\boldsymbol{\theta})) > b$ for all $|\boldsymbol{\theta}|_p^p > b/\lambda$, hence $\widehat{\mathcal{R}}_{\ell,\lambda}$ is minimized somewhere in $\{\boldsymbol{\theta} : |\boldsymbol{\theta}|_p^p \le b/\lambda\}$ and the minimum is attained by compactness and continuity.

Now let $0 \le \lambda \le \lambda'$ and $\boldsymbol{\theta}, \boldsymbol{\theta}'$ respectively in argmin* $\widehat{\mathcal{R}}_{\ell,\lambda}$ and argmin* $\widehat{\mathcal{R}}_{\ell,\lambda'}$. By optimality we have

$$
\begin{aligned}
\widehat{\mathcal{R}}_\ell(\boldsymbol{\theta}) + \lambda|\boldsymbol{\theta}|_p^p &\le \widehat{\mathcal{R}}_\ell(\boldsymbol{\theta}') + \lambda|\boldsymbol{\theta}'|_p^p \\
&= \widehat{\mathcal{R}}_\ell(\boldsymbol{\theta}') + \lambda'|\boldsymbol{\theta}'|_p^p + (\lambda - \lambda')|\boldsymbol{\theta}'|_p^p \\
&\le \widehat{\mathcal{R}}_\ell(\boldsymbol{\theta}) + \lambda'|\boldsymbol{\theta}|_p^p + (\lambda - \lambda')|\boldsymbol{\theta}'|_p^p
\end{aligned}
$$

Hence we have shown

$$
0 \le (\lambda - \lambda')\left(|\boldsymbol{\theta}'|_p^p - |\boldsymbol{\theta}|_p^p\right) \tag{25}
$$

which implies that $|\boldsymbol{\theta}'|_p \leq |\boldsymbol{\theta}|_p$ whenever $\lambda \leq \lambda'$. Finally, by basic properties of $\ell_p$ norms, we have

$$
\begin{aligned}
\sup_{\lambda \geq 0, n \geq 1} \left\{ |\boldsymbol{\theta}|_\infty : \boldsymbol{\theta} \in \operatorname{argmin}^* \widehat{\mathcal{R}}_{\ell,\lambda} \right\} &\leq \sup_{\lambda \geq 0, n \geq 1} \left\{ |\boldsymbol{\theta}|_p : \boldsymbol{\theta} \in \operatorname{argmin}^* \widehat{\mathcal{R}}_{\ell,\lambda} \right\} \\
&\leq \sup_{n \geq 1} \left\{ |\boldsymbol{\theta}|_p : \boldsymbol{\theta} \in \operatorname{argmin}^* \widehat{\mathcal{R}}_{\ell,n} \right\} \\
&\leq \sup_{n \geq 1} \left\{ P(\mathbf{a})^{1/p} |\boldsymbol{\theta}|_\infty : \boldsymbol{\theta} \in \operatorname{argmin}^* \widehat{\mathcal{R}}_{\ell,n} \right\} \\
&= R_0 P(\mathbf{a})^{1/p}
\end{aligned}
$$

Proof of (ii.): First note that for any $f \in L^2(\rho_X)$, we have

$$
\begin{aligned}
\mathcal{R}_\ell(f) &:= \mathbb{E}_{(x,y)\sim\rho} \left[ (f(x) - y)^2 \right] \\
&= \mathbb{E}_{(x,y)\sim\rho} \left[ (f(x) - \eta(x))^2 \right] + \mathbb{E}_{(x,y)\sim\rho} \left[ (\eta(x) - y)^2 \right] + 2\mathbb{E}_{(x,y)\sim\rho} \left[ (f(x) - y)(\eta(x) - y) \right] \\
&= \|f - \eta\|_{L^2(\rho_X)}^2 + C + 2\mathbb{E}_{(x,y)\sim\rho} \mathbb{E} \left[ (f(x) - \eta(x))(\eta(x) - y) \mid x \right] \\
&= \|f - \eta\|_{L^2(\rho_X)}^2 + C + 2\mathbb{E}_{(x,y)\sim\rho} \left[ (f(x) - \eta(x))(\eta(x) - \mathbb{E}[y \mid x]) \right] \\
&= \|f - \eta\|_{L^2(\rho_X)}^2 + C + 0
\end{aligned}
$$

Where $C \equiv \mathbb{E}_{(x,y)\sim\rho} \left[ (\eta(x) - y)^2 \right] \geq 0$ is a constant which does not depend on $f$. This shows that minimizing $\mathcal{R}_\ell$ is equivalent to minimizing the $L^2(\rho_X)$ distance to $\eta$. It is thus enough to show that there exists $\boldsymbol{\theta} \in \operatorname{argmin}_{\boldsymbol{\theta} \in \mathcal{P}_{\mathbf{a},\infty}} \mathcal{R}_\ell$ with $|\boldsymbol{\theta}|_\infty \leq R^*$ to conclude.

To that end, observe that by the strong law of large numbers, we have for any $\boldsymbol{\theta} \in [-R^*, R^*]^{P(\mathbf{a})}$ that $\widehat{\mathcal{R}}_{\ell,n}(\boldsymbol{\theta}) \to \mathcal{R}_\ell(\boldsymbol{\theta})$ almost surely as $n \to \infty$. Furthermore, because the realization mapping $\mathcal{F}$ is Lipschitz, its composition with the mapping $\ell : (f, x, y) \mapsto (f(x) - y)^2$ is uniformly Lipschitz over $(x, y) \in \mathcal{X} \times \{-1, +1\}$, and we can denote by $L_R > 0$ its Lipschitz constant. Finally, for any convergent subsequence $(\boldsymbol{\theta}_n)_{n \geq 1} \subseteq [-R^*, R^*]^{P(\mathbf{a})}$ satisfying $\boldsymbol{\theta}_n \in \operatorname{argmin}_{\boldsymbol{\theta} \in \mathcal{P}_{\mathbf{a},\infty}} \widehat{\mathcal{R}}_{\ell,n}$ for all $n \geq 1$ (which is guaranteed to exist by Assumption (A3)) and with limit $\boldsymbol{\theta}^*$, we have

$$
|\widehat{\mathcal{R}}_{\ell,n}(\boldsymbol{\theta}_n) - \mathcal{R}_\ell(\boldsymbol{\theta}^*)| \leq L_R |\boldsymbol{\theta}_n - \boldsymbol{\theta}^*|_\infty + |\mathcal{R}_\ell(\boldsymbol{\theta}_n) - \mathcal{R}_\ell(\boldsymbol{\theta}^*)|.
$$

Thus for any $\boldsymbol{\theta} \in \mathcal{P}_{\mathbf{a},\infty}$, we can take the limit $n \to \infty$ to find

$$
\widehat{\mathcal{R}}_{\ell,n}(\boldsymbol{\theta}_n) \leq \widehat{\mathcal{R}}_{\ell,n}(\boldsymbol{\theta}) \implies \mathcal{R}_\ell(\boldsymbol{\theta}^*) \leq \mathcal{R}_\ell(\boldsymbol{\theta}),
$$

which implies that $\boldsymbol{\theta}^* \in \operatorname{argmin}_{\boldsymbol{\theta} \in \mathcal{P}_{\mathbf{a},\infty}} \mathcal{R}_\ell$, as desired. $\qquad\square$

## 6.2 Proof of Proposition 1

To prove Proposition 1, we will need the following growth estimate for functions satisfying the KL property, due to (van Ngai and Théra, 2009):

**Theorem 11** (Adapted from Corollary 2.(ii) in (van Ngai and Théra, 2009)). *Let $f : \mathbb{R}^k \to [0, +\infty)$ be a lower semi-continuous function, and $f(x_0) = 0$. If there exist $c, \gamma, \varepsilon > 0$ such that $\gamma |x^*|_2 [f(x)]^{\gamma-1} \geq c$ for all $x \in \{x : |x - x_0|_2 < \varepsilon\} \setminus \{x : f(x) = 0\}$ and $x^* \in \hat{\partial} f(x)$, then*

$$
\operatorname{dist}_2(x, \{x : f(x) = 0\}) \leq \frac{1}{c} [f(x)]^\gamma, \quad \text{for all } x \in \{x : |x - x_0|_2 < \varepsilon/2\},
$$

*where $\operatorname{dist}_2(a, A)$ denotes the $\ell_2$ distance between a vector $a \in \mathbb{R}^k$ and a set $A \subseteq \mathbb{R}^k$.*

*Proof of Proposition 1.* Let $\Lambda > 0$ be arbitrary, and write $R := R_0 P(\mathbf{a})^{1/p}$. By reproducing the argument which led to inequality (25) above, we have that all population minimizers under consideration lie in $\mathcal{P}_{\mathbf{a},R}$ uniformly over $0 < \lambda \leq \Lambda$. For $0 \leq \lambda \leq \Lambda$, let

$$
S_\lambda := \operatorname*{argmin}_{\boldsymbol{\theta} \in \mathcal{P}_{\mathbf{a},R}} \mathcal{R}_{\ell,\lambda}(\boldsymbol{\theta}), \qquad m_\lambda := \min_{\boldsymbol{\theta} \in \mathcal{P}_{\mathbf{a},R}} \mathcal{R}_{\ell,\lambda}(\boldsymbol{\theta}), \qquad \psi_\lambda(\boldsymbol{\theta}) := \mathcal{R}_{\ell,\lambda}(\boldsymbol{\theta}) - m_\lambda.
$$

Then $S_\lambda$ is nonempty and compact, $\psi_\lambda \geq 0$ on $\mathcal{P}_{\mathbf{a},R}$, and $\{\psi_\lambda = 0\} = S_\lambda$. Moreover, $\mathrm{Lip}_R(\psi_\lambda) = \mathrm{Lip}_R(\mathcal{R}_{\ell,\lambda})$.

Define
$$\psi(\lambda, \boldsymbol{\theta}) := \psi_\lambda(\boldsymbol{\theta}), \qquad \Omega := \{(\lambda, \boldsymbol{\theta}) \in [0, \Lambda] \times \mathcal{P}_{\mathbf{a},R} : \psi(\lambda, \boldsymbol{\theta}) = 0\} = \{(\lambda, \boldsymbol{\theta}) : \boldsymbol{\theta} \in S_\lambda\}.$$
Then $\Omega$ is compact by continuity of $\psi$. Furthermore, closure of subanalyticity under addition and multiplication shows that $(\lambda, \boldsymbol{\theta}) \mapsto \mathcal{R}_{\ell,\lambda}(\boldsymbol{\theta})$ is subanalytic on $[0, \Lambda] \times \mathcal{P}_{\mathbf{a},R}$. Since $m_\lambda = \min_{\boldsymbol{\theta} \in \mathcal{P}_{\mathbf{a},R}} \mathcal{R}_{\ell,\lambda}(\boldsymbol{\theta})$ is obtained by minimizing a subanalytic function over a compact set, $\lambda \mapsto m_\lambda$ is subanalytic on $[0, \Lambda]$, hence $\psi$ is subanalytic on $[0, \Lambda] \times \mathcal{P}_{\mathbf{a},R}$ (Shiota, 1997).

We claim that there exist constants $c, \rho > 0$ and $0 < \kappa < 1$, independent of $\lambda \in [0, \Lambda]$, such that for all $0 \leq \lambda \leq \Lambda$,
$$|\boldsymbol{\theta}^*|_2 \, \psi_\lambda(\boldsymbol{\theta})^{-\kappa} \geq c, \quad \text{for all } \boldsymbol{\theta} \in \mathcal{P}_{\mathbf{a},R} \setminus S_\lambda \text{ with } \psi_\lambda(\boldsymbol{\theta}) \leq \rho, \text{ and all } \boldsymbol{\theta}^* \in \partial\psi_\lambda(\boldsymbol{\theta}). \tag{26}$$
Indeed, fix $(\bar{\lambda}, \bar{\boldsymbol{\theta}}) \in \Omega$. Applying Theorem 1 to the subanalytic function $\psi$ at $(\bar{\lambda}, \bar{\boldsymbol{\theta}})$ (viewing $\lambda$ as an additional variable) yields an open neighborhood $U_{\bar{\lambda}, \bar{\boldsymbol{\theta}}}$ and constants $c_{\bar{\lambda}, \bar{\boldsymbol{\theta}}}, \rho_{\bar{\lambda}, \bar{\boldsymbol{\theta}}} > 0$ and $0 < \kappa_{\bar{\lambda}, \bar{\boldsymbol{\theta}}} < 1$ such that for all $(\lambda, \boldsymbol{\theta}) \in U_{\bar{\lambda}, \bar{\boldsymbol{\theta}}}$ with $0 < \psi_\lambda(\boldsymbol{\theta}) \leq \rho_{\bar{\lambda}, \bar{\boldsymbol{\theta}}}$ and all $\boldsymbol{\theta}^* \in \partial\psi_\lambda(\boldsymbol{\theta})$,
$$|\boldsymbol{\theta}^*|_2 \geq c_{\bar{\lambda}, \bar{\boldsymbol{\theta}}} \, \psi_\lambda(\boldsymbol{\theta})^{\kappa_{\bar{\lambda}, \bar{\boldsymbol{\theta}}}}.$$
By compactness of $\Omega$, we can extract a finite subcover $\{U_i\}_{i=1}^N$ with associated constants $(c_i, \rho_i, \kappa_i)$, and set
$$c := \min_{1 \leq i \leq N} c_i, \qquad \kappa := \max_{1 \leq i \leq N} \kappa_i, \qquad \rho_0 := \min_{1 \leq i \leq N} \rho_i, \qquad U := \bigcup_{i=1}^N U_i.$$
Since $([0, \Lambda] \times \mathcal{P}_{\mathbf{a},R}) \setminus U$ is compact and disjoint from $\Omega$, we have $\psi > 0$ on it, hence
$$\rho_1 := \min_{(\lambda, \boldsymbol{\theta}) \in ([0,\Lambda] \times \mathcal{P}_{\mathbf{a},R}) \setminus U} \psi_\lambda(\boldsymbol{\theta}) > 0.$$
Finally set $\rho := \min\{\rho_0, \rho_1, 1\}$. If $0 < \psi_\lambda(\boldsymbol{\theta}) \leq \rho$, then $(\lambda, \boldsymbol{\theta}) \in U$, hence lies in some $U_i$, and
$$|\boldsymbol{\theta}^*|_2 \geq c_i \, \psi_\lambda(\boldsymbol{\theta})^{\kappa_i} \geq c \, \psi_\lambda(\boldsymbol{\theta})^\kappa,$$
using $\psi_\lambda(\boldsymbol{\theta}) \leq 1$ and $\kappa_i \leq \kappa$. This proves (26).

Fix $0 \leq \lambda \leq \Lambda$ and denote $L_\lambda := \mathrm{Lip}_R(\mathcal{R}_{\ell,\lambda}) = \mathrm{Lip}_R(\psi_\lambda)$. For any $\bar{\boldsymbol{\theta}} \in S_\lambda$ and any $\boldsymbol{\theta}$ with $|\boldsymbol{\theta} - \bar{\boldsymbol{\theta}}|_2 \leq \rho/L_\lambda$, Lipschitz continuity gives $\psi_\lambda(\boldsymbol{\theta}) \leq \rho$, hence (26) holds (and also for $\hat{\partial}\psi_\lambda$ since $\hat{\partial} \subseteq \partial$). Applying Theorem 11 with $f = \psi_\lambda$, $\gamma = 1 - \kappa$, and $\varepsilon = \rho/L_\lambda$, we obtain
$$\mathrm{dist}_2(\boldsymbol{\theta}, S_\lambda) \leq \frac{1}{(1-\kappa)c} \, \psi_\lambda(\boldsymbol{\theta})^{1-\kappa}, \quad \text{for all } \boldsymbol{\theta} \in \mathbb{R}^{P(\mathbf{a})} \text{ s.t. } \mathrm{dist}_2(\boldsymbol{\theta}, S_\lambda) \leq \frac{\rho}{2L_\lambda}.$$
Since $\mathrm{dist}(\cdot, S_\lambda) \leq \mathrm{dist}_2(\cdot, S_\lambda)$, this implies
$$\psi_\lambda(\boldsymbol{\theta}) \geq K_0 \, \mathrm{dist}(\boldsymbol{\theta}, S_\lambda)^r, \quad \text{for all } \boldsymbol{\theta} \in \mathcal{P}_{\mathbf{a},R} \text{ s.t. } \mathrm{dist}(\boldsymbol{\theta}, S_\lambda) \leq \frac{\rho}{2\sqrt{P(\mathbf{a})} \, L_\lambda},$$
where $r := 1/(1-\kappa) > 1$ and $K_0 := ((1-\kappa)c)^{1/(1-\kappa)}$. Slightly abusing notation and writing $\rho \equiv \rho/\sqrt{P(\mathbf{a})}$ yields the same estimate with the threshold $\rho/(2L_\lambda)$.

Set $L_* := \sup_{0 \leq \lambda \leq \Lambda} \mathrm{Lip}_R(\mathcal{R}_{\ell,\lambda}) < \infty$ and $\delta := \rho/(2L_*)$. Consider the compact set
$$A := \{(\lambda, \boldsymbol{\theta}) \in [0, \Lambda] \times \mathcal{P}_{\mathbf{a},R} : \mathrm{dist}(\boldsymbol{\theta}, S_\lambda) \geq \delta\}.$$
By continuity and the fact that $\psi_\lambda(\boldsymbol{\theta}) > 0$ on $A$, we have
$$\alpha := \min_{(\lambda, \boldsymbol{\theta}) \in A} \psi_\lambda(\boldsymbol{\theta}) > 0.$$
Finally, let $K := \min\{K_0, \alpha/(2R)^r\}$. Fix $0 \leq \lambda \leq \Lambda$ and $0 < t < \rho/(2\,\mathrm{Lip}_R(\mathcal{R}_{\ell,\lambda}))$. If $\boldsymbol{\theta} \in \mathcal{P}_{\mathbf{a},R}$ satisfies $\mathrm{dist}(\boldsymbol{\theta}, S_\lambda) \geq t$, then either $\mathrm{dist}(\boldsymbol{\theta}, S_\lambda) \leq \rho/(2\,\mathrm{Lip}_R(\mathcal{R}_{\ell,\lambda}))$, in which case
$$\psi_\lambda(\boldsymbol{\theta}) \geq K_0 \, t^r \geq K t^r,$$
or $\mathrm{dist}(\boldsymbol{\theta}, S_\lambda) \geq \delta$, in which case $\psi_\lambda(\boldsymbol{\theta}) \geq \alpha \geq K(2R)^r \geq K t^r$ since $\mathrm{dist}(\boldsymbol{\theta}, S_\lambda) \leq 2R$ on $\mathcal{P}_{\mathbf{a},R}$. This proves (14) (with $\arg\min \mathcal{R}_{\ell,\lambda} = S_\lambda$ on $\mathcal{P}_{\mathbf{a},R}$) with constants $K, \rho, r$ independent of $\lambda \in [0, \Lambda]$. $\qquad \square$

### 6.3 Proof of Lemma 3

*Proof of Lemma 3.* Fix $\boldsymbol{\theta}, \boldsymbol{\theta}' \in \mathcal{P}_{\mathbf{a}, R}$ and set $r := |\boldsymbol{\theta} - \boldsymbol{\theta}'|_\infty$. For $s \in \{1, \ldots, L\}$ let $f_s$ (resp. $g_s$) denote the depth-$s$ truncation of the network associated with $\boldsymbol{\theta}$ (resp. $\boldsymbol{\theta}'$), i.e.

$$f_1(x) = W_1 x + B_1, \qquad f_s(x) = W_s \,\sigma\big(f_{s-1}(x)\big) + B_s \quad (s \geq 2),$$

and similarly for $g_s$. For vector-valued functions we use the $\ell_\infty$ norm on the codomain and define

$$e_0 := 0, \quad e_s := \|f_s - g_s\|_{L^\infty(\mathcal{X})}, \qquad m_0 := 1, \quad m_s := \max\{1, \|f_s\|_{L^\infty(\mathcal{X})}, \|g_s\|_{L^\infty(\mathcal{X})}\}.$$

Since $\mathcal{X} = [0,1]^d$, we have $|x|_\infty \leq 1$ for all $x \in \mathcal{X}$. Moreover, note that if $A \in [-R, R]^{k \times n}$ and $z \in \mathbb{R}^n$, then $|Az|_\infty \leq nR|z|_\infty \leq WR|z|_\infty$ because $n \leq W$. Finally, with $C_\sigma := \max\{1, L_\sigma + |\sigma(0)|\}$ we have for all $z \in \mathbb{R}^n$,

$$|\sigma(z)|_\infty \leq L_\sigma |z|_\infty + |\sigma(0)| \leq C_\sigma \max\{1, |z|_\infty\}.$$

Hence, for every $s \in \{2, \ldots, L\}$,

$$m_s \leq RW \, C_\sigma \, m_{s-1} + R.$$

Likewise, for $s \geq 2$ we decompose

$$f_s - g_s = (W_s - W_s')\sigma(f_{s-1}) + W_s'\big(\sigma(f_{s-1}) - \sigma(g_{s-1})\big) + (B_s - B_s')$$

and bound each term using the previous estimates and the Lipschitz property of $\sigma$ to get

$$e_s \leq W\big(r \, C_\sigma \, m_{s-1} + R \, C_\sigma \, e_{s-1}\big) + r.$$

Set $A := RWC_\sigma$. First note that $m_1 \leq WR \sup_{x \in \mathcal{X}} |x|_\infty + R \leq WR + R$, and for every $s \in \{2, \ldots, L\}$ the inequality $m_s \leq A \, m_{s-1} + R$ holds. Hence, for every $s \in \{2, \ldots, L\}$ we have the explicit bound

$$m_{s-1} \leq (RW + R)A^{s-2} + R \sum_{j=0}^{s-3} A^j. \tag{27}$$

Moreover, $e_1 \leq (W+1)r$, and for every $s \in \{2, \ldots, L\}$ the recursion

$$e_s \leq A \, e_{s-1} + WC_\sigma \, r \, m_{s-1} + r$$

holds. Iterating it gives

$$e_L \leq A^{L-1}e_1 + r \sum_{s=2}^{L} A^{L-s}\big(WC_\sigma \, m_{s-1} + 1\big). \tag{28}$$

Plugging (27) into (28) and using the identities

$$\sum_{s=2}^{L} A^{L-s} A^{s-2} = (L-1)A^{L-2}, \qquad \sum_{s=3}^{L} \sum_{j=0}^{s-3} A^{L-s+j} = \sum_{t=0}^{L-3} (t+1)A^t,$$

we obtain

$$e_L \leq r\Big[(W+1)A^{L-1} + (L-1)WC_\sigma(RW+R)A^{L-2} + WC_\sigma R \sum_{t=0}^{L-3}(t+1)A^t + \sum_{t=0}^{L-2} A^t\Big]. \tag{29}$$

Since $A = RWC_\sigma \geq 1$ and $WC_\sigma R = A$, we bound

$$WC_\sigma R \sum_{t=0}^{L-3}(t+1)A^t = \sum_{t=0}^{L-3}(t+1)A^{t+1} \leq \frac{(L-2)(L-1)}{2}\, A^{L-1}, \qquad \sum_{t=0}^{L-2} A^t \leq (L-1)\, A^{L-1}.$$

Moreover, $WC_\sigma(RW+R) = (W+1)A$, hence the bracketed expression in (29) is at most

$$A^{L-1}\Big(L(W+1) + \frac{L(L-1)}{2}\Big) \leq 2L^2 W A^{L-1} = 2L^2(C_\sigma R)^{L-1}W^L,$$

which implies the desired Lipschitz bound. $\square$

### 6.4 Upper bounds

#### 6.4.1 Preliminary results

We start by collecting a number of useful lemmas which will be needed to prove the main results. Throughout the following, recall the definition of the *misclassification risk* $\mathcal{R}(\text{sign } f)$ (1) for a real-valued function $f$ :

$$\mathcal{R}(\text{sign } f) := \mathbb{P}_{(X,Y)\sim\rho}(\text{sign } f(X) \neq Y)$$

Our first lemma is a bound on the difference of the misclassification risks of classifiers induced by measurable functions $f, g \in L^\infty(\mathcal{X})$ :

**Lemma 6.** *For any two $f, g \in L^\infty(\mathcal{X})$, we have*

$$|\mathcal{R}(\text{sign } f) - \mathcal{R}(\text{sign } g)| \leq \mathbb{P}_{x\sim\rho_X}\left(\|f - g\|_{L^\infty(\mathcal{X})} \geq |f(x)|\right)$$

*Proof of Lemma 6.* We have

$$
\begin{aligned}
|\mathcal{R}(\text{sign } f) - \mathcal{R}(\text{sign } g)| &= |\mathbb{E}\left[\mathbb{1}\{\text{sign } f(X) \neq Y\} - \mathbb{1}\{\text{sign } g(X) \neq Y\}\right]| \\
&\leq \mathbb{E}\left[|\mathbb{1}\{\text{sign } f(X) \neq Y\} - \mathbb{1}\{\text{sign } g(X) \neq Y\}|\right] \\
&\leq \mathbb{E}\left[\mathbb{1}\{\text{sign } f(X) \neq \text{sign } g(X)\}\right] = \mathbb{P}\left(\text{sign } f(X) \neq \text{sign } g(X)\right)
\end{aligned}
$$

But now observe that for any $x \in \mathcal{X}$, $\text{sign } f(x) \neq \text{sign } g(x) \implies |f(x) - g(x)| \geq |f(x)|$. Hence the inclusion of events

$$\{\text{sign } f(X) \neq \text{sign } g(X)\} \subseteq \left\{\|f - g\|_{L^\infty(\rho_X)} \geq |f(X)|\right\},$$

which implies the claimed inequality. $\qquad\square$

We next have an upper bound on the excess misclassification risk of a classifier $\text{sign } f$ in terms of the $L^2(\rho_X)$ distance between $f$ and the regression function $\eta$.

**Lemma 7.** *For any $f \in L^2(\rho_X)$, we have the inequality*

$$\mathcal{R}(\text{sign } f) - \mathcal{R}(\text{sign } \eta) \leq \|f - \eta\|_{L^2(\rho_X)}$$

*Proof.* Note that we have

$$\eta(X) = \mathbb{E}[Y \mid X] = \mathbb{P}(Y = 1 \mid X) - \mathbb{P}(Y = -1 \mid X),$$

hence by the law of total expectation :

$$
\begin{aligned}
\mathcal{R}(\text{sign } f) - \mathcal{R}(\text{sign } \eta) &= \mathbb{E}_X\left[\mathbb{E}_Y\left[\mathbb{1}\{\text{sign } f(X) \neq Y\} - \mathbb{1}\{\text{sign } \eta(X) \neq Y\} \mid X\right]\right] \\
&= \mathbb{E}_X\left[(\mathbb{1}\{\text{sign } f(X) \neq 1\} - \mathbb{1}\{\text{sign } \eta(X) \neq 1\}) \cdot \mathbb{P}(Y = 1 \mid X)\right. \\
&\quad \left.+ (\mathbb{1}\{\text{sign } f(X) \neq -1\} - \mathbb{1}\{\text{sign } \eta(X) \neq -1\}) \cdot \mathbb{P}(Y = -1 \mid X)\right] \\
&\leq \mathbb{E}_X\left[|\eta(X)|\mathbb{1}\{\text{sign } f(X) \neq \text{sign } \eta(X)\}\right] \\
&\leq \mathbb{E}_X\left[|\eta(X) - f(X)|\mathbb{1}\{\text{sign } f(X) \neq \text{sign } \eta(X)\}\right] \\
&\leq \|f - \eta\|_{L^2(\rho_X)}
\end{aligned}
$$

$\qquad\square$

The following result states that, whenever $\eta$ satisfies either the low-noise Assumption **(A1)** or the hard margin condition **(A2)**, any sufficiently good $L^2(\rho_X)$ approximation of $\eta$ will satisfy the same assumption with high probability.

**Lemma 8.** *Let $f \in L^2(\rho_X)$ be such that $\|f - \eta\|_{L^2(\rho_X)} \leq \varepsilon$ for some $\varepsilon > 0$. The following is true :*

- *If $\eta$ satisfies the low-noise Assumption **(A1)**, we have for all $\delta > \varepsilon$ and $0 < \nu < \delta$ :*

$$\mathbb{P}(|f(X)| \leq \nu) \leq \frac{\varepsilon^2}{(\delta - \nu)^2} + C\delta^q$$

- *If $\eta$ satisfies the hard-margin Assumption **(A2)** with margin $\delta > 0$ and $\varepsilon < \delta$, we have for all $\nu < \delta$ :*

$$\mathbb{P}(|f(X)| \leq \nu) \leq \frac{\varepsilon^2}{(\delta - \nu)^2}$$

*Proof.*
- Assume that Assumption **(A1)** holds. Observe that for any $\delta > 0$

$$\mathbb{P}(|f(X)| \leq \nu) = \mathbb{P}(|f(X)| \leq \nu; |\eta(X)| > \delta) + \mathbb{P}(|f(X)| \leq \nu; |\eta(X)| \leq \delta)$$
$$\leq \mathbb{P}(|f(X)| \leq \nu; |\eta(X)| > \delta) + C\delta^q$$

Now note that on the event $|\eta(X)| > \delta$, we have by triangle inequality

$$|f(X) - \eta(X)| + |f(X)| \geq |\eta(X)| > \delta \implies |f(X) - \eta(X)| \geq \delta - |f(X)|$$

Finally, Chebyshev's inequality yields

$$\mathbb{P}(|f(X)| \leq \nu; |\eta(X)| > \delta) \leq \mathbb{P}(|f(X) - \eta(X)| \geq \delta - \nu)$$
$$\leq \frac{\|f - \eta\|_{L^2(\rho_X)}^2}{(\delta - \nu)^2}$$
$$\leq \frac{\varepsilon^2}{(\delta - \nu)^2},$$

this yields the claimed inequality.

- If we now assume that $\eta$ satisfies the hard-margin condition **(A2)**, we proceed similarly as in the previous case, with the only difference being that the term $\mathbb{P}(|f(X)| \leq \nu; |\eta(X)| \leq \delta)$ is now equal to zero. The rest of the argument carries through.

$\square$

The following lemma quantifies the approximation error of minimizers $\boldsymbol{\theta}_\lambda$ of the regularized population risk $\mathcal{R}_{\ell,\lambda}$ over $\mathcal{H}_{\mathcal{F},\mathbf{a},R}$ in terms of the approximation error of the parametric function class $\mathcal{H}_{\mathcal{F},\mathbf{a},R}$.

**Lemma 9.** *Let $\mathbf{a} \in \mathbb{N}^{L+1}$ be an architecture, and $R > 0$ a parameter bound such that*

$$\inf_{f \in \mathcal{H}_{\mathcal{F},\mathbf{a},R}} \|f - \eta\|_{L^2(\rho_X)} \leq \varepsilon$$

*for some constant $\varepsilon \geq 0$. Then, for any $\lambda \geq 0$, we have that any minimizer $\boldsymbol{\theta}_\lambda$ of the regularized population risk $\mathcal{R}_{\ell,\lambda}$ over $\mathcal{H}_{\mathcal{F},\mathbf{a},R}$ satisfies*

$$\|f(\cdot; \boldsymbol{\theta}_\lambda) - \eta\|_{L^2(\rho_X)} \leq \varepsilon + \sqrt{\lambda P(\mathbf{a}) R^p},$$

*where $P(\mathbf{a})$ denotes the number of parameters in the architecture $\mathbf{a}$.*

*Proof.* First note that for any $g \in L^2(\rho_X)$, we have

$$\mathcal{R}_\ell(g) := \mathbb{E}_{(x,y)\sim\rho} \left[ (g(x) - y)^2 \right]$$
$$= \mathbb{E}_{(x,y)\sim\rho} \left[ (g(x) - \eta(x))^2 \right] + \mathbb{E}_{(x,y)\sim\rho} \left[ (\eta(x) - y)^2 \right] + 2\mathbb{E}_{(x,y)\sim\rho} \left[ (g(x) - y)(\eta(x) - y) \right]$$
$$= \|g - \eta\|_{L^2(\rho_X)}^2 + C + 2\mathbb{E}_{(x,y)\sim\rho}\mathbb{E}\left[ (g(x) - \eta(x))(\eta(x) - y) \mid x \right]$$
$$= \|g - \eta\|_{L^2(\rho_X)}^2 + C + 2\mathbb{E}_{(x,y)\sim\rho} \left[ (g(x) - \eta(x))(\eta(x) - \mathbb{E}[y \mid x]) \right]$$
$$= \|g - \eta\|_{L^2(\rho_X)}^2 + C + 0$$

Where $C \equiv \mathbb{E}_{(x,y) \sim \rho} \left[ (\eta(x) - y)^2 \right] \geq 0$ is a constant which does not depend on $g$. This shows that minimizing $\mathcal{R}_\ell$ is equivalent to minimizing the $L^2(\rho_X)$ distance to $\eta$, and in particular for two square-integrable functions $f, g \in L^2(\rho_X)$, we have the identity

$$\mathcal{R}_\ell(f) - \mathcal{R}_\ell(g) = \|f - \eta\|_{L^2(\rho_X)}^2 - \|g - \eta\|_{L^2(\rho_X)}^2. \tag{30}$$

Now denote by $\boldsymbol{\theta}^*$ any minimizer of $\|f(\cdot; \boldsymbol{\theta}) - \eta\|_{L^2(\rho_X)}^2$ over $\mathcal{P}_{\mathbf{a},R}$. For any positive $\lambda$, we have

$$\begin{aligned}
\mathcal{R}_\ell(f(\cdot; \boldsymbol{\theta}_\lambda)) &= \mathcal{R}_{\ell,\lambda}(f(\cdot; \boldsymbol{\theta}_\lambda)) - \lambda |\boldsymbol{\theta}_\lambda|_p^p \\
&\leq \mathcal{R}_{\ell,\lambda}(f(\cdot; \boldsymbol{\theta}_\lambda)) \\
&\leq \mathcal{R}_{\ell,\lambda}(f(\cdot; \boldsymbol{\theta}^*)) \\
&= \mathcal{R}_\ell(f(\cdot; \boldsymbol{\theta}^*)) + \lambda |\boldsymbol{\theta}^*|_p^p \\
&\leq \mathcal{R}_\ell(f(\cdot; \boldsymbol{\theta}^*)) + \lambda P(\mathbf{a}) R^p
\end{aligned}$$

Where $P(\mathbf{a})$ is the number of parameters in the architecture $\mathbf{a}$. From the identity (30) above, we deduce that $\|f(\cdot; \boldsymbol{\theta}_\lambda) - \eta\|_{L^2(\rho_X)}^2$ differs from $\|f(\cdot; \boldsymbol{\theta}^*) - \eta\|_{L^2(\rho_X)}^2$ by at most $\lambda P(\mathbf{a}) R^p$. We thus find that

$$\begin{aligned}
\|f(\cdot; \boldsymbol{\theta}_\lambda) - \eta\|_{L^2(\rho_X)}^2 &\leq \|f(\cdot; \boldsymbol{\theta}^*) - \eta\|_{L^2(\rho_X)}^2 + \lambda P(\mathbf{a}) R^p \\
&\leq \|f(\cdot; \boldsymbol{\theta}^*) - \eta\|_{L^\infty(\rho_X)}^2 + \lambda P(\mathbf{a}) R^p \\
&\leq \varepsilon^2 + \lambda P(\mathbf{a}) R^p,
\end{aligned}$$

and we conclude the proof by using the subadditivity of $x \mapsto \sqrt{x}$. $\qquad \square$

The last result we will need is a large deviation type estimate on the probability that a minimizer $\widehat{\boldsymbol{\theta}}_\lambda$ of the empirical risk $\widehat{\mathcal{R}}_{\ell,\lambda}$ is far away from the argmin of $\mathcal{R}_{\ell,\lambda}$. Such estimate can be readily obtained by applying covering number based concentration bounds, which are a standard tool in Learning Theory literature (Györfi et al., 2002).

**Lemma 10.** *For any $\lambda \geq 0$, let $\widehat{\boldsymbol{\theta}}_\lambda \in \mathcal{P}_{\mathbf{a},R}$ be a minimum-norm solution of the $\lambda$-ERM problem* (9), *and denote by $\mathcal{R}_{\ell,\lambda}$ the regularized population risk* (10). *If Assumption (A4) holds, then for all $t > 0$, we have the estimate*

$$\mathbb{P}(\mathrm{dist}(\widehat{\boldsymbol{\theta}}_\lambda, \mathrm{argmin}\, \mathcal{R}_{\ell,\lambda}) \geq t) \leq 4 \, \mathbf{Cov}_\infty \left( \mathcal{H}, \frac{Kt^r}{24 \, \mathrm{Lip}_R(\mathcal{F})} \right) \exp \left( \frac{-nK^2 t^{2r}}{288} \right)$$

*Proof.* Observe the inclusion of events

$$\begin{aligned}
\mathrm{dist}(\widehat{\boldsymbol{\theta}}_\lambda, \mathrm{argmin}\, \mathcal{R}_{\ell,\lambda}) \geq t &\implies \mathcal{R}_{\ell,\lambda}(f(\cdot, \widehat{\boldsymbol{\theta}}_\lambda)) \geq \inf_{\boldsymbol{\theta} \in \mathcal{P}_{\mathbf{a},R}: \mathrm{dist}(\boldsymbol{\theta}, \mathrm{argmin}\, \mathcal{R}_{\ell,\lambda}) \geq t} \mathcal{R}_{\ell,\lambda}(f(\cdot, \boldsymbol{\theta})) \\
&\implies \mathcal{R}_{\ell,\lambda}(f(\cdot, \widehat{\boldsymbol{\theta}}_\lambda)) - \mathcal{R}_{\ell,\lambda}(f(\cdot, \boldsymbol{\theta}_\lambda)) \geq Kt^r \\
&\implies \mathcal{R}_{\ell,\lambda}(f(\cdot, \widehat{\boldsymbol{\theta}}_\lambda)) - \widehat{\mathcal{R}}_{\ell,\lambda}(f(\cdot, \widehat{\boldsymbol{\theta}}_\lambda)) \\
&\quad + \widehat{\mathcal{R}}_{\ell,\lambda}(f(\cdot, \boldsymbol{\theta}_\lambda)) - \mathcal{R}_{\ell,\lambda}(f(\cdot, \boldsymbol{\theta}_\lambda)) \geq Kt^r \\
&\implies \widehat{\mathcal{R}}_{\ell,\lambda}(f(\cdot, \boldsymbol{\theta}_\lambda)) - \mathcal{R}_{\ell,\lambda}(f(\cdot, \boldsymbol{\theta}_\lambda)) \geq Kt^r/2 \\
&\quad \text{OR} \quad \mathcal{R}_{\ell,\lambda}(f(\cdot, \widehat{\boldsymbol{\theta}}_\lambda)) - \widehat{\mathcal{R}}_{\ell,\lambda}(f(\cdot, \widehat{\boldsymbol{\theta}}_\lambda)) \geq Kt^r/2
\end{aligned}$$

Where we used Proposition 1 in the second line. Now set $\varepsilon := Kt^r/2$ and let

$$\{f(\cdot; \boldsymbol{\theta}_\varepsilon) : \boldsymbol{\theta}_\varepsilon \in \boldsymbol{\Theta}_\varepsilon\}$$

be a minimal size $\varepsilon/(12 \, \mathrm{Lip}(\mathcal{F}))$-cover of $\mathcal{H}_{\mathcal{F},\mathbf{a},R}$. By observing that the map

$$\varphi : \mathcal{P}_{\mathbf{a},R} \to \mathbb{R}, \quad \boldsymbol{\theta} \mapsto (f(x; \boldsymbol{\theta}) - y)^2$$

is $4 \operatorname{Lip}_R(\mathcal{F})$-Lipschitz continuous uniformly over $(x, y) \in \mathcal{X} \times \{-1, 1\}$, we get that for any $\boldsymbol{\theta} \in \mathcal{P}_{\mathbf{a}, R}$, and $\boldsymbol{\theta}_\varepsilon \in \boldsymbol{\Theta}_\varepsilon$ such that $|\boldsymbol{\theta} - \boldsymbol{\theta}_\varepsilon|_\infty \leq \varepsilon/(12 \operatorname{Lip}_R(\mathcal{F}))$:

$$
\begin{aligned}
|\widehat{\mathcal{R}}_{\ell,\lambda}(f(\cdot, \boldsymbol{\theta})) - \mathcal{R}_{\ell,\lambda}(f(\cdot, \boldsymbol{\theta}))| &= \left| \frac{1}{n} \sum_{i=1}^n (f(x_i; \boldsymbol{\theta}) - y_i)^2 - \mathbb{E}\left[(f(x; \boldsymbol{\theta}) - y)^2\right] \right| \\
&\leq \left| \frac{1}{n} \sum_{i=1}^n (f(x_i; \boldsymbol{\theta}_\varepsilon) - y_i)^2 - \frac{1}{n} \sum_{i=1}^n (f(x_i; \boldsymbol{\theta}) - y_i)^2 \right| \\
&\quad + \left| \mathbb{E}\left[(f(x; \boldsymbol{\theta}_\varepsilon) - y)^2\right] - \mathbb{E}\left[(f(x; \boldsymbol{\theta}) - y)^2\right] \right| \\
&\quad + \left| \frac{1}{n} \sum_{i=1}^n (f(x_i; \boldsymbol{\theta}_\varepsilon) - y_i)^2 - \mathbb{E}\left[(f(x; \boldsymbol{\theta}_\varepsilon) - y)^2\right] \right| \\
&\leq 2 \operatorname{Lip}(\varphi) |\boldsymbol{\theta} - \boldsymbol{\theta}_\varepsilon|_\infty + \left| \frac{1}{n} \sum_{i=1}^n (f(x_i; \boldsymbol{\theta}_\varepsilon) - y_i)^2 - \mathbb{E}\left[(f(x; \boldsymbol{\theta}_\varepsilon) - y)^2\right] \right| \\
&\leq \frac{2\varepsilon}{3} + \left| \frac{1}{n} \sum_{i=1}^n (f(x_i; \boldsymbol{\theta}_\varepsilon) - y_i)^2 - \mathbb{E}\left[(f(x; \boldsymbol{\theta}_\varepsilon) - y)^2\right] \right|
\end{aligned}
$$

After taking the supremum over $\boldsymbol{\theta} \in \mathcal{P}_{\mathbf{a}, R}$ in the above inequality, and observing that the $Z_i := (f(x_i; \boldsymbol{\theta}_\varepsilon) - y_i)^2$ are i.i.d. and taking value in $[0, 4]$ almost surely, we apply the union bound together with Hoeffding's inequality to find:

$$
\begin{aligned}
\mathbb{P}\left(\operatorname{dist}(\widehat{\boldsymbol{\theta}}_\lambda, \operatorname{argmin} \mathcal{R}_{\ell,\lambda}) \geq t\right) &\leq \mathbb{P}\left(\widehat{\mathcal{R}}_{\ell,\lambda}(f(\cdot, \boldsymbol{\theta}_\lambda)) - \mathcal{R}_{\ell,\lambda}(f(\cdot, \boldsymbol{\theta}_\lambda)) \geq Kt^r/2\right) \\
&\quad + \mathbb{P}\left(\mathcal{R}_{\ell,\lambda}(f(\cdot, \widehat{\boldsymbol{\theta}}_\lambda)) - \widehat{\mathcal{R}}_{\ell,\lambda}(f(\cdot, \widehat{\boldsymbol{\theta}}_\lambda)) \geq Kt^r/2\right) \\
&\leq 2\mathbb{P}\left(\sup_{\boldsymbol{\theta} \in \mathcal{P}_{\mathbf{a}, R}} |\mathcal{R}_{\ell,\lambda}(f(\cdot, \boldsymbol{\theta})) - \widehat{\mathcal{R}}_{\ell,\lambda}(f(\cdot, \boldsymbol{\theta}))| \geq Kt^r/2\right) \\
&= 2\mathbb{P}\left(\sup_{\boldsymbol{\theta} \in \mathcal{P}_{\mathbf{a}, R}} |\mathcal{R}_{\ell,\lambda}(f(\cdot, \boldsymbol{\theta})) - \widehat{\mathcal{R}}_{\ell,\lambda}(f(\cdot, \boldsymbol{\theta}))| \geq \varepsilon\right) \\
&\leq 2\mathbb{P}\left(\sup_{\boldsymbol{\theta}_\varepsilon \in \boldsymbol{\Theta}_\varepsilon} |\mathcal{R}_{\ell,\lambda}(f(\cdot, \boldsymbol{\theta}_\varepsilon)) - \widehat{\mathcal{R}}_{\ell,\lambda}(f(\cdot, \boldsymbol{\theta}_\varepsilon))| \geq \varepsilon/3\right) \\
&\leq 4 \mathbf{Cov}_\infty\left(\mathcal{H}, \frac{\varepsilon}{12 \operatorname{Lip}_R(\mathcal{F})}\right) \exp\left(\frac{-n\varepsilon^2}{72}\right).
\end{aligned}
$$

Finally, after substituting $\varepsilon \equiv Kt^r/2$, we find

$$
\mathbb{P}\left(\operatorname{dist}(\widehat{\boldsymbol{\theta}}_\lambda, \operatorname{argmin} \mathcal{R}_{\ell,\lambda}) \geq t\right) \leq 4 \mathbf{Cov}_\infty\left(\mathcal{H}, \frac{Kt^r}{24 \operatorname{Lip}_R(\mathcal{F})}\right) \exp\left(\frac{-nK^2 t^{2r}}{288}\right),
$$

as desired. $\qquad\square$

### 6.4.2   Proof of Theorem 2

We prove Theorem 2 under the low-noise Assumption **(A1)** only, as the case **(A2)** can be shown using the exact same argument.

To begin, we decompose the excess risk in two parts :

$$
\begin{aligned}
\mathcal{R}(\operatorname{sign} f(\cdot; \widehat{\boldsymbol{\theta}}_\lambda)) - \mathcal{R}^* &:= \mathcal{R}(\operatorname{sign} f(\cdot; \widehat{\boldsymbol{\theta}}_\lambda)) - \mathcal{R}(\operatorname{sign} \eta) \\
&= \mathcal{R}(\operatorname{sign} f(\cdot; \widehat{\boldsymbol{\theta}}_\lambda)) - \mathcal{R}(\operatorname{sign} f(\cdot; \boldsymbol{\theta}_\lambda)) \\
&\quad + \mathcal{R}(\operatorname{sign} f(\cdot; \boldsymbol{\theta}_\lambda)) - \mathcal{R}(\operatorname{sign} \eta)
\end{aligned}
$$

Where $\widehat{\boldsymbol{\theta}}_\lambda \in \mathcal{P}_{\mathbf{a},R}$ and $\boldsymbol{\theta}_\lambda \in \mathcal{P}_{\mathbf{a},2R}$ are respectively minimum-norm minimizers of the empirical and population risk (9), such that

$$|\widehat{\boldsymbol{\theta}}_\lambda - \boldsymbol{\theta}_\lambda|_\infty = \text{dist}(\widehat{\boldsymbol{\theta}}_\lambda, \underset{\mathcal{P}_{\mathbf{a},R}}{\text{argmin}} \, \mathcal{R}_{\ell,\lambda}).$$

Note that by Assumption (A3) and closedness of $\text{argmin}_{\mathcal{P}_{\mathbf{a},R}} \mathcal{R}_{\ell,\lambda}$, the above is always possible as long as the parameter bound $R$ has been chosen larger than $R_0 \cdot P(\mathbf{a})^{1/p}$, but the $\ell_\infty$ norm of $\boldsymbol{\theta}_\lambda$ can only be bounded by $2R$ instead of $R$.

Combining Lemma 7 and Lemma 9, we immediately get the bound on the first summand :

$$\mathcal{R}(\text{sign} f(\cdot; \boldsymbol{\theta}_\lambda)) - \mathcal{R}(\text{sign} \eta) \leq \|f(\cdot; \boldsymbol{\theta}_\lambda) - \eta\|_{L^2(\rho_X)} \leq \varepsilon_{\text{approx}} + \sqrt{\lambda P(\mathbf{a}) 2^p R^p}. \tag{31}$$

It only remains to bound the second summand. To that end, we apply Lemma 6, which yields :

$$\mathcal{R}(\text{sign} f(\cdot; \widehat{\boldsymbol{\theta}}_\lambda)) - \mathcal{R}(\text{sign} f(\cdot; \boldsymbol{\theta}_\lambda)) \leq \mathbb{P}\left\{\|f(\cdot; \widehat{\boldsymbol{\theta}}_\lambda) - f(\cdot; \boldsymbol{\theta}_\lambda)\|_{L^\infty(\rho_X)} \geq |f(X; \boldsymbol{\theta}_\lambda)|\right\}.$$

Now note that thanks to inequality (31), we can apply the "high-probability margin" property from Lemma 8 to get for all $\delta > \varepsilon_{\text{approx}}$, $\lambda < (\delta - \varepsilon_{\text{approx}})^2 (P(\mathbf{a}) 2^p R^p)^{-1}$, and $0 < \nu < \delta$:

$$\mathbb{P}\left\{\|f(\cdot; \widehat{\boldsymbol{\theta}}_\lambda) - f(\cdot; \boldsymbol{\theta}_\lambda)\|_{L^\infty(\rho_X)} \geq |f(X; \boldsymbol{\theta}_\lambda)|\right\} = \mathbb{P}\left\{\|f(\cdot; \widehat{\boldsymbol{\theta}}_\lambda) - f(\cdot; \boldsymbol{\theta}_\lambda)\|_{L^\infty(\rho_X)} \geq |f(X; \boldsymbol{\theta}_\lambda)|; |f(X; \boldsymbol{\theta}_\lambda)| > \nu\right\}$$

$$+ \mathbb{P}\left\{\|f(\cdot; \widehat{\boldsymbol{\theta}}_\lambda) - f(\cdot; \boldsymbol{\theta}_\lambda)\|_{L^\infty(\rho_X)} \geq |f(X; \boldsymbol{\theta}_\lambda)|; |f(X; \boldsymbol{\theta}_\lambda)| \leq \nu\right\}$$

$$\leq \mathbb{P}\left\{\|f(\cdot; \widehat{\boldsymbol{\theta}}_\lambda) - f(\cdot; \boldsymbol{\theta}_\lambda)\|_{L^\infty(\rho_X)} \geq \nu\right\}$$

$$+ (\delta - \nu)^{-2}\left(\varepsilon_{\text{approx}} + \sqrt{\lambda P(\mathbf{a}) 2^p R^p}\right)^2 + C\delta^q$$

We are now left with estimating the probability that $\|f(\cdot; \widehat{\boldsymbol{\theta}}_\lambda) - f(\cdot; \boldsymbol{\theta}_\lambda)\|_{L^\infty} \geq \nu$. By Lipschitzness of $\mathcal{F}$, we have

$$\mathbb{P}\left\{\|f(\cdot; \widehat{\boldsymbol{\theta}}_\lambda) - f(\cdot; \boldsymbol{\theta}_\lambda)\|_{L^\infty(\rho_X)} \geq \nu\right\} \leq \mathbb{P}\left(|\widehat{\boldsymbol{\theta}}_\lambda - \boldsymbol{\theta}_\lambda|_\infty \geq \nu / \text{Lip}_{2R}(\mathcal{F})\right)$$

$$= \mathbb{P}\left(\text{dist}(\widehat{\boldsymbol{\theta}}_\lambda, \text{argmin} \, \mathcal{R}_{\ell,\lambda}) \geq \nu / \text{Lip}_{2R}(\mathcal{F})\right)$$

$$\leq 4 \, \mathbf{Cov}_\infty\left(\mathcal{H}, \frac{K\nu^r}{24 \, \text{Lip}_{2R}(\mathcal{F})^{1+r}}\right) \exp\left(\frac{-nK^2\nu^{2r}}{288 \, \text{Lip}_{2R}(\mathcal{F})^{2r}}\right),$$

where the exponential inequality follows from Lemma 10. Combining all of these inequalities, we have thus shown that for all $\delta > \varepsilon_{\text{approx}}$, $\lambda < (\delta - \varepsilon_{\text{approx}})^2 (P(\mathbf{a}) 2^p R^p)^{-1}$, and $0 < \nu < \delta$:

$$\mathcal{R}(\text{sign} f(\cdot; \widehat{\boldsymbol{\theta}}_\lambda)) - \mathcal{R}^* \leq \varepsilon_{\text{approx}} + \sqrt{\lambda P(\mathbf{a}) 2^p R^p} + C\delta^q$$

$$+ (\delta - \nu)^{-2}\left(\varepsilon_{\text{approx}} + \sqrt{\lambda P(\mathbf{a}) 2^p R^p}\right)^2$$

$$+ 4 \, \mathbf{Cov}_\infty\left(\mathcal{NN}, \frac{K\nu^r}{24 \, \text{Lip}_{2R}(\mathcal{F})^{1+r}}\right) \exp\left(\frac{-nK^2\nu^{2r}}{288 \, \text{Lip}_{2R}(\mathcal{F})^{2r}}\right)$$

which concludes the proof of Theorem 2 under Assumption (A1). As was mentioned in the beginning, the proof under (A2) can be done with the exact same argument, the only difference being that the $C\delta^q$ term will disappear when applying Lemma 8.

### 6.4.3 Proof of Theorem 4

Start by fixing $\alpha > 0$, and recall the approximation error bound given by Assumption (A5), according to which

$$\inf_{f \in \mathcal{NN}(\mathbf{a}, W, L, R)} \|f - \eta\|_{L^2(\rho_X)} \leq C_3 W_0^{-2s/d},$$

for some architecture $\mathbf{a}$ such that $W(\mathbf{a}) = C_1 W_0 \log_2(W_0)$, $L(\mathbf{a}) = C_2 L_0 \log_2(L_0) + 2d$, where $W_0 \in \mathbb{N}_{\geq 2}$ is arbitrary, $C_1 = d(3s)^d d$, $C_3 = C_\eta s^d 8^s$, and $L_0$ and $C_2 \equiv C_2(s)$ are fixed.

By letting $W_0 = n^{\alpha d/2s} \times C_3^{d/2s}$, we deduce that there is a Neural Network architecture $\mathbf{a}_n$ with respective depth and width

$$L_n = C_2 L_0 \log_2(L_0) + 2d, \quad W_n = C_1 W_0 \log_2(W_0) = \tilde{\mathcal{O}}\left(n^{\alpha d/2s}\right),$$

where $\tilde{\mathcal{O}}$ hides logarithmic factors, such that

$$\inf_{f \in \mathcal{NN}(\mathbf{a}_n, W_n, L_n, R)} \|f - \eta\|_{L^2(\rho_X)} \leq n^{-\alpha}$$

Furthermore, the number of parameters in $\mathbf{a}_n$ is bounded as

$$P(\mathbf{a}_n) = \sum_{l=1}^{L_n} \mathbf{a}_n^{(l)} \mathbf{a}_n^{(l-1)} + \mathbf{a}_n^{(l)} \leq L_n(W_n^2 + W_n) = \tilde{\mathcal{O}}\left(n^{\alpha d/s}\right).$$

Similarly, recall the Lipschitz constant bound given by Lemma 3:

$$\sup_{\substack{\boldsymbol{\theta}, \boldsymbol{\theta}' \in \mathcal{P}_{\mathbf{a}_n, R} \\ \boldsymbol{\theta} \neq \boldsymbol{\theta}'}} \frac{\|\mathcal{F}_\sigma(\boldsymbol{\theta}) - \mathcal{F}_\sigma(\boldsymbol{\theta}')\|_{\mathcal{C}(\mathcal{X})}}{|\boldsymbol{\theta} - \boldsymbol{\theta}'|_\infty} \leq 2L_n^2 R^{L_n-1} W_n^{L_n},$$

and note that with $R \equiv R_n \equiv R_0 P(\mathbf{a_n})^{1/p}$, we have

$$R_n = \tilde{\mathcal{O}}\left(n^{\alpha d/ps}\right)$$

Putting these together we get

$$\sup_{\substack{\boldsymbol{\theta}, \boldsymbol{\theta}' \in \mathcal{P}_{\mathbf{a}_n, R_n} \\ \boldsymbol{\theta} \neq \boldsymbol{\theta}'}} \frac{\|\mathcal{F}_\sigma(\boldsymbol{\theta}) - \mathcal{F}_\sigma(\boldsymbol{\theta}')\|_{\mathcal{C}(\mathcal{X})}}{|\boldsymbol{\theta} - \boldsymbol{\theta}'|_\infty} \leq 2L_n^2 \tilde{\mathcal{O}}\left(\left(n^{\alpha d/ps}\right)^{L_n-1}\left(n^{\alpha d/s}\right)^{L_n}\right)$$

$$\leq \tilde{\mathcal{O}}\left(n^{\frac{\alpha d}{s} \cdot \left[\frac{L_n-1}{p} + L_n\right]}\right)$$

$$= \tilde{\mathcal{O}}\left(n^{\frac{\alpha d}{ps} \cdot [L_n(1+p)-1]}\right)$$

$$= \tilde{\mathcal{O}}\left(n^{\frac{\alpha d}{ps} \cdot \left[\left(C_2(s)L_0 \log_2 L_0 + 2d\right)(1+p)-1\right]}\right),$$

where all the logarithmic factors and terms which do not depend on $n$ are hidden in the $\tilde{\mathcal{O}}$.

After noting that $\mathrm{Lip}_{2R}(\mathcal{F}_{NN}) \leq 2^{L-1} \mathrm{Lip}_R(\mathcal{F}_{NN})$, we are left with bounding the quantity

$$\mathbf{Cov}_\infty\left(\mathcal{NN}, \frac{K(2^{1-L_n}\nu)^r}{24 \mathrm{Lip}_{R_n}(\mathcal{F}_{NN})^{1+r}}\right),$$

which by Lemma 4, we know is bounded by

$$\left(1 + \frac{48 R_n \mathrm{Lip}_{R_n}(\mathcal{F}_{NN})^{2+r}}{K(2^{1-L_n}\nu)^r}\right)^{P(\mathbf{a}_n)} \leq \left(\frac{49 R_n \mathrm{Lip}_{R_n}(\mathcal{F}_{NN})^{2+r}}{K(2^{1-L_n}\nu)^r}\right)^{P(\mathbf{a}_n)}.$$

Using the bounds on $R_n$ and $\mathrm{Lip}_{R_n}(\mathcal{F}_{NN})$ above, we thus find that

$$\frac{49 R_n \mathrm{Lip}_{R_n}(\mathcal{F}_{NN})^{2+r}}{K \cdot 2^{r(1-L_n)}} \leq \tilde{\mathcal{O}}\left(n^{\frac{\alpha d}{ps}} \cdot n^{\frac{(2+r)\alpha d}{ps} \cdot \left[\left(C_2 L_0 \log_2 L_0 + 2d\right)(1+p)-1\right]}\right).$$

The above quantity being polynomial in $n$, we thus find that the covering number grows as the exponential of $P(\mathbf{a}_n)$, up to a multiplicative logarithmic factor:

$$\log\left[\mathbf{Cov}_\infty\left(\mathcal{NN}, \frac{K(2^{1-L}\nu)^r}{24\operatorname{Lip}_{R_n}(\mathcal{F}_{NN})^{1+r}}\right)\right] = \mathcal{O}\left(P(\mathbf{a}_n)\log(n^\beta \cdot \nu^{-r})\right),$$

where

$$\beta \equiv \frac{\alpha d}{ps}\left(1 + (2+r)\cdot\left[\left(C_2(s)L_0\log_2 L_0 + 2d\right)(1+p) - 1\right]\right)$$

To conclude the proof for the case **(A1)**, we let $\varepsilon_{\text{approx}} \equiv n^{-\frac{\alpha}{r}}$, $\delta \equiv 2n^{-\frac{\alpha}{2r}}$ and $\nu \equiv n^{-\frac{\alpha}{2r}}$: observe that by picking $\lambda$ such that

$$0 \leq \lambda \leq \varepsilon_{\text{approx}}^2 (P(\mathbf{a}_n)2^p R_n^p)^{-1} = \mathcal{O}\left(n^{-\frac{2\alpha(s+d)}{rs}}\right),$$

we have $\lambda < (\delta - \varepsilon_{\text{approx}})^2 (P(\mathbf{a}_n)2^p R_n^p)^{-1}$ and

$$\varepsilon_{\text{approx}} + \sqrt{\lambda P(\mathbf{a}_n)2^p R^p} \leq 2\varepsilon_{\text{approx}}.$$

We are thus allowed to apply Theorem 2 with these values of $\lambda$, which yields the excess risk bound:

$$\mathcal{R}\left(\operatorname{sign} f(\cdot; \widehat{\boldsymbol{\theta}}_\lambda)\right) - \mathcal{R}_* \leq 2n^{-\frac{\alpha}{r}} + 2Cn^{-\frac{\alpha q}{2r}} + 4n^{-\frac{\alpha}{r}}$$
$$+ 4\exp\left(-A_1 n^{1-A_2} + n^{\frac{\alpha d}{s}}\log(\gamma n^{(\alpha+2\beta)/2})\right),$$

where

$$A_1 \equiv \frac{K^2 2^{2r(1-L_n)}}{288}, \quad A_2 \equiv \alpha\left(1 + \frac{rd}{ps}\cdot\left(\left[C_2(s)L_0\log_2(L_0) + 2d\right]\cdot(2+2p) - 2\right)\right),$$

and $\gamma > 0$ is a quantity which does not depend on $n$. Hence we see that the exponential term converges to zero as $n \to \infty$ if $1 - A_2 > 0$ and $1 - A_2 > \alpha d/s$, or equivalently if $1 - A_2 > \alpha d/s$, which is equivalent to the following inequality for $\alpha$:

$$\alpha < \left(1 + \frac{rd}{ps}\cdot\left(\left[C_2(s)L_0\log_2(L_0) + 2d\right]\cdot(2+2p) + p - 2\right)\right)^{-1}.$$

This concludes the proof under Assumption **(A1)**. The proof under Assumption **(A2)** with margin $\delta > 0$ is very similar: we now pick $\varepsilon_{\text{approx}} \equiv n^{-\frac{\alpha}{r}}$, $\nu \equiv \delta/2$, and

$$0 \leq \lambda \leq \varepsilon_{\text{approx}}^2 (P(\mathbf{a}_n)2^p R_n^p)^{-1} = \mathcal{O}\left(n^{-\frac{2\alpha(s+d)}{rs}}\right),$$

such that Theorem 2 can be applied, to yield for all $n \geq \left\lceil(\delta/2)^{-r/\alpha}\right\rceil$:

$$\mathcal{R}\left(\operatorname{sign} f(\cdot; \widehat{\boldsymbol{\theta}}_\lambda)\right) - \mathcal{R}_* \leq 2n^{-\frac{\alpha}{r}} + 16n^{-\frac{\alpha}{r}} + 4\exp\left(-A_1 n^{1-A_2} + n^{\alpha d/s}\log(\gamma n^\beta(\delta/2)^{-r})\right),$$

where

$$A_1 \equiv \frac{K^2(\delta 2^{-L_n})^{2r}}{288}, \quad A_2 \equiv \alpha\frac{rd}{sp}\left(\left[C_2(s)L_0\log_2 L_0 + 2d\right]\cdot(2+2p) - 2\right).$$

Hence, we see as before that in this case the term $4\exp\left(-A_1 n^{1-A_2} + n^{\alpha d/s}\log(\gamma n^\beta(\delta/2)^{-r})\right)$ vanishes exponentially fast as $n \to \infty$ if $1 - A_2 > \alpha d/s$, which equivalently means that $\alpha$ needs to satisfy the following inequality

$$\alpha < \frac{sp}{rd\left(\left[C_2(s)L_0\log_2 L_0 + 2d\right]\cdot(2+2p) + p - 2\right)}.$$

### 6.4.4 Proof of Corollary 1 and Theorem 8

*Proof of Corollary 1.* Since **(A3)** holds, First, we have by Theorem 5 that the hypothesis space $\mathcal{NN}^\sigma(\mathbf{a}, 3W, 2L, \infty)$ of $\sigma$-activated neural networks achieves the same approximation error as the space $\mathcal{NN}^{\mathrm{ReLU}}(\mathbf{a}, W, L, \infty)$, which by **(A5)** yields

$$\inf_{f \in \mathcal{NN}^\sigma(\mathbf{a}, 3W, 2L, \infty)} \|f - \eta\|_{L^2(\rho_X)} \le C_3 W_0^{-2s/d},$$

where $W, L, C_3$ and $W_0$ are as in **(A5)**. Secondly, by Assumption **(A3)** and Lemma 2, the minimizers of the $\lambda$-ERM objective (3) are uniformly contained in a $\ell_\infty$-ball of radius $R_0'(P(\mathbf{a}))^{1/p}$, where the overhead in the number of parameters is absorbed by $R_0$. Since Lemma 3 gives, up to absolute constants, the same Lipschitz bound for $\sigma$ neural networks as for ReLU networks, we can carry the argument that we used for the proof of Theorem 4 to $\sigma$-activated neural networks and thus establish the Corollary. □

*Proof of Theorem 8.* Follows from the exact same argument as the proof of Theorem 4. □

## 6.5 Characterization of Assumption (A5)

In this section, we give several non-trivial examples for which Assumption **(A5)** is satisfied. Establishing a concrete characterization of the pairs $(\eta, \rho_X)$ for which **(A5)** holds being challenging, we will restrict our attention to piecewise linear functions only, which is justified by the following result:

**Proposition 2** (Adapted from (Chen et al., 2022), Theorem 1). *For any integer $N \in \mathbb{N}$, and any continuous piece-wise linear (CPWL) function $f : \mathbb{R}^d \to \mathbb{R}$, there exists a ReLU neural network with width at most $c_1 N^2$ and depth at most $c_2 \log N$ which represents $f$ exactly. Here $c_1, c_2 > 0$ are universal constants.*

Proposition 2, shows that in many cases, a rate of approximation for CPWL functions can translate to comparable rates for ReLU neural networks. We will thus in the following exhibit examples of pairs $(\eta, \rho_X)$ for which CPWL functions can achieve faster rates of approximation in the $L^2(\rho_X)$ sense.

### 6.5.1 Ahlfors regular distribution and local flatness

We now give the first example for which faster $L^2(\rho_X)$ can be achieved: when $\rho_X$ is an *Ahlfors regular* distribution[1] which concentrates its mass on regions where $\eta$ is very flat. More specifically:

**Lemma 11.** *Let $\rho_X$ be the marginal data distribution on $\mathcal{X} = [0,1]^d$ with support $S := \mathrm{supp}(\rho_X) \subset \mathcal{X}$. Assume that there exist $k \in (0, d)$ and constants $c_\rho, C_\rho, r_0 > 0$ such that*

$$c_\rho r^k \le \rho_X(B(x, r)) \le C_\rho r^k \quad \text{for all } x \in S, \ 0 < r \le r_0. \tag{32}$$

*Note that $\dim_{\mathcal{H}}(S) = k < d$, and $\rho_X$ is singular with respect to $d$-dimensional Lebesgue measure on $\mathcal{X}$. Let $\beta \in \mathbb{N}$, and assume that there exist an open set $U \subset \mathbb{R}^d$ with $S \subset U$ and a function $\tilde\eta \in C^\beta(U)$ such that $\tilde\eta = \eta$ on $S$ and $D^\alpha \tilde\eta(x) = 0$ for all $x \in S$ and all multi-indices $\alpha$ with $1 \le |\alpha| \le \beta - 1$. Then there exist $K_1, K_2 > 0$, depending only on $d, k, \beta, c_\rho, C_\rho, r_0$ and the $C^\beta$-norm of $\tilde\eta$, such that for every integer $N \ge 1$ there exists a CPWL function $g_N : \mathcal{X} \to \mathbb{R}$ with at most $K_1 N$ pieces and approximation error*

$$\|\eta - g_N\|_{L^2(\rho_X)} \le K_2 N^{-\beta/k}.$$

*In particular, Assumption **(A5)** holds with approximation rate $s$ as soon as $\beta/k > 4s/d$.*

*Proof.* In what follows, we work with the extension $\tilde\eta$ on $U$ and drop the tilde from the notation for convenience. First note that by Taylor's theorem at points $x \in S$, together with the vanishing assumption on derivatives of order $1, \dots, \beta - 1$ and the boundedness of the order-$\beta$ derivatives give

$$|\eta(y) - \eta(x)| \le C |y - x|^\beta \quad \text{for all } x \in S, \ y \in U \tag{33}$$

---

[1] We refer the reader to (Mattila, 1999) for a comprehensive overview of Ahlfors regularity.

for some constant $C$. Now, for $x \in S$ and $0 < r \le r_0$ let

$$\alpha(x, r) := \inf_{a \text{ affine}} \left( \rho_X(B(x, r))^{-1} \int_{B(x, r)} |\eta(y) - a(y)|^2 \, d\rho_X(y) \right)^{1/2} \tag{34}$$

be the local best affine approximation error. Using the constant affine function $a : y \mapsto \eta(x)$ in (34), and using inequality (33) we obtain

$$\alpha(x, r) \le C \, r^\beta \quad \text{for all } x \in S, \ 0 < r \le r_0. \tag{35}$$

Now fix $N \ge 1$, and choose $r \sim N^{-1/k}$ small enough so that $r \le r_0$. Let $\{x_j\}_{j=1}^J \subset S$ be a family of $r$-separated points of maximal cardinality, and set $B_j := B(x_j, r)$. Then $S \subset \bigcup_j B_j$ and the balls $B(x_j, r/2)$ are disjoint. By (32) and the fact that $\rho_X$ is a probability measure, we get

$$1 \ge \sum_{j=1}^J \rho_X\big(B(x_j, r/2)\big) \ge J \, c_\rho (r/2)^k,$$

so $J \lesssim r^{-k} \sim N$. Similarly, we can show that the number of balls $(B_j)_j$ to which an arbitrary $x \in S$ can belong is bounded above by a constant depending only on $k, c_\rho, C_\rho$: indeed, if we set $J(x) := \{j : x \in B_j\}$, then the disjointness of the balls $B(x_j, r/2)$ and (32) give

$$|J(x)| \, c_\rho (r/2)^k \le \sum_{j \in J(x)} \rho_X(B(x_j, r/2)) \le \rho_X(B(x, 2r)) \le C_\rho (2r)^k.$$

Hence $|J(x)| \le (C_\rho / c_\rho) \, 4^k$ for all $x$. Now, choose a measurable partition $(P_j)_{j=1}^J$ of $S$ with $P_j \subset B_j$ for each $j$, and add $\mathcal{X} \setminus S$ as an extra cell to cover $\mathcal{X}$. For each $j$ pick an affine function $a_j$ such that

$$\left( \rho_X(B_j)^{-1} \int_{B_j} |\eta - a_j|^2 \, d\rho_X \right)^{1/2} \le 2 \, \alpha(x_j, r).$$

Define $g_N(x) := a_j(x)$ for $x \in P_j$ and define $g_N$ arbitrarily on $\mathcal{X} \setminus S$. By construction, $g_N$ is affine on each $P_j$, hence CPWL with at most $J \lesssim N$ pieces. Furthermore, since $P_j \subset B_j$, we have

$$\int_{P_j} |\eta - g_N|^2 \, d\rho_X \le \int_{B_j} |\eta - a_j|^2 \, d\rho_X \lesssim \alpha(x_j, r)^2 \, \rho_X(B_j) \lesssim r^{2\beta} \, \rho_X(B_j),$$

where the last inequality follows from (35). Summing over $j$ and using the boundedness of the overlap $J(x)$ gives

$$\|\eta - g_N\|_{L^2(\rho_X)}^2 = \int_{\mathcal{X}} |\eta - g_N|^2 \, d\rho_X \lesssim r^{2\beta}.$$

Since $r \sim N^{-1/k}$, this yields $\|\eta - g_N\|_{L^2(\rho_X)} \lesssim N^{-\beta/k}$, as claimed. $\qquad \square$

### 6.5.2 Discrete distribution

We now give a second, simpler example for which the faster approximation rates in Assumption (A5) can be achieved. We begin by formally defining *discrete distributions* on the cube $\mathcal{X} = [0, 1]^d$.

**Definition 9** (Discrete distribution). *For any $x \in \mathcal{X}$, let $\delta_x$ denote the Dirac probability measure at $x$, which satisfies $\delta_x(A) = \mathbb{1}_A(x)$ for any measurable $A \subseteq \mathcal{X}$. We say that the marginal data distribution $\rho_X$ is discrete if there exist a sequence $x_1, x_2, \ldots \subseteq \mathcal{X}$ of pairwise distinct points, and a sequence $\lambda_1, \lambda_2, \ldots$ of non-negative real numbers, such that $\sum_{i \ge 1} \lambda_i = 1$ and*

$$\rho_X := \sum_{i \ge 1} \lambda_i \delta_{x_i}.$$

**Lemma 12.** *Assume that $\rho_X = \sum_{i \geq 1} \lambda_i \delta_{x_i}$ is a discrete distribution, and fix $s > 0$. If the coefficients $(\lambda_i)_{i \geq 1}$ are such that*

$$\sum_{i \geq n+1} \lambda_i \leq C_\rho n^{-4s/d} \quad \text{for all } n \geq 1,$$

*where $C_\rho$ is a positive universal constant, then Assumption (A5) is satisfied with approximation rate $s$ and constant $C_2(s) = 0$.*

*Proof of Lemma 12.* We will prove the lemma by constructing a sequence $(\Psi_n)_{n \geq 1}$ of DNN with 2 hidden layers and width $O(n)$ which, for every $n \in \mathbb{N}$, interpolate the Bayes regression function $\eta$ at every $x \in \{x_1, \ldots, x_n\}$.

To that end, fix $n \in \mathbb{N}$: since $x_1, \ldots, x_n$ are pairwise distinct, the set $\mathbb{R}^d \setminus \cup_{i \neq j}(x_i - x_j)^\perp$ is not empty, and we can find a direction $v \in \mathbb{R}^d$ such that $v \cdot x_1, \ldots, v \cdot x_n$ are pairwise distinct. Now let $b = 2\max_{1 \leq i \leq n} |v \cdot x_i|$, and note that the map $NN_n : x \mapsto \text{ReLU}(v \cdot x + b)$ maps each $x_i$ to $v \cdot x_i + b$.

Given the $n$ real numbers $v \cdot x_1 + b < \ldots < v \cdot x_n + b$ (reindexing as necessary), one can construct a continuous piecewise linear map $PL_n : \mathbb{R} \to \mathbb{R}$ with breakpoints $(v \cdot x_1 + b, \eta(x_1)), \ldots, (v \cdot x_n + b, \eta(x_n))$. Such a map $PL_n$ can easily be realized by a shallow ReLU network of width $n$ (see e.g. (Arora et al., 2018) for an explicit construction). Each $\Psi_n = PL_n \circ NN_n$ is thus realized by a DNN with 2 hidden layers and width equal to $\max(d, n)$. Furthermore we have $|\Psi_n(x)| \leq \|\eta(x)\|_{L^\infty(\mathcal{X})}$ for all $x \in [0,1]^d$, and by construction, we have

$$\|\Psi_n - \eta\|^2_{L^2(\rho_X)} = \sum_{i \geq n+1} \lambda_i (\Psi_n(x_i) - \eta(x_i))^2 \leq 4\|\eta\|^2_{L^\infty(\mathcal{X})} \sum_{i \geq n+1} \lambda_i.$$

Lastly, note that for any depth $L \geq 2$, the identity mapping $Id : x \in \mathbb{R}^d \mapsto x$ can be realized by the following parameter vector

$$\boldsymbol{\theta} = \left( \left( \begin{pmatrix} I_d \\ -I_d \end{pmatrix}, 0 \right), (I_{2d}, 0), \ldots, (I_{2d}, 0), ((I_d - I_d), 0) \right),$$

where $I_d$ is the $d \times d$ identity matrix, and the tuple $(I_{2d}, 0)$ is repeated $L - 2$ times. Note that it is easy to modify the above architecture so that it yields a representation of the identity mapping with same depth and arbitrary width $W \geq 2d$. Hence, by "padding" the width and depth of $\Psi_n$ as necessary, we see that Assumption (A5) is indeed satisfied with rate $s$ and constant $C_2(s) = 0$ whenever $\sum_{i \geq n+1} \lambda_i \leq C_\rho n^{-4s/d}$, as claimed. □

### 6.5.3 Quasi-discrete distribution with adaptivity

Slightly generalizing the discrete example from above, we propose to consider the following:

**Definition 10** (Quasi-discrete distribution). *We say that the marginal data distribution $\rho_X$ is quasi-discrete if there exist a sequence $E_1, E_2$, of measurable and pairwise disjoint subsets of $\mathcal{X}$, and a sequence $\lambda_1, \lambda_2, \ldots$ of non-negative real numbers, such that $\sum_{i \geq 1} \lambda_i = 1$ and*

$$\rho_X := \sum_{i \geq 1} \lambda_i \mathbb{1}_{E_i}.$$

*If the Bayes regression function $\eta$ is such that, for all $i \geq 1$, $\eta$ can be exactly represented on $E_i$ by a CPWL function whose number of pieces does not depend on $i$, then we say that $\rho_X$ is* adapted *to $\eta$.*

In the same vein as the Ahlfors distribution example discussed earlier, the notion of "adaptivity" captures the idea that $\rho_X$ concentrates its mass only where $\eta$ is not just flat, but actually (piecewise) linear. Trivially, we get the following Lemma:

**Lemma 13.** *Assume that $\rho_X = \sum_{i \geq 1} \lambda_i \mathbb{1}_{E_i}$ is a quasi-discrete distribution adapted to $\eta$, and fix $s > 0$. If the coefficients $(\lambda_i)_{i \geq 1}$ are such that*

$$\sum_{i \geq n+1} \lambda_i \leq C_\rho n^{-4s/d} \quad \text{for all } n \geq 1,$$

*where $C_\rho$ is a positive universal constant, then Assumption (A5) is satisfied with approximation rate $s$ and constant $C_2(s) = 0$.*

### 6.5.4 Distribution with fractal support

The last example we give of pairs $(\rho_X, \eta)$ for which **(A5)** holds is that of $\rho_X$ having fractal support of sufficiently small dimension, with $\eta$ being an arbitrary $\mathcal{C}^2$ function. In contrast with the previous examples where $\rho_X$ was concentrating its mass on the regions where $\eta$ looks "almost linear", here we don't need to assume such nice regions exist, and the low dimensionality of $\rho_X$'s support alone ensures the result. We will be using the following notion of fractal dimension, which can be found, e.g., in (Mattila, 1999):

**Definition 11** (Upper Minkowski dimension). *For $A$ a non-empty bounded subset of $\mathbb{R}^d$, and $\varepsilon > 0$, denote by $N(A, \varepsilon)$ the smallest number of cubes of sidelength $\varepsilon$ needed to cover $A$. The upper Minkowski dimension of $A$ is then defined as*

$$\overline{\dim}_M A := \inf \left\{ t : \limsup_{\varepsilon \downarrow 0} N(A, \varepsilon) \varepsilon^t = 0 \right\}.$$

In words, the upper Minkowski dimension of $A$ is the smallest exponent $d_M$ such that $N(A, \varepsilon) \leq C \varepsilon^{-d_M}$. We can now state our result:

**Lemma 14.** *Denote by $\rho_X$ the marginal data distribution, and by $S := supp(\rho_X) \subset \mathcal{X}$ its support. Let $d_\rho := \overline{\dim}_M S$ denote the upper Minkowski dimension of $S$. If $\eta \in \mathcal{C}^2(\mathcal{X})$, then for any $\epsilon > 0$, there exist constants $K_1, K_2 > 0$, which may depend on $\|\eta\|_{\mathcal{C}^2(\mathcal{X})}, S, d_\rho, d$ and $\epsilon$ only, such that for every integer $N \geq 1$ there exists a CPWL function $g_N : \mathcal{X} \to \mathbb{R}$ with at most $K_1 N$ pieces and approximation error satisfying*

$$\|\eta - g_N\|_{L^2(\rho_X)} \leq K_2 \, N^{-2/(d_\rho + \epsilon)}.$$

*In particular, Assumption **(A5)** holds with approximation rate $s$ as soon as $2/d_\rho > 4s/d$.*

*Proof.* Let $S = \text{supp}(\rho_X) \subset \mathcal{X}$ and $d_\rho = \overline{\dim}_M S$. By Definition 11, for every $t > d_\rho$ there exist constants $C_t > 0$ and $\varepsilon_0 > 0$ such that $N(S, \varepsilon) \leq C_t \varepsilon^{-t}$ for all $0 < \varepsilon \leq \varepsilon_0$, where $N(S, \varepsilon)$ is the minimal number of cubes of sidelength $\varepsilon$ needed to cover $S$.

Fix $t > d_\rho$ and $N \geq 1$ and set $\varepsilon_N := (N/C_t)^{-1/t}$. For all $N$ large enough we have $\varepsilon_N \leq \varepsilon_0$, hence there exist cubes $Q_1, \ldots, Q_M$ of sidelength $\varepsilon_N$ with $M \leq C_t \varepsilon_N^{-t} \leq N$ and $S \subset \bigcup_{j=1}^M Q_j$. For the finitely many smaller $N$ we can enlarge the constants $K_1, K_2$ at the end of the proof.

For each $j$ choose $x_j \in Q_j \cap S$ and let $p_j$ be the first order Taylor polynomial of $\eta$ at $x_j$. Since $\eta \in \mathcal{C}^2(\mathcal{X})$ and $\mathcal{X}$ is compact, there exists $C_\eta > 0$, depending only on $\|\eta\|_{\mathcal{C}^2(\mathcal{X})}$ and $d$, such that

$$|\eta(x) - p_j(x)| \leq C_\eta \, \varepsilon_N^2 \quad \text{for all } x \in Q_j \cap S \text{ and all } j.$$

Define $g_N : \mathcal{X} \to \mathbb{R}$ by $g_N(x) = p_j(x)$ if $x \in Q_j$ for some $j$ and $g_N(x) = 0$ otherwise. Then $g_N$ is CPWL with at most $M + 1 \leq 2N =: K_1 N$ pieces. Since $\rho_X$ is supported on $S$, we have $|\eta(x) - g_N(x)| \leq C_\eta \varepsilon_N^2$ for $\rho_X$-almost every $x$, hence

$$\|\eta - g_N\|_{L^2(\rho_X)} \leq C_\eta \varepsilon_N^2 = C_\eta (N/C_t)^{-2/t} = K_2(t) \, N^{-2/t}$$

for a constant $K_2(t) > 0$ depending only on $t$, $S$, $d$ and $\|\eta\|_{\mathcal{C}^2(\mathcal{X})}$. This concludes the proof. $\square$

### 6.6 Lower bounds

### 6.6.1 Preliminary results

The main tool we will need to obtain our minimax lower bounds is Assouad's lemma. Before stating the lemma, we define the *probability hypercube*, following notation from (Audibert, 2004).

**Definition 12** (Probability Hypercube). *Let $m \in \mathbb{N}, w \in (0, 1], b \in (0, 1]$, and $b' \in (0, 1]$. A $(m, w, b, b')$-hypercube of probability distributions is a family*

$$\{\mathbb{P}_{\vec{\sigma}} \mid \vec{\sigma} = (\sigma_1, \ldots, \sigma_m) \in \{-1, 1\}^m\}$$

*of $2^m$ probability distributions on $\mathcal{X} \times \{-1, 1\}$ having the same first marginal:*

$$\mathbb{P}_{\vec{\sigma}}(dX) = \mathbb{P}_{(+1,\dots,+1)}(dX) =: \mu \quad \text{for all } \vec{\sigma} \in \{-1, 1\}^m,$$

*and such that there exists a partition $\mathcal{X}_0, \dots, \mathcal{X}_m$ of the unit cube $\mathcal{X}$ satisfying*

- *for any $j \in \{1, \dots, m\}$, we have $\mu(\mathcal{X}_j) = w$*

- *for any $j \in \{0, \dots, m\}$, and any $x \in \mathcal{X}_j$, we have*

$$\mathbb{E}_{\vec{\sigma}}[Y \mid X = x] = \sigma_j \, \xi(x),$$

*where $\xi : \mathcal{X} \to [0, 1]$ is such that for any $j \in \{1, \dots, m\}$,*

$$\begin{cases} wb = \sqrt{w^2 - \left( \mathbb{E}_{X \sim \mu} \left[ \sqrt{1 - \xi^2(X)} \mathbb{1}_{\mathcal{X}_j} \right] \right)^2}, \\ wb' = \mathbb{E}_{X \sim \mu} \left[ \xi(X) \mathbb{1}_{\mathcal{X}_j} \right]. \end{cases}$$

We are now ready to state Assouad's lemma, in a version adapted to the setting in which the index set is given by $\mathcal{Y} = \{-1, 1\}$:

**Lemma 15** (Adapted from Lemma 5.1 in (Audibert, 2004)). *If a set $\mathcal{P}$ of probability distributions contains a $(m, w, b, b')$-hypercube, then for any measurable estimator*

$$\hat{c} : (\mathcal{X} \times \{-1, 1\})^N \to \mathcal{M}(\mathcal{X}, \{-1, 1\}),$$

*we have*

$$\sup_{\mathbb{P} \in \mathcal{P}} \left\{ \mathbb{E}_{\mathbb{P} \otimes N} \left[ \mathcal{R}_{\mathbb{P}}(\hat{c}) \right] - \mathcal{R}_{\mathbb{P}}^* \right\} \geq \frac{1 - b\sqrt{Nw}}{2} \, m \, w \, b',$$

*where $\mathcal{R}_{\mathbb{P}}(f) = \mathbb{P}\{f(X) \neq Y\}$ and $\mathcal{R}_{\mathbb{P}}^*$ is the Bayes optimal risk under $\mathbb{P}$.*

Although the original result in (Audibert, 2004) only proves an analogous version of Lemma 15 for the case $\mathcal{Y} = \{0, 1\}$, one can readily check that Lemma 15 is a direct consequence of applying the transformation $t \mapsto (t + 1)/2$ to the labels.

### 6.6.2 Proof of Theorem 6

We now prove the minimax lower bound. Note that our argument mostly follows the approach taken in (Audibert and Tsybakov, 2007), with suitable adjustments made as needed. We first prove the result for the low noise condition **(A1)**. The lower bound for the hard margin condition **(A2)** follows from a straightforward modification of the argument, which we explain at the end.

For an integer $h \geq 1$, we partition the unit cube $\mathcal{X} = [0, 1]^d$ as follows: define the regular grid $G_h$ as

$$G_h := \left\{ \left( \frac{2k_1 + 1}{2h}, \dots, \frac{2k_d + 1}{2h} \right) : k_i \in \{0, \dots, h - 1\}, i = 1, \dots, d \right\},$$

and for $x \in \mathcal{X}$ let $n_h(x) \in G_h$ be the minimum-norm element of $G_h$ closest to $x$ in Euclidean norm, so that $n_h : \mathcal{X} \to G_h$ is a well-defined (single-valued) function. We then define $\mathcal{X}'_1, \dots \mathcal{X}'_{h^d}$ as the canonical partition of $\mathcal{X}$ induced by $n_h$. That is, $x, y \in \mathcal{X}$ belong to the same subset $\mathcal{X}_i$ if and only if $n_h(x) = n_h(y)$. Now fix an integer $m \geq 1$, and for $1 \leq i \leq m$, define $\mathcal{X}_i := \mathcal{X}'_i$, and let $\mathcal{X}_0 := [0, 1]^d \setminus \cup_{1 \leq i \leq m} \mathcal{X}_i$, so that $\mathcal{X}_0, \dots, \mathcal{X}_m$ is a partition of $\mathcal{X}$ as well.

Let $u : \mathbb{R}^+ \to \mathbb{R}^+$ be the continuous piecewise linear function defined by $u(x) = 1$ for $x \in [0, 1/4]$, $u(x) = -4x + 2$ for $x \in [1/4, 1/2]$, and $u(x) = 0$ for $x \in [1/2, \infty)$, and define $\phi : \mathcal{X} \to \mathbb{R}^+$ by $\phi(x) = u(|x|_2)$, where $|\cdot|$ denotes the $\ell_2$ (Euclidean) norm.

We now define the hypercube $\mathcal{H} = \{\mathbb{P}_{\vec{\sigma}} \mid \vec{\sigma} \in \{-1, 1\}^m\}$ of probability distributions on $\mathcal{X} \times \{-1, 1\}$. For all $\vec{\sigma}$, we set the marginal distribution $\mathbb{P}_{\vec{\sigma}}(dX) =: \mu$ on $\mathcal{X}$ as a discrete distribution (as per Definition 9) defined in the following way: denote $z_1, \ldots, z_{h^d}$ the centers of the grid $G_h$, and for $1 \leq i \leq m$, fix $\widetilde{\mathcal{X}}^{(i)} := (x_j^{(i)})_{j \geq 1}$ as a dense countable subset of $B(z_i, 1/4h)$, the Euclidean ball with center $z_i$ and radius $1/4h$. Likewise, let $\widetilde{\mathcal{X}}^{(0)} := (x_j^{(0)})_{j \geq 1}$ be a dense countable subset of $\mathcal{X}_0$. Finally, let $0 < w \leq m^{-1}$, and define the marginal probability $\mu$ for all $x \in \mathcal{X}$ by

$$
\mu(\{x\}) := \begin{cases} C_s w 2^{-js} & \text{if } x = x_j^{(i)} \text{ for some } (i, j) \in \{1, \ldots, m\} \times \mathbb{N} \\ C_s(1 - mw)2^{-js} & \text{if } x = x_j^{(0)} \text{ for some } j \geq 1 \\ 0 & \text{else,} \end{cases}
$$

where $C_s := \left(\sum_{j \geq 1} 2^{-js}\right)^{-1}$ is a normalization constant. Observe that $\mu$ does not depend on $\vec{\sigma}$.

We now define for each $\vec{\sigma} \in \{-1, 1\}^m$ the regression function $\eta_{\vec{\sigma}} : x \mapsto \mathbb{E}_{(X,Y) \sim \mathbb{P}_{\vec{\sigma}}}[Y \mid X = x]$, which will fully determine $\mathbb{P}_{\vec{\sigma}}$. For all $x \in \mathcal{X}_j$, $1 \leq j \leq m$, we set $\eta_{\vec{\sigma}}(x) := \sigma_j \varphi(x)$, where $\varphi(x) := h^{-1}\phi(h[x - n_h(x)])$, and for $x \in \mathcal{X}_0$, we let $\eta_{\vec{\sigma}}(x) = 1$. Note that by Lemma 12, we have that Assumption (A5) is satisfied.

We now check the margin assumption (A1): fix $z_1 = (1/2h, \ldots, 1/2h) \in \mathcal{X}_1$. For any $\vec{\sigma} \in \{-1, 1\}^m$ we have

$$
\begin{aligned}
\mathbb{P}_{\vec{\sigma}}(|\eta_{\vec{\sigma}}(X)| \leq t) &= m\mathbb{P}_{\vec{\sigma}}(\phi(h[X - z_1]) \leq th) \\
&= m \sum_{x \in B(z_1, 1/4h)} \mathbb{1}_{\{\phi(h[X-z_1]) \leq th\}}(x)\mu(\{x\}) \\
&= C_s m w \sum_{j \geq 1} 2^{-js} \mathbb{1}_{\{\phi(h[X-z_1]) \leq th\}}(x_j^{(1)}) \\
&= C_s m w \sum_{j \geq 1} 2^{-js} \mathbb{1}_{\{\phi^{-1}([0,th])\}}(x_j^{(1)}/h + z_1) \\
&= m w \mathbb{1}_{\{th \geq 1\}}.
\end{aligned}
$$

We thus see that the low noise condition (A1) holds as long as $mw \leq Ch^{-q}$.

We are now ready to prove the lower bound. By applying Lemma 15, we have for any classifier $\hat{c}_n$

$$
\sup_{\mathbb{P} \in \mathcal{H}} \left\{ \mathbb{E}_{\mathbb{P}^{\otimes n}}\left[\mathcal{R}_{\mathbb{P}}(\hat{c}_n)\right] - \mathcal{R}_{\mathbb{P}}^* \right\} \geq \frac{1 - b\sqrt{nw}}{2} \, m \, w \, b', \tag{36}
$$

where $b = b' = h^{-1}$. Denote by $\alpha^* := s/(s + C_2 B_1 + B_2)$ the optimal rate, and remember that $q > 0$ denotes the noise exponent in Assumption (A1). By taking

$$
h = \begin{cases} \lceil n^{2/(d+q-2)} \rceil & \text{if } q \leq 1, \\ \lceil n^{\alpha^*/(q-1)} \rceil & \text{if } q > 1, \end{cases} \quad m = h^d, \quad w = h^{-d-q},
$$

and replacing in (36), we obtain the claimed lower bounds in equation 20, which proves the first case of Theorem 6.

We now address the case where we require the probability hypercube $\mathcal{H}$ to satisfy the stronger Assumption (A2). In that case, we can simply modify the previous construction as follows: define the grid $G_h$, partition $(\mathcal{X}_i)_{1 \leq i \leq m}$, functions $u, \phi$, and marginal probability distribution $\mu$ in the exact same way as before, but now let $\varphi(x) = \delta\phi(h[x - n_h(x)])$ for all $x \in \mathcal{X}$. It is straightforward to check that with these choices, all distributions in this probability hypercube satisfy Assumptions (A2) and (A5). Furthermore, $\mathcal{H}$ as defined is a $(m, w, b, b')$-hypercube with $b = b' = \delta$. Hence by taking

$$
h = \left\lceil m^{1/d} \right\rceil, \quad m = \max\{1, \lceil n^{1-\alpha^*} \rceil\}, \quad w = \frac{4}{9\delta^2 n},
$$

where $\alpha^* = s/(C_2 B_1 + B_2)$ is the optimal rate, and plugging these values into (36), we find that the lower bound (21) holds. The proof is thus complete.

**Acknowledgments**

The authors would like to thank the reviewers for their constructive feedback which greatly improved the quality of this paper. X.Z. acknowledges support from the Hong Kong General Research Funds (Grants No. 11304525, No. 11318522, and No. 11308323). N.T. was supported by the Hong Kong PhD Fellowship Scheme, which also funded a research visit to The University of Sydney where part of this work was carried out.

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

# A    Margin conditions for real-life datasets

The goal of this section is to empirically evaluate the validity of margin conditions on two widely-used benchmark classification datasets: Fashion MNIST and CIFAR-10. Since both datasets contain more than two classes, we fall back to the binary classification setting by restricting our analysis to two arbitrarily selected classes from each dataset. Following a similar approach to (Kim et al., 2021), we investigate the margin conditions in two ways: first, we generate interpolated images between the two classes and visually examine whether such interpolated samples, which are inherently harder to classify, could realistically appear in the original datasets. Second, we train a deep Convolutional Neural Network (CNN) with ReLU activation and squared loss until convergence, and analyze the histogram of its outputs for all images in the test set. For both cases, the resulting histograms strongly suggest that the weak margin condition **(A1)** hold.

The CNN used to obtain the numerical results consists of two convolutional layers with 32 and 64 filters of size $3 \times 3$, each followed by ReLU activation and $2 \times 2$ max pooling. The output of the second convolutional layer goes through a fully connected ReLU network with architecture $\mathbf{a} = (64 \times 8 \times 8, 128, 1)$.

## A.1    Fashion MNIST: "T-Shirt" vs "Pullover"

Interpolation Between 'T-shirt' and 'Pullover'

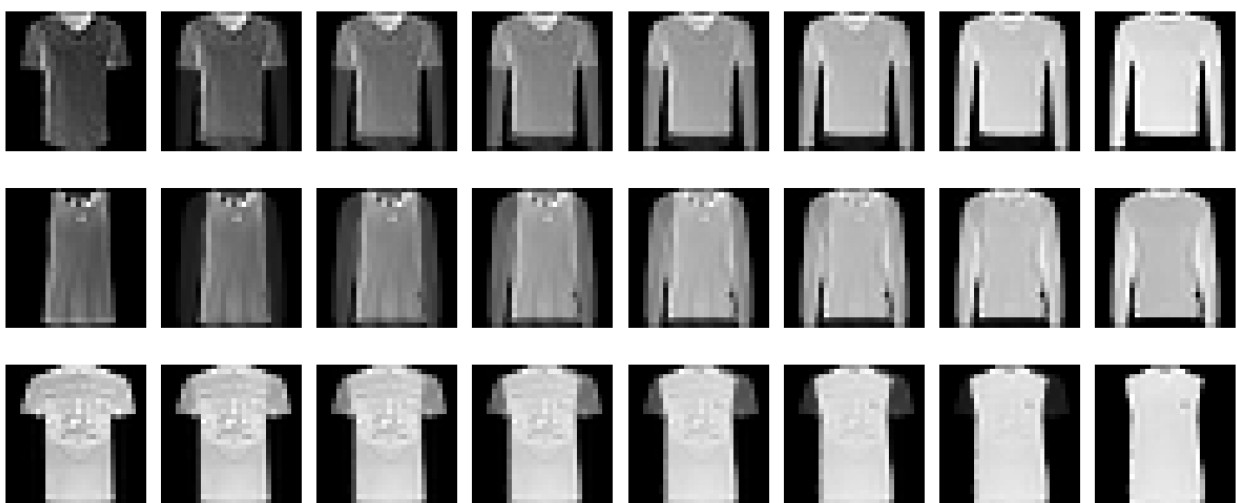

Figure 7: Interpolation of randomly selected images respectively in the "T-shirt" and "Pullover" class.

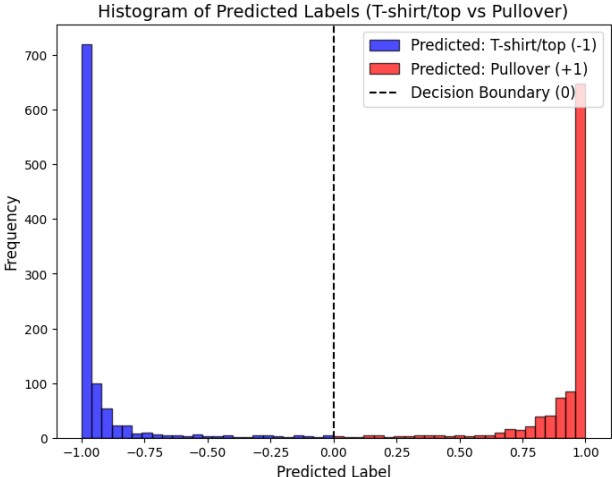

Figure 8: Histogram of predicted labels for images in the testing dataset: the histogram matches the function $t \mapsto C|t|^q$.

Despite the classes "T-Shirt" and "Pullover" exhibiting a relatively high level of visual similarity, the obtained histogram for the CNN approximation $\hat{\eta}$ of the Bayes regression function strongly suggests that the low-noise condition **(A1)** holds with a seemingly large exponent $q$.

## A.2 CIFAR-10: "Automobile" vs "Truck"

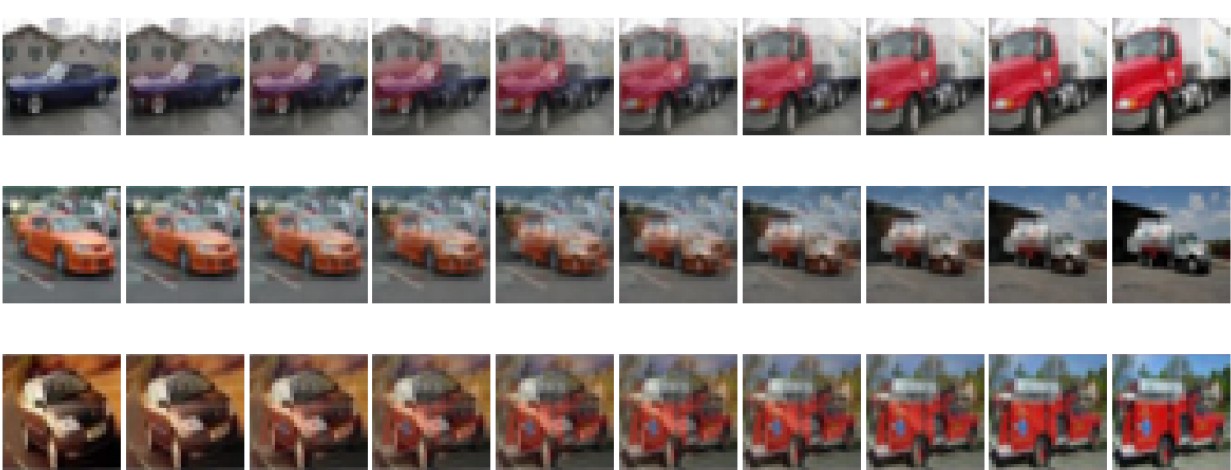

Figure 9: Interpolation of randomly selected images respectively in the "Automobile" and "Truck" class.

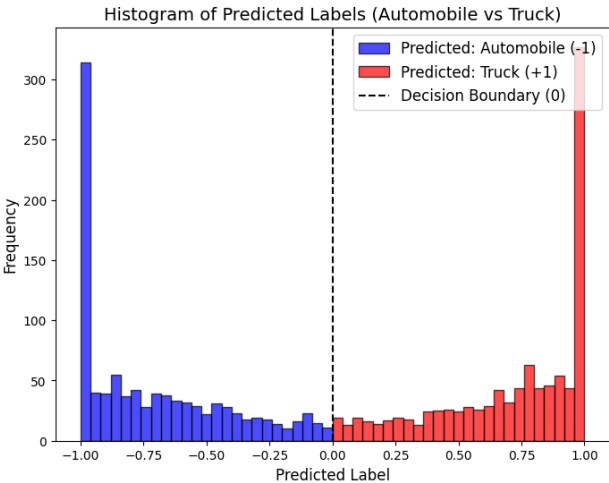

Figure 10: Histogram of predicted labels for images in the testing dataset: the decay as $t \to 0$ is much slower than for the Fashion MNIST dataset.

As for the Fashion MNIST dataset, the two selected classes have a relatively high amount of visual similarity. In this case, the histogram of labels predicted by CNN approximation $\hat{\eta}$ of the Bayes regression function suggests that the low-noise condition **(A1)** holds, but for noticeably lower values of the exponent $q$ and multiplicative constant $C$.

