# OpenReview forum: "Super-fast Rates of Convergence for Neural Network Classifiers under the Hard Margin Condition"
_TMLR — Accepted by TMLR_

### Review · Reviewer_oykH · 2026-01-05

**Summary Of Contributions:**

This paper investigates the binary classification problem using Deep Neural Networks (DNNs) with ReLU activation. The authors focus on the excess risk convergence rates under the "hard-margin condition". The study focuses on the statistical properties of the ERM solutions rather than the optimization dynamics.

**The key contributions are:**

*
**New Error Decomposition:** The authors introduce a new error decomposition for general Empirical Risk Minimization (ERM) based classifiers.


*
**Super-fast Rates:** Demonstrate that conventional DNNs can achieve "super-fast" rates of for  under hard-margin conditions, effectively breaking the standard  barrier $O(n^{-1})$.


*
**Minimax Optimality:** The authors establish lower bounds, proving that these super-fast rates are essentially optimal for the some considered hypothesis spaces.


*
**Exponential Convergence:** In a teacher-student setting where the teacher network is realizable by the student, the authors prove an exponential convergence rate.



**Key Strengths:**

- Provides a strong theoretical bridge explaining why DNNs outperform traditional methods in high-dimensional classification.


- Give lower bound results.

- Includes numerical experiments that validate the theoretical findings.



**Key Weaknesses:**

- The presentation of the "arbitrarily large " claim in the Abstract and Introduction lacks immediate clarity regarding its dependence on the smoothness parameter .

**Audience:**

Yes

**Audience Explanation:**

Understanding the theoretical limits of DNN generalization is a core interest of the TMLR community. This paper addresses a significant gap: why DNNs appear to defy the "curse of dimensionality" and why they can achieve rates faster than traditional kernel methods or local polynomial estimators in specific regimes.

**Broader Impact Concerns:**

NA. This is a theoretical paper focusing on the mathematical properties of convergence rates in classification. There are no immediate ethical concerns or negative social impacts arising from this work.

**Claims And Evidence:**

Yes

**Claims Explanation:**

The claims are supported by rigorous mathematical proofs, specifically Theorems 4 and 5. However, there is a stylistic lack of transparency in the early sections. While the Abstract claims  can be "arbitrarily large" , this is a limiting property tied to the smoothness parameter  and the capacity of the network. While technically accurate within the proof of Theorem 4, the Abstract and Introduction do not sufficiently emphasize that this "arbitrary" speed is not a fixed property but a function of the underlying distribution's regularity.

**Requested Changes:**

**1. Clarification of "Arbitrarily Large " (Critical):**
The authors should revise the Abstract and intro to explicitly state the functional dependency of  on the smoothness parameter  and the separation exponent . Currently, stating that  is "arbitrarily large" without qualification  may mislead readers into thinking this is a universal constant.

**2. Discussion on Network Capacity Trade-offs:**
To achieve these super-fast rates, the architecture (width  and depth ) must grow. The authors should include a brief discussion in the Introduction or Conclusion regarding the "price" of these rates.

**3. Intuition behind the Separation Exponent**
The parameter  (from the KL property/Proposition 1) plays a crucial role in the final rates. Providing more qualitative intuition on how  relates to the geometry of the risk landscape would significantly improve the paper's accessibility.

---

> ### Author Response · Authors · 2026-03-10
> **Response to Reviewer oykH**
>
> We thank the reviewer for their careful reading and for the positive remarks about our theoretical contributions and empirical validation. We appreciate the constructive suggestions for clarifying some points in the presentation. Below we respond to the requested changes in detail.
>
> ---
>
> **Clarification of "Arbitrarily Large $\alpha$:"**
>
> Regarding the clarification about the “arbitrarily large” growth of $\alpha$ in relation to $s$ and $r$:
>
> We appreciate the reviewer’s suggestion to clarify the dependencies. However, the expression of the best $\alpha$ given by our Theorem 4 involves a multiplicative constant $C_2$ that depends on $s$ in a way which is, as seen in Assumption A5, unknown. There is therefore no closed-form expression of $\alpha$ as a function of $s$ (or $r$) in either the hard- or weak-margin cases. We have therefore kept the "arbitrarily large" wording, as we believe it is a factually accurate reflection of the fact that $\alpha\to\infty$ whenever $s\to\infty$ in the hard-margin regime. That being said, as we do know that $\alpha$ converges to $1/r$ as $s \to \infty$ in the weak-margin case, we have now made this convergence explicit in the revised “Our contributions” section in the introduction (page 3).
>
> In an attempt to keep the abstract concise and accessible, we chose not to include these nuanced technical details there, which is standard practice in theory papers. Instead, we have opted to reword the abstract, making it clearer that $\alpha$ does not converge exactly to $1$, and we refer the interested readers to the discussion in the main text for a better grasp of the interactions between $s$, $r$ and the convergence rates.
>
> We hope this clarification addresses the concern effectively.
>
> ---
>
> **Discussion on Network Capacity Trade-offs:**
>
> We agree that the super-fast rates require network architectures (width, depth) to grow with the sample size $n$. The quantitative dependencies of the required architecture parameters on $n$ to achieve the stated rates are already given explicitly in Theorem 4. These bounds are quite standard and similar to those found in related recent works, such as:
> - Schmidt-Hieber, Johannes. "Nonparametric regression using deep neural networks with ReLU activation function." The Annals of Statistics 48.4 (2020): 1875.,
> - Kim, Yongdai, Ilsang Ohn, and Dongha Kim. "Fast convergence rates of deep neural networks for classification." Neural Networks 138 (2021): 179-197.,
> - Zhang, Zihan, Lei Shi, and Ding-Xuan Zhou. "Classification with deep neural networks and logistic loss." Journal of Machine Learning Research 25.125 (2024): 1-117.
>
> Because these network parameter scaling laws are well-known in the community and given explicitly in the main theorem, we believe there is limited added value in repeating or expanding on this discussion in the Introduction or Conclusion beyond a brief remark. Nonetheless, following the reviewer’s recommendation, we have reworded the Conclusion to highlight that the stated rates only hold for appropriately overparametrized network architectures.
>
> ---
>
> **Intuition behind the Separation Exponent $r$**
>
> We agree fully that providing qualitative intuition on the exponent $r$ is valuable and enhances accessibility.
>
> Accordingly, we have added a dedicated discussion immediately following Proposition 1 (page 11), which offers more insight into the geometric meaning of $r$ in relation and its role in shaping the risk landscape.
>
> ---
>
> We hope these clarifications address the reviewer’s remarks satisfactorily, and we thank them again for their constructive feedback.

---

### Review · Reviewer_6SjY · 2026-01-30

**Summary Of Contributions:**

The paper studies the binary classification problem within the class of feedforward neural networks, utilizing a squared error loss function. Initially, the authors derive an excess risk upper bound under weak and hard margin conditions when the class is general parametric, leveraging regularity assumptions. Subsequently, for the class of ReLU neural networks, they obtain sharp rates of convergence under low-noise conditions, supported by lower bounds analysis.


**Strengths**

* The paper demonstrates, for the first time in the literature, that feedforward neural networks can achieve super-fast convergence rates under regularity conditions

* As a valuable byproduct, the authors provide a proof of exponential convergence rates within the knowledge distillation setting, under the stated regularity conditions

**Weaknesses**

* Assumption (A3) appears inconsistent with the mode connectivity phenomenon observed in practice [1], where minima are connected via simple paths. The reliance on this assumption facilitates the use of concentration inequalities within the parameter space. However, the potential for removing it remains unclear, representing a key limitation.

* As for Assumptions (A5) and (A6), the set of Bayes regression functions $\eta$ for which this condition hold is not well characterized. Section 6.4 provides only a limiting discrete example. An extension to more general settings and, if possible, a simple sufficient criterion for (A5) would strengthen the analysis

* The introduction of Section 3.3 stated that Assumptions (A1) and (A2) guarantee sharp convergence rates. However, Theorem 6 is formulated exclusively with Assumption (A2), failing to provide the necessary guarantees when Assumption (A1) is satisfied

* The lower bounds are tight only when (A2) holds and, for (A1), when $q \geq 2$. The manuscript asserts tightness for all $q > 1$, which is false. Under (A1), for $q \in (1, 2)$ the upper bound is $O(n^{-q\alpha / (2r)})$, which is looser than the lower bound $O(n^{-\alpha / r})$

**References**

[1] Garipov, Timur, et al. 2018. “Loss Surfaces, Mode Connectivity, and Fast Ensembling of DNNs.” In *Advances in Neural Information Processing Systems* (NIPS), 31.

**Audience:**

Yes

**Audience Explanation:**

The question of how rapidly a feed‑forward neural network generalizes is a cornerstone of both statistical theory and practical machine learning, and it is highly relevant to the TMLR audience.

**Claims And Evidence:**

Yes

**Claims Explanation:**

Overall, the paper delivers solid theoretical support through carefully detailed, self‑contained proofs.

**Requested Changes:**

* The convergence rate in Theorem 4 includes a hidden constant, which is independent of the choice of $s$. A careful inspection of the proof confirms this. This critical detail should be explicitly mentioned. Therefore, when $s$ is taken sufficiently large, it does not destroy the rate.

* The hyperlink from Theorem 4 in the contributions (page 3) currently jumps to the technical Lemma 4. This must be corrected.

* The paragraph following Theorem 2 contains no citations. Please add a brief comparison with existing bounds for related problems and provide the appropriate references.

* On page 9, replace "Lemma 1" with "Proposition 1".

* On page 4, rename the optimal risk as $R^*$ (using an upper index) to align with the notation used later in the manuscript.

---

> ### Author Response · Authors · 2026-03-10
> **Response to Reviewer 6SjY**
>
> We thank the reviewer for their extremely careful and thorough reading of our work, and for the insightful comments and constructive suggestions. We address each point below.
>
> ---
>
> **Assumption (A3) and Mode Connectivity**
>
> We agree that Assumption (A3) on the finiteness of the set of minimizers appears inconsistent with the mode connectivity phenomenon observed in practice [1]. This is an excellent remark. We have found a new proof of Proposition 1 which no longer relies on Assumption (A3). As a result, *this limiting assumption has been completely removed from the updated manuscript*. Please see pages 23–25 for the updated proof. We thank the reviewer for highlighting this important issue.
>
> ---
>
> **Characterization and Extensions of Assumptions (A5) and (A6)**
>
> We concur that Assumption (A5) (now Assumption (A3) in the revised manuscript) about boundedness of minimizers was poorly characterized in the initial version of the manuscript, and have now added three concrete examples based on linear and kernel classification in Remark 1 (pages 8–9) for which this boundedness assumption can be verified, in order to provide readers with better intuition.
>
> Likewise, we fully agree that the set of pairs $(\eta,\rho_X)$ satisfying Assumption (A6) (now Assumption (A5)) was insufficiently characterized before. We have now added a Section 6.5, dedicated to four non-trivial examples where this assumption holds, with full proofs provided. We acknowledge that a complete characterization of the validity of this assumptions remains a challenging open problem, which we note explicitly in the manuscript.
>
> ---
>
> **Formulation of Theorem 6 Regarding Assumptions (A1) and (A2)**
>
> This is a valid point. The goal of Section 3.5 was to show that, in an idealized (but still somewhat plausible) setting, the hard-margin condition could lead to exponential rates of convergence, as has been observed for other hypothesis spaces discussed in the paper. Theorem 6 (now Theorem 10) therefore focuses on the exponential convergence rate under Assumption (A2) only.
>
> The analogue of Theorem 10 under Assumption (A1) would still be the “nearly fast-rate” of $1/r$, which is the same as the limit $s\to\infty$ of the rate given by Theorem 4, and therefore offers little to no new insights. To avoid undue distraction from the main result of Section 3.5, we therefore chose not to state this case explicitly in the theorem, and have instead added some comments addressing this (see Remark 2 on page 19).
>
> ---
>
> **Tightness of Lower Bounds for Different Values of $q$**
>
> The reviewer correctly points out that the lower bounds are tight only when $(A2)$ holds and for $(A1)$ when $q \geq 2$. The manuscript incorrectly asserted tightness for all $q > 1$. This was an oversight on our part and has now been corrected in the discussion following Theorem 6 on page 17. We thank you for this correction.
>
> ---
>
> **Hidden Constant in Theorem 4’s Convergence Rate**
>
> We respectfully did not fully understand the reviewer’s concern here. The hidden constant in Theorem 4, even if it depended on $s$, does not depend on $n$ and therefore does not affect the convergence rate in terms of $n$.
>
> ---
>
> **Correction of Hyperlink for Theorem 4**
>
> Thank you for noticing this error. The hyperlink from Theorem 4 in the contributions section (page 3) is now correctly linked to Theorem 4 itself instead of Lemma 4.
>
> ---
>
> **References After Theorem 2**
>
> A discussion with comparisons to relevant literature on “standard oracle inequalities” at the end of Theorem 2, on page 13, has now been added.
>
> ---
>
> **Terminology and Notation Corrections**
> We thank the reviewer for noticing these typo's:
> - On page 11, what was formerly "Lemma 1" has been corrected to "Proposition 1".
> - On page 4, we have correctly renamed the optimal risk as $R^*$ (using an upper index).
>
> ---
>
> We hope these revisions and clarifications address the reviewer’s comments satisfactorily. We thank them again for their valuable and detailed feedback which helped strengthen our paper.

---

### Review · Reviewer_KiFG · 2026-02-24

**Summary Of Contributions:**

This paper investigates the binary classification problem using DNNs with ReLU activations. The authors propose general framework to obtain the excess risk convergence rate for empirical risk minimizers under a regularized square loss.

Following the classic Tsybakov’s low-noise condition, two margin conditions, weak margin condition (A1) and hard-margin condition (A2), are considered that characterize how separated the two classes are. A key technical contribution is a novel error decomposition for the excess risk of classifiers induced by general parametric functions.

Using this framework, the authors establish both upper and lower bounds for convergence. Under the weak margin condition, the authors showed that ReLU neural networks can achieve fast convergence rate, while under the hard-margin condition, super-fast rate of order $\mathcal{O}(n^{-\alpha})$ for arbitrarily large $\alpha > 1$ can be achieved, which is shown to be minimax-optimal.

Additionally, the authors considered an idealized teacher-student learning scenario where they prove that if the "teacher" network is exactly realizable by the "student" network, the excess risk converges at an exponential rate ($\mathcal{O}(e^{-\beta n})$).

### Strength:

- Solid analytical framework: The paper provides a comprehensive approach to analyzing convergence rates through a novel error decomposition
- Significant theoretical findings: The super-fast convergence rate under the hard-margin condition is a notable contribution. To the best of my knowledge, similar results have only been shown for a stronger margin condition where the supports of the two classes are separated with positive margin.

### Weakness:

- Limited scope: The framework relies on the square loss and does not currently extend to the dominant cross-entropy loss. Also some assumptions about the estimatior family appear very strong.
- Practical relevance: While the "super-fast" rates are theoretically sound, the hard-margin condition and idealized teacher-student settings represent highly optimistic scenarios that may not fully reflect the complexities of typical real-world datasets.

**Audience:**

Yes

**Audience Explanation:**

Even though this is a highly technical paper, I think this paper would still be interesting to people working with statistical learning theory of deep neural network classifiers.

It addresses a meaningful gap in existing literature by showing that conventional DNNs can achieve "super-fast" convergence rates (faster than $\mathcal{O}(n^{-1})$) under hard-margin conditions. Meaningful extension can be made to further provide insights for classifiers built with neural networks.

**Broader Impact Concerns:**

No concerns.

**Claims And Evidence:**

Yes

**Claims Explanation:**

The claims in this submission are supported by both theoretical analysis and empirical validation.

Notations and assumptions are clearly presented. The authors provide detailed mathematical proofs for all primary lemmas and theorems.
Theoretical findings are supplemented by numerical experiments on toy datasets that empirically confirm the predicted convergence slopes.

**Requested Changes:**

My primary concern is the practical relevance of assumption (A3) (finiteness of minimizers) and (A5) (existence and boundedness of minimum-norm minimizers).

This work is targeting on classifiers built on neural networks. Modern deep neural networks are often overparameterized and there is a line of work studying the inherent symmetries in the parameter space [1]. For instance, if the neural network has $W_1 \cdot W_2$ in the parameterization (e.g., attention mechanism in transformers), scaling such as $(c \cdot W_1)(W_2/c)$ doesn't change the function. If there are some transformations on the parameter that doesn’t change the neural network as a function, both Assumption (A3) and (A5) would be violated.
Please discuss how this will change the theoretical analysis.


Below are some minor suggested changes:

Writing:

- In page 8, before introducing (A3), it says “satisfies the two following assumptions”. But (A4) is not shown until one page later. I suggest moving (A3) and (A4) closer and move technical materials to earlier sections.
- In Definition 4, the wording for Fréchet subdifferential is a bit confusing.
- In Proposition 1, the parameter $r$ is very important in the technical proofs. But how does $r$ depends on the estimation problem is not clear. A brief discussion on $r$ would be helpful for the reader.
- In Lemma 5, how to ensure that the range of $f$ is between [-1, 1] is not clear.

Typo:

- In page 7, (A2), it should be $|\eta(X)-1/2|$ instead of $\eta(X)$.
- In page 9, after proposition 1, it should be Proposition 1 instead of Lemma 1.
- In Lemma 5 of page 15, it should be $a^* \in N^{L^*+1}$

Reference:

[1] Zhao, Bo, Robin Walters, and Rose Yu. "Symmetry in neural network parameter spaces." TMLR.

---

> ### Author Response · Authors · 2026-03-10
> **Response to Reviewer KiFG**
>
> We thank the reviewer for the careful reading and valuable comments and suggestions. We address the points raised below.
>
> ---
>
> **Limited scope: reliance on square loss and strong assumptions**
>
> We agree with the concern raised by the reviewer that our current analysis only covers the square loss, and not the more popular cross-entropy loss. This limitation is due to the technical tools we use to establish our main theorem, which, as far as we can tell, are not available for other surrogate losses. As a recent work [2] has established fast-rates for DNNs trained with cross-entropy loss, we intend to investigate in future work whether their analysis can be combined with ours to derive super-fast rates in the hard-margin regime as well.
>
> Regarding the strength of some assumptions, we refer the reviewer to our detailed responses below.
>
> ---
>
> **Practical relevance: hard-margin condition and teacher-student settings**
>
> We agree that these assumptions are somewhat idealized and do not fully capture the complexities of real-world datasets. Nevertheless, the main goal of this work is to address an important theoretical gap by establishing super-fast convergence rates for deep neural network classifiers. Previously, such rates have only been proven for hypothesis spaces that are known to be weaker than neural networks in practice. Viewed from this perspective, our hard-margin condition and teacher-student setting represent assumptions of comparable strength to those used in related works proving similar results:
>
> - Smale and Zhou. *"Learning theory estimates via integral operators and their approximations."* Constructive approximation (2007)
> - Audibert and Tsybakov. *"Fast learning rates for plug-in classifiers."* Annals of statistics (2007)
> - Vigogna et al. *"Multiclass learning with margin: exponential rates with no bias-variance trade-off."* ICML 2022.
>
> We believe that our idealized assumptions are of similar strength and spirit to those in the referenced works.
>
> ---
>
> **On assumptions (A3) and (A5) and symmetries in parameter space**
>
> This is a crucial observation, and we thank the reviewer for bringing it up. We have now found a new proof of Proposition 1 that *completely removes the need for Assumption (A3)*. This assumption is thus eliminated from the updated manuscript. Please see pages 23–25 for the revised proof.
>
> As for Assumption (A5) (renamed as (A3) in the updated manuscript) on boundedness of minimizers, we respectfully believe that the presence of parameter symmetries does not invalidate it. This assumption focuses on boundedness of *minimum-norm minimizers*, which can still hold even if multiple equivalent parameterizations exist. To clarify this point, we provide three illustrative examples in Remark 1 (pages 8–9).
>
> ---
>
> **Minor writing suggestions**
>
> - The phrase before (A3) on page 8 referring to “the two following assumptions” is now obsolete since (A3) (as it appeared in the previous version of the manuscript) has now been removed.
>
> - Regarding Definition 4 and the Fréchet subdifferential, we aimed for a minimal presentation that suffices to define the necessary objects. We therefore adopted the definition from Rockafellar & Wets, allowing interested readers to easily refer to the book for a deeper understanding.
>
> ---
>
> **Parameter $r$ in Proposition 1**
>
> We agree that the meaning of $r$ is crucial and may not be immediately clear. We have added qualitative remarks about the parameter $r$ right after its introduction in Proposition 1 (page 11) for better intuition.
>
> ---
>
> **Ensuring the range of $f$ in Lemma 5**
>
> Ensuring that $f$ stays within a given range can be achieved via clipping the outputs, which is implicitly assumed throughout this work, as discussed in Section 2.2.2. Moreover, this clipping operation can be implemented by a depth-3 ReLU network, as shown in Lemma A.2 of [3].
>
> ---
>
> **Typographical corrections**
>
> - On page 7, we believe that (A2) should remain as stated, since the two classes are $\pm 1$ rather than $\{0,1\}$. Hence, the "bad region" is where $|\eta|$ is close to zero, rather than $1/2$.
>
> - On page 9, after Proposition 1, "Lemma 1" has now been corrected to "Proposition 1."
>
> - In Lemma 5 on page 15, the correction $a^* \in \mathbb{N}^{L^* + 1}$ has now been made.
>
> ---
>
> We hope these responses address the reviewer’s comments thoroughly. We thank the reviewer once again for the thoughtful feedback that helped improve our work.
>
> [2] Zhang, Zihan, Lei Shi, and Ding-Xuan Zhou *"Classification with deep neural networks and logistic loss."* JMLR 25.125 (2024): 1-117
> [3] Berner, Julius, Philipp Grohs, and Arnulf Jentzen. *"Analysis of the generalization error: Empirical risk minimization over deep artificial neural networks overcomes the curse of dimensionality in the numerical approximation of Black--Scholes partial differential equations."* SIAM Journal on Mathematics of Data Science 2.3 (2020): 631-657.

---

### Author Response · Authors · 2026-03-10
**Summary of Revisions**

We would like to sincerely thank all the reviewers for their time and insightful feedback, which has greatly helped us improve the manuscript.

We have uploaded the revised version of the paper, addressing most of the issues and suggestions raised during the review process as well as our internal discussions, notably on the strength of some Assumptions and the lack of sufficient examples. We would also like to highlight that **our main result has been extended from ReLU networks to networks using essentially all (Lipschitz) activations used in practice**. For tanh and sigmoid networks, the rates even hold when the distribution-adapted smoothness assumption on $\eta$ is replaced by the more standard $\eta\in \mathcal{C}^s$.

Below is a summary of the main changes compared to the previous version:

- **Assumption (A3) on the finiteness of the set minimizers has been removed**. We realized the proof of Proposition 1 can be carried out without this assumption. The updated proof is given in Section 6.2 on pages 23–25.

- Three concrete examples where the previous Assumption (A5) on boundedness of minimizers holds are now included, based on linear and kernel classification settings. See Remark 1 on pages 8–9.

- The main theoretical results have been extended beyond ReLU activation functions by leveraging the recent work of Zhang, Lu, and Zhao (JMLR 2024). This allows us to transfer both "fast" and "super-fast" rates from ReLU networks to essentially all Lipschitz activations commonly used in practice. See Theorem 5 and Corollary 1 on page 16.

- Using an approximation result for shallow tanh networks by De Ryck, Lanthaler, and Mishra (Neural Networks 2021), we derive an equivalent of Theorem 4 for tanh networks under the standard $\mathcal{C}^s$-smoothness assumption on the regression function, as opposed to distribution-adapted smoothness. See Section 3.4 and Theorem 8 (page 17).

- We provide three additional examples where our updated “distribution-adapted” smoothness assumption (now Assumption (A5)) provably holds, including detailed proofs in Section 6.5 (page 34).

We hope these extensive revisions address the reviewers’ concerns effectively, and we look forward to any further feedback.

---

### Decision · Action_Editor_H1qG · 2026-05-01

**Recommendation:** Accept as is

**Additional Comments:**

The paper is technically very strong and the most natural consideration (among the three non-exclusive choices) would be for "Featured Certification". Yet, the restriction to squared loss does not quite meet the bar for something on the level of a spotlight presentation at a conference (which is the minimum bar for a featured certification).

**Audience:**

Yes

**Audience Explanation:**

I completely stand behind the following quote from one reviewer:
> "The question of how rapidly a feed‑forward neural network generalizes is a cornerstone of both statistical theory and practical machine learning, and it is highly relevant to the TMLR audience."

There would be higher interest if the authors could go beyond squared loss, but already this work is strong enough (especially with the highly satisfying avoidance of Assumption A3). Well done!

**Claims And Evidence:**

Yes

**Claims Explanation:**

All reviewers are satisfied with the quality of the theoretical results as well as the numerical experiments.  Moreover, the authors' revised paper is great in that it avoids the need for Assumption A3 (which is also related to the interest from the audience).